# UNCERTAINTY-DRIVEN EMBEDDING CONVOLUTION

**Sungjun Lim    Kangjun Noh    Youngjun Choi    Heeyoung Lee    Kyungwoo Song**[*]

Yonsei University

## ABSTRACT

Text embeddings are essential components in modern NLP pipelines. Although numerous embedding models have been proposed, no single model consistently dominates across domains and tasks. This variability motivates the use of ensemble techniques to combine complementary strengths. However, most existing ensemble methods operate on deterministic embeddings and fail to account for model-specific uncertainty, limiting their robustness and reliability in downstream applications. To address these limitations, we propose **Uncertainty-driven Embedding Convolution (UEC)**. UEC first transforms deterministic embeddings into probabilistic ones in a post-hoc manner. It then computes adaptive ensemble coefficients based on embedding uncertainty, derived from a principled surrogate-loss formulation. Additionally, UEC employs an uncertainty-aware similarity function that directly incorporates uncertainty into the similarity scoring, providing a theoretically grounded and efficient surrogate to distributional distances. Extensive experiments on diverse benchmarks demonstrate that UEC consistently improves both performance and robustness by leveraging principled uncertainty modeling.[1]

## 1 INTRODUCTION

Embeddings are core building blocks in modern NLP, capturing semantic meaning for words, sentences, and documents. They support tasks like similarity (Gao et al., 2021), retrieval (MacAvaney et al., 2019), QA (Devlin et al., 2019), and classification (Cer et al., 2018). Numerous embedding models (Devlin et al., 2019; Liu et al., 2019; Reimers & Gurevych, 2019) have emerged with diverse architectures and training objectives, but their performance varies across tasks and domains. No single model excels universally; instead, models offer complementary strengths. This motivates combining multiple embeddings to leverage their diverse capabilities.

Embedding models can be ensembled at two levels: parameter and representation. At the parameter level, techniques like model merging combine model weights (Wortsman et al., 2022; Ilharco et al., 2022; Yang et al., 2024). However, this approach is often limited by strict architectural constraints. On the other hand, combining embeddings at the representation level, i.e., ensembling the output embeddings themselves, is a more practical and widely applicable strategy. However, most existing ensemble approaches rely on deterministic aggregation methods such as uniform averaging, which fail to consider the reliability or uncertainty of individual embeddings (Khan et al., 2023; Fang et al., 2019). Without modeling uncertainty, these methods treat all embeddings as equally reliable, which can lead to suboptimal or unstable performance, especially when some models are poorly calibrated or mismatched to the target task. This is evident in Figure 1, where averaging conflicting 'animal' and 'car' interpretations for 'jaguar' leads directly to a retrieval failure.

To address these limitations, we propose **Uncertainty-driven Embedding Convolution (UEC)**, a framework for combining embeddings in a principled, uncertainty-aware manner. UEC consists of three key components. First, it converts pre-trained deterministic embeddings into probabilistic embeddings in a post-hoc fashion, allowing each embedding to represent both its mean and uncertainty. Second, it computes adaptive ensemble weights based on estimated uncertainty, down-weighting less reliable embeddings. This weighting strategy is grounded in a principled, uncertainty-aware surrogate-loss formulation. Third, UEC introduces an uncertainty-aware similarity function that incorporates both distance and variance into the similarity score, offering a theoretically grounded and

---

[*]Corresponding Author: `kyungwoo.song@gmail.com`

[1]Code is available at: `https://github.com/MLAI-Yonsei/UEC`.

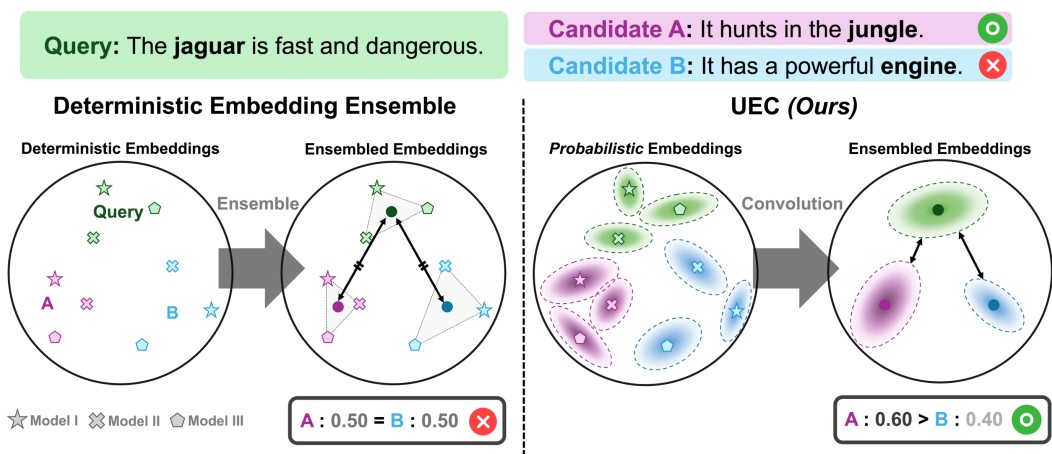

Figure 1: Comparison of deterministic embedding ensemble and UEC. Deterministic ensemble (left) uniformly averages embeddings without considering their reliability, often leading to suboptimal decisions. In this example, both candidate embeddings contribute equally, resulting in an incorrect retrieval. In contrast, the proposed UEC (right) adjusts weights based on the uncertainty, assigning higher importance to the more reliable candidate and successfully retrieving the correct answer.

efficient surrogate to distributional distances. Consequently, empirical results on diverse benchmarks confirm that UEC reliably improves performance and robustness by leveraging principled uncertainty modeling.

In summary, our key contributions are as follows:

- We propose a post-hoc mechanism to transform pre-trained deterministic embeddings into probabilistic representations, thereby enabling inherent uncertainty quantification for diverse existing embedding models.
- We introduce the UEC framework, which adaptively and data-dependently combines multiple embeddings by weighting their contributions based on estimated, query-specific uncertainties, following a principled surrogate-loss formulation.
- We develop an uncertainty-aware similarity function that explicitly incorporates embedding variance into the similarity scoring process, serving as a theoretically grounded and efficient surrogate to distributional distances.

## 2 RELATED WORKS

### 2.1 EMBEDDING ENSEMBLE

Embedding ensemble methods aim to improve performance and robustness by combining multiple pre-trained models. Shuang et al. (2019) proposed CDWE, which handles polysemy by generating multiple word prototypes through deconvolution. Fang et al. (2019) enhanced knowledge-aware embeddings by re-weighting knowledge-graph edges. Liu et al. (2025) fused BERT (Devlin, 2018) variants using cross-attention guided by pseudo-labels. Sahlgren (2021) distilled multiple encoders into one by matching their averaged outputs. Khan et al. (2023) explored fusion strategies for domain-specific BERT models, yielding minor improvements. Despite these advances, most approaches rely on deterministic embeddings, limiting their ability to model uncertainty. In contrast, our method ensembles probabilistic embeddings, capturing both distributional information and model diversity through principled uncertainty estimation.

### 2.2 PROBABILISTIC EMBEDDING

Probabilistic embeddings represent inputs as distributions rather than points, thereby modeling uncertainty in representation learning. Early approaches such as Gaussian (Vilnis & McCallum, 2014), Gaussian mixture (Chen et al., 2015), and Bayesian embeddings (Barkan, 2017) captured word-level distributional semantics. Sen2Pro (Shen et al., 2023) extended this idea to sentences using pre-trained models, but it requires fine-tuning, lacks post-hoc conversion, and does not support scalable similarity.

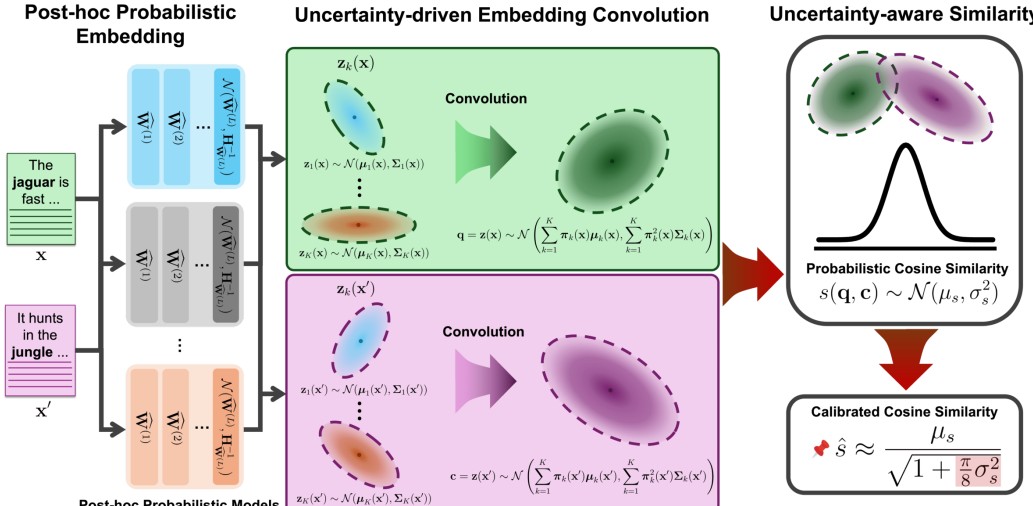

Figure 2: Overview of the UEC framework: UEC first transforms deterministic embeddings from multiple encoder models into probabilistic representations using Laplace approximation. These probabilistic embeddings are then adaptively combined by computing uncertainty-driven ensemble coefficients based on per-dimension variances. Finally, similarity is measured using an uncertainty-aware metric that accounts for both the mean and uncertainty of the ensembled embedding.

Subsequent works explored multimodal and vision–language tasks: Chun et al. (2021); Chun (2023) improved cross-modal retrieval and image–text alignment, yet remained modality-specific; Prob-VLM (Upadhyay et al., 2023) and BayesVLM (Baumann et al., 2024) incorporated uncertainty into frozen vision–language models through adapters or post-hoc strategies, but with limited generality. Other studies investigated hedging against ambiguous inputs (Oh et al., 2018), compositional multi-modal retrieval (Neculai et al., 2022), uncertainty-aware multimodal pre-training (Ji et al., 2023), and scalable probabilistic embeddings with Gaussian process latent variable models (Venkataramanan et al., 2025). Our work provides a Gaussian convolution formulation that not only generates probabilistic embeddings in a post-hoc manner but also supports theoretically grounded and lightweight similarity estimation for downstream tasks.

## 3 UNCERTAINTY-DRIVEN EMBEDDING CONVOLUTION

We propose **Uncertainty-driven Embedding Convolution (UEC)**, a principled framework for combining multiple embedding models by modeling predictive uncertainty.

UEC performs three simple steps (Figure 2): (1) converts each deterministic embedding model into a probabilistic one that outputs a Gaussian embedding (Section 3.1), (2) combines these probabilistic embeddings so that more confident models contribute more (Section 3.2), and (3) computes an uncertainty-aware similarity score (Section 3.3). This enables a robust and adaptive ensemble without manual tuning.

### 3.1 POST-HOC PROBABILISTIC EMBEDDING MODEL

**Laplace Approximation**  To convert a deterministic embedding model into a probabilistic one, we estimate a Gaussian posterior over its last-layer parameters using the Laplace Approximation (LA) (MacKay, 1992; Ritter et al., 2018). Given training data $\mathcal{D}$ and last-layer weights $\mathbf{W}^{(L)}$, LA constructs a second-order Taylor expansion of the negative log-posterior $-\log p(\mathbf{W}^{(L)}|\mathcal{D})$ around the maximum a posteriori (MAP) estimate $\widehat{\mathbf{W}}^{(L)}$:

$$-\log p(\mathbf{W}^{(L)}|\mathcal{D}) \approx -\log p(\widehat{\mathbf{W}}^{(L)}|\mathcal{D}) + \frac{1}{2}(\mathbf{W}^{(L)} - \widehat{\mathbf{W}}^{(L)})^\top \mathbf{H}_{\widehat{\mathbf{W}}^{(L)}}(\mathbf{W}^{(L)} - \widehat{\mathbf{W}}^{(L)}),$$

where $\mathbf{H}_{\widehat{\mathbf{W}}^{(L)}}$ is the Hessian of the negative log-posterior evaluated at the MAP. This MAP solution corresponds to the final-layer parameters of the pre-trained embedding model. Prior works show that final-layer LA is effective and efficient (Daxberger et al., 2021; Hobbhahn et al., 2022), yielding a Gaussian weight posterior without retraining. The full derivation of LA is provided in Appendix A.1.

**Gaussian Embedding Generation**    Building on the LA posterior derived above, we convert the pre-trained embedding model into a probabilistic one in a post-hoc manner. Let $f(\mathbf{x})$ be an embedding model with last-layer parameters $\mathbf{W}^{(L)}$ and penultimate representation $\mathbf{h}^{(L-1)}(\mathbf{x})$. In the deterministic setting, the embedding is $\mathbf{z}(\mathbf{x}) = \mathbf{W}^{(L)\top}\mathbf{h}^{(L-1)}(\mathbf{x})$. By applying last-layer LA to pre-trained deterministic embedding model, we can convert it to probabilistic embedding model where the last-layer following Gaussian posterior $p(\mathbf{W}^{(L)} \mid \mathcal{D}) \approx \mathcal{N}\left(\widehat{\mathbf{W}}^{(L)}, \mathbf{H}_{\widehat{\mathbf{W}}^{(L)}}^{-1}\right)$. The embedding produced by the model becomes a Gaussian random vector obtained by propagating the probabilistic last-layer through the fixed representation:

$$\mathbf{z}(\mathbf{x}) \sim \mathcal{N}\left(\widehat{\mathbf{W}}^{(L)\top}\mathbf{h}^{(L-1)}(\mathbf{x}),\ \mathbf{h}^{(L-1)}(\mathbf{x})^\top \mathbf{H}_{\widehat{\mathbf{W}}^{(L)}}^{-1}\mathbf{h}^{(L-1)}(\mathbf{x})\right).$$

We adopt a diagonal approximation to $\mathbf{H}_{\widehat{\mathbf{W}}^{(L)}}$ for computational efficiency, following prior works (Daxberger et al., 2021; Zhdanov et al., 2025). These Gaussian embeddings are then leveraged to construct an uncertainty-aware convolution across multiple embedding models.

### 3.2    Uncertainty-driven Coefficients for Embedding Convolution

**Gaussian Convolution**    Suppose we have $K$ independent embedding models, each transformed through the procedure in Section 3.1 into a probabilistic embedding model that produces $\mathbf{z}_k(\mathbf{x}) \sim \mathcal{N}(\boldsymbol{\mu}_k(\mathbf{x}), \boldsymbol{\Sigma}_k(\mathbf{x}))$. To integrate these into a single representation, we perform a *Gaussian convolution* in the embedding space. Formally, we define the convolutional embedding $\mathbf{z}(\mathbf{x})$ as the result of a weighted combination of independent Gaussian variables:

$$\mathbf{z}(\mathbf{x}) = \sum_{k=1}^{K} \boldsymbol{\pi}_k(\mathbf{x}) \cdot \mathbf{z}_k(\mathbf{x}),$$

where $\boldsymbol{\pi}_k(\mathbf{x})$ denotes the convolution coefficient assigned to model $k$, and $\sum_{k=1}^{K} \boldsymbol{\pi}_k(\mathbf{x}) = \mathbf{1}$. Since each $\mathbf{z}_k(\mathbf{x})$ is Gaussian and the coefficients $\boldsymbol{\pi}_k(\mathbf{x})$ are deterministic, the resulting embedding $\mathbf{z}(\mathbf{x})$ follows a Gaussian distribution as well, according to the properties of linear combinations of independent Gaussians (Shao, 2008):

$$\mathbf{z}(\mathbf{x}) \sim \mathcal{N}\left(\sum_{k=1}^{K} \boldsymbol{\pi}_k(\mathbf{x})\boldsymbol{\mu}_k(\mathbf{x}), \sum_{k=1}^{K} \boldsymbol{\pi}_k^2(\mathbf{x})\boldsymbol{\Sigma}_k(\mathbf{x})\right). \tag{1}$$

Gaussian convolution aggregates the centers (means) and spreads (covariances) of the input Gaussians into a single distribution. This process inherently allows for the propagation of uncertainty in a closed form, enabling the ensemble to capture and represent epistemic uncertainty arising from disagreements or variations across different models.

**Uncertainty-aware Coefficients**    To determine the coefficients $\boldsymbol{\pi}_k(\mathbf{x})$ for Gaussian convolution, we introduce an uncertainty-aware surrogate loss that captures how reliably the embeddings of positive pairs are expected to align. This formulation enables a principled aggregation of multiple embedding models while explicitly accounting for model-specific epistemic uncertainty.

Text embeddings are often trained with contrastive objectives such as InfoNCE (Oord et al., 2018). For $\ell_2$-normalized features, the squared Euclidean distance is directly related to cosine similarity, $\|\mathbf{u} - \mathbf{v}\|^2 = 2(1 - \mathbf{u}^\top \mathbf{v})$ (Wang & Isola, 2020). These losses also encourage alignment of positive pairs and uniformity on the unit hypersphere. Leveraging this connection, we adopt the squared loss as a principled surrogate for contrastive objectives.

Let $(\mathbf{x}, \mathbf{x}')$ denote a positive query–document pair. For each model $k$, the corresponding embeddings follow $\mathbf{z}_k(\mathbf{x}) \sim \mathcal{N}(\boldsymbol{\mu}_k(\mathbf{x}), \boldsymbol{\Sigma}_k(\mathbf{x}))$, $\mathbf{z}_k(\mathbf{x}') \sim \mathcal{N}(\boldsymbol{\mu}_k(\mathbf{x}'), \boldsymbol{\Sigma}_k(\mathbf{x}'))$. We define the uncertainty-aware surrogate loss, a multi-task objective that aggregates model-specific embedding errors weighted by uncertainty-aware task coefficients, as:

$$\mathcal{L}_{\text{sur}}(\boldsymbol{\pi}; \mathbf{x}, \mathbf{x}') = \sum_{k=1}^{K} \boldsymbol{\pi}_k(\mathbf{x})\, \mathbb{E}_{\mathbf{z}_k(\mathbf{x})}\left[\|\mathbf{z}_k(\mathbf{x}) - \mathbf{z}_k(\mathbf{x}')\|^2\right]$$

$$= \sum_{k=1}^{K} \boldsymbol{\pi}_k(\mathbf{x})\left(\underbrace{\|\boldsymbol{\mu}_k(\mathbf{x}) - \boldsymbol{\mu}_k(\mathbf{x}')\|^2}_{\text{Fidelity}} + \underbrace{\text{tr}(\boldsymbol{\Sigma}_k(\mathbf{x})) + \text{tr}(\boldsymbol{\Sigma}_k(\mathbf{x}'))}_{\text{Uncertainty}}\right). \tag{2}$$

This surrogate loss decomposes into fidelity and epistemic uncertainty. While $\boldsymbol{\mu}_k(\mathbf{x})$ and $\boldsymbol{\Sigma}_k(\mathbf{x})$ are available at retrieval time, computing $\boldsymbol{\mu}_k(\mathbf{x}')$ and $\boldsymbol{\Sigma}_k(\mathbf{x}')$ requires evaluating the embedding model on every document. However, in practical retrieval systems, document embeddings are precomputed once and stored in the index, whereas the query $\mathbf{x}$ arrives only at inference time (Zhou et al., 2022; Zhao et al., 2024). Since the surrogate loss in Eq. 2 contains terms that depend on $\mathbf{x}'$, which cannot be recomputed per query, we discard all document-dependent terms and retain only the query-dependent components. This yields a retrieval-feasible approximation of Eq. 2, leading to the following entropy-regularized convex optimization:

$$\min_{\boldsymbol{\pi}} \sum_{k=1}^{K} \boldsymbol{\pi}_k(\mathbf{x}) \left( \operatorname{tr}(\boldsymbol{\Sigma}_k(\mathbf{x})) + \|\boldsymbol{\mu}_k(\mathbf{x})\|^2 \right) - T \operatorname{H}(\boldsymbol{\pi}(\mathbf{x})) \quad \text{s.t.} \sum_{k=1}^{K} \boldsymbol{\pi}_k(\mathbf{x}) = 1, \ \boldsymbol{\pi}_k(\mathbf{x}) \geq 0, \quad (3)$$

where $T > 0$ controls sensitivity to uncertainty and $\operatorname{H}(\boldsymbol{\pi}(\mathbf{x}))$ denotes the Shannon entropy of the coefficient vector $\boldsymbol{\pi}(\mathbf{x}) := (\boldsymbol{\pi}_1(\mathbf{x}), \ldots, \boldsymbol{\pi}_K(\mathbf{x}))$, encouraging smooth allocations across models and preventing concentrated solutions. From a Bayesian perspective, this entropy regularization is equivalent (up to a constant) to a KL divergence between $\boldsymbol{\pi}(\mathbf{x})$ and a uniform prior over models, encoding a preference against overconfident model selection when uncertainty evidence is limited. Such entropy-based regularization has been widely adopted to stabilize soft reweighting and gating mechanisms under heterogeneous or noisy conditions (Sagawa et al., 2019; Nguyen et al., 2024).

The entropy-regularized objective admits a closed-form solution with temperature-controlled coefficients:

$$\boldsymbol{\pi}_k(\mathbf{x}; T) \approx \frac{\exp\left(-\operatorname{tr}(\boldsymbol{\Sigma}_k(\mathbf{x}))/T\right)}{\sum_{j=1}^{K} \exp\left(-\operatorname{tr}(\boldsymbol{\Sigma}_j(\mathbf{x}))/T\right)}. \tag{4}$$

This approximation follows from $\ell_2$ normalization, under which $\|\boldsymbol{\mu}_k(\mathbf{x})\|^2$ is approximately constant across models. With this formulation, UEC aggregates multiple embedding models in a *query-adaptive* and *data-dependent* manner. Rather than using fixed or global weights, each encoder's contribution is modulated by its estimated epistemic uncertainty for the query $\mathbf{x}$, assigning lower influence to unreliable models and emphasizing stable representations. Consequently, UEC performs a principled convolution over probabilistic embeddings that adapts to query-wise heterogeneity and distributional shift. Full derivation and discussion are provided in Appendix A.2.

### 3.3 UNCERTAINTY-AWARE SIMILARITY ESTIMATION

Measuring distances between embeddings is fundamental to many downstream applications. While distributional distances like KL divergence (Kullback & Leibler, 1951) or Wasserstein distance (Chhachhi & Teng, 2023; Gelbrich, 1990) are theoretically well-founded, their computational cost is often prohibitive. To address this, we propose a lightweight and principled similarity estimator.

Let the ensembled embeddings for $\mathbf{x}$ and $\mathbf{x}'$ be $\mathbf{q} = \mathbf{z}(\mathbf{x}) \sim \mathcal{N}(\boldsymbol{\mu}_{\mathbf{q}}, \boldsymbol{\Sigma}_{\mathbf{q}})$ and $\mathbf{c} = \mathbf{z}(\mathbf{x}') \sim \mathcal{N}(\boldsymbol{\mu}_{\mathbf{c}}, \boldsymbol{\Sigma}_{\mathbf{c}})$, respectively. Our similarity measure $s$ follows the properties of the base embedding models, which are typically trained with contrastive objectives on $\ell_2$-normalized outputs. This training results in mean vectors $\boldsymbol{\mu}_{\mathbf{q}}, \boldsymbol{\mu}_{\mathbf{c}}$ that are near unit-norm, making the small-variance assumption plausible. We explicitly normalize the mean vectors, allowing us to approximate cosine similarity by the dot product, $s \approx \mathbf{q}^\top \mathbf{c}$. This preserves the Gaussian form, and the small variance ensures the approximation remains accurate. We then model the resulting dot product $s$ as a Gaussian via moment matching (Randone et al., 2024; Mallik & Sagias, 2011):

$$s \sim \mathcal{N}(\mu_s, \sigma_s^2), \quad \mu_s = \boldsymbol{\mu}_{\mathbf{q}}^\top \boldsymbol{\mu}_{\mathbf{c}}, \quad \sigma_s^2 = \boldsymbol{\mu}_{\mathbf{q}}^\top \boldsymbol{\Sigma}_{\mathbf{c}} \boldsymbol{\mu}_{\mathbf{q}} + \boldsymbol{\mu}_{\mathbf{c}}^\top \boldsymbol{\Sigma}_{\mathbf{q}} \boldsymbol{\mu}_{\mathbf{c}} + \operatorname{tr}(\boldsymbol{\Sigma}_{\mathbf{q}} \boldsymbol{\Sigma}_{\mathbf{c}}). \tag{5}$$

To incorporate uncertainty, we adopt a probit approximation (Eschenhagen et al., 2021; Gibbs, 1998):

$$\hat{s} \approx \frac{\mu_s}{\sqrt{1 + \frac{\pi}{8}\sigma_s^2}}. \tag{6}$$

The $\hat{s}$ incorporates predictive uncertainty in a lightweight manner without additional sampling. Beyond its computational efficiency, the estimator $\hat{s}$ is theoretically grounded, as it approximates the squared 2-Wasserstein distance with provably bounded error, ensuring consistent ranking behavior. The following result holds under a mild small-variance scaling assumption, formalized in Appendix A.4.

**Theorem 1** (Bounded approximation to the squared 2-Wasserstein distance). *Let* $\mathbf{q} \sim \mathcal{N}(\boldsymbol{\mu_q}, \boldsymbol{\Sigma_q})$ *and* $\mathbf{c} \sim \mathcal{N}(\boldsymbol{\mu_c}, \boldsymbol{\Sigma_c})$ *be two Gaussian embeddings with* $\ell_2$-*normalized mean vectors and diagonal covariances* $\boldsymbol{\Sigma_q} = \mathrm{diag}(\sigma^2_{\mathbf{q},1}, \ldots, \sigma^2_{\mathbf{q},d})$, $\boldsymbol{\Sigma_c} = \mathrm{diag}(\sigma^2_{\mathbf{c},1}, \ldots, \sigma^2_{\mathbf{c},d})$. *Assume the small-variance regime, where* $\varepsilon := \max_i \{\sigma^2_{\mathbf{q},i}, \sigma^2_{\mathbf{c},i}\} < 1$. *Then, our similarity estimator* $\hat{s}$ *approximates the squared 2-Wasserstein distance* $W^2_2$ *as:*

$$\hat{s} = 1 - \tfrac{1}{2}W^2_2 + O(\varepsilon^2).$$

*Hence, ranking by* $\hat{s}$ *induces the same ranking as minimizing* $W^2_2$, *up to* $O(\varepsilon^2)$ *error.*

Our similarity estimator thus serves as a theoretically grounded surrogate for established distributional distances, achieving nearly identical ranking performance while being significantly more efficient. We provide a detailed derivation of the estimator in Appendix A.3, the proof for Theorem 1 in Appendix A.4, and empirical comparisons with alternative distances in Appendix B.2.

## 4 EXPERIMENTS

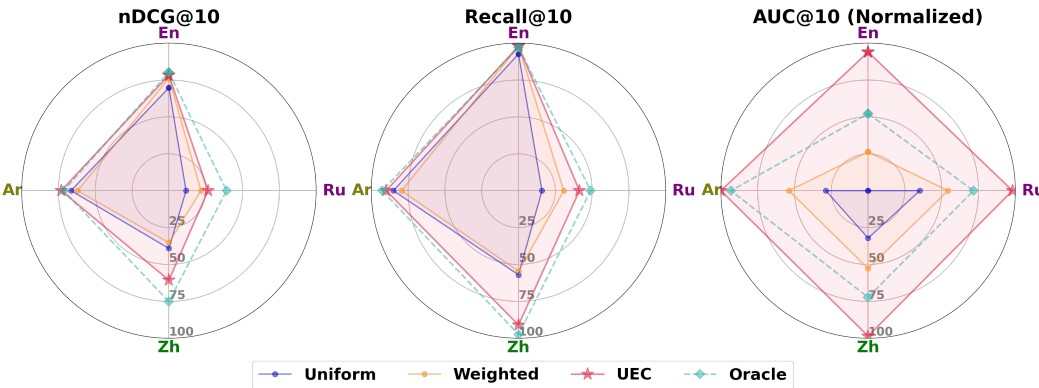

Figure 3: Performance on MIRACL Subset across ensemble methods. The oracle represents the upper bound by selecting the best language-specific model per language. UEC achieves performance comparable to the oracle and even surpasses in some cases, with particularly strong gains in AUC@10.

### 4.1 MIRACL SUBSET

**Setting** We first build a toy example with language-specialized models: E5 (Wang et al., 2022) (English/Russian), BGE-zh (Xiao et al., 2023) (Chinese), and MATRYOSHKA (Kusupati et al., 2022; Henderson et al., 2017) (Arabic). From the MIRACL Hard Negative dataset (Zhang et al., 2023), we sample 100 query-passage pairs each in English, Russian, Chinese, and Arabic. Baselines include uniform and weighted ensembles, along with an oracle upper bound that selects the best model per language. We report nDCG@10 and Recall@10 for retrieval and AUC@10 for uncertainty (Enevoldsen et al., 2025). Full experimental details are in Appendix B.3.

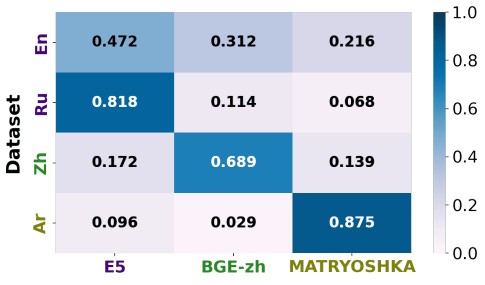

Figure 4: Heatmap of model-wise coefficients assigned by UEC per language. Each row corresponds to a language-specific input, and each column to an ensemble coefficient. UEC computes ensemble coefficients that are adaptively modulated by the uncertainty of each embedding.

**Results** As shown in Figure 3, UEC consistently outperforms uniform and weighted ensembles. For retrieval metrics (nDCG@10, Recall@10), it achieves performance comparable to the oracle. UEC explicitly models embedding uncertainty and adaptively down-weights unreliable representations, leading to similarity scores that are better aligned with retrieval correctness. This enables UEC to surpass the oracle on the uncertainty metric AUC@10, highlighting that uncertainty-aware ensembling not only preserves strong retrieval performance but also delivers superior confidence calibration.

To qualitatively assess how UEC adapts ensemble weights via uncertainty, we visualize the averaged model-wise coefficients per language in Figure 4. The $4 \times 3$ heatmap shows languages (rows: English, Russian, Chinese, Arabic) against ensembled models (columns). UEC assigns distinct patterns by language—for example, Arabic inputs emphasize Arabic embedding, while Chinese inputs favor Chinese embedding. It demonstrates that UEC effectively leverages embedding uncertainty to assign coefficients.

## 4.2 MMTEB

We further evaluate on a subset of MMTEB (Enevoldsen et al., 2025), focusing on three representative tasks: retrieval, classification, and Semantic Textual Similarity (STS). For each task, we carefully select datasets to cover a diverse range of domains and topics. Additional task results, which show performance consistent with the main results, are provided in the Appendix B.4.5. We use three SBERT-style base models—BGE (Xiao et al., 2023), E5 (Wang et al., 2022), and GTE (Li et al., 2023)—and a competitive multilingual baseline (GTE-MB (Zhang et al., 2024)). As a probabilistic embedding baseline, we adopt GroVE (Venkataramanan et al., 2025), applying it on top of the strongest individual model GTE-MB for fair comparison. To the best of our knowledge, this work presents the first systematic evaluation of model-merging (uniform/weighted, task arithmetic (Ilharco et al., 2022)) and ensemble (uniform/weighted) techniques for this benchmark. See Appendix B.4 for more experimental details.

### 4.2.1 RETRIEVAL

**Setting** We evaluate retrieval performance on five datasets: SCIDOCS (Cohan et al., 2020), Legal-BenchCorporateLobbying (Guha et al., 2023), BelebeleRetrieval (Lovenia et al., 2024), WikipediaRetrievalMultilingual, and StackOverflowQA (Li et al., 2024). We use nDCG@10 and Recall@100 as retrieval metrics. In addition to standard retrieval metrics, we use AUC@10 (Enevoldsen et al., 2025) as a key quantitative measure to assess how well the model's confidence scores are calibrated with its actual performance, directly evaluating the quality of its uncertainty estimation. We additionally compare Borda count (Emerson, 2013) as a representative rank aggregation technique, a natural strategy for retrieval tasks where each embedding model yields its own ranks.

Table 1: Comparison of retrieval performance. The best and second-best results per row are **bolded** and underlined. UEC achieves the highest average performance across all metrics, *with particularly strong gains on the AUC@10 uncertainty metric, highlighting the efficiency for uncertainty calibration.* (Borda=Borda count, Uni.=Uniform, Wt.=Weighted, TA=Task arithmetic).

| Dataset / Metric | Individual Models | | | | | Borda | Model Merging | | | Ensemble | | |
|---|---|---|---|---|---|---|---|---|---|---|---|---|
| | BGE | E5 | GTE | GTE-MB | GroVE | | Uni. | Wt. | TA | Uni. | Wt. | UEC |
| **SCIDOCS** | | | | | | | | | | | | |
| nDCG@10 | 21.47 | 18.30 | 23.14 | 19.46 | 21.88 | 22.18 | 21.73 | 22.43 | 22.66 | 22.67 | 22.11 | **24.01** |
| Recall@100 | 51.04 | 41.71 | 53.27 | 45.03 | 50.16 | 51.97 | 49.00 | 51.31 | 52.87 | 52.98 | 50.53 | **54.36** |
| AUC@10 | 27.25 | 34.42 | 31.63 | 32.52 | 34.03 | 5.84 | 29.42 | 30.99 | 30.72 | 28.52 | 32.08 | **35.21** |
| **Lobbying** | | | | | | | | | | | | |
| nDCG@10 | 90.42 | 91.54 | 91.81 | 90.40 | 90.91 | 91.23 | 92.35 | 92.32 | 90.91 | 91.79 | 92.15 | **93.56** |
| Recall@100 | 99.70 | **100.00** | 99.70 | 99.70 | 99.70 | 99.70 | **100.00** | 99.70 | 99.41 | 100.0 | 100.0 | 100.0 |
| AUC@10 | 55.56 | 58.24 | 55.02 | 54.62 | 57.81 | 32.83 | 57.49 | 53.17 | 53.18 | 52.07 | 55.79 | **60.04** |
| **Belebele** | | | | | | | | | | | | |
| nDCG@10 | 93.64 | 94.91 | 93.76 | 94.09 | 94.14 | 96.26 | 94.60 | 94.86 | 93.87 | 95.25 | 93.64 | **95.82** |
| Recall@100 | 99.44 | 99.55 | **99.77** | 99.66 | 99.44 | 99.44 | 99.66 | 99.66 | 99.44 | 99.66 | 99.55 | **99.77** |
| AUC@10 | 83.83 | 84.78 | 86.18 | 82.53 | 86.57 | 69.17 | 86.39 | 85.11 | 86.03 | 82.99 | 83.34 | **88.72** |
| **Wikipedia** | | | | | | | | | | | | |
| nDCG@10 | 92.17 | 93.32 | 92.71 | 91.55 | 92.71 | 93.12 | 91.88 | 93.08 | 92.99 | 93.32 | 93.84 | **94.24** |
| Recall@100 | 99.80 | **99.93** | **99.93** | 99.80 | 99.86 | 99.55 | 99.86 | 99.86 | 99.80 | 99.86 | 99.86 | **99.93** |
| AUC@10 | 59.84 | 61.06 | 61.57 | 62.08 | 62.36 | 49.26 | 58.61 | 60.60 | 62.62 | 62.41 | 62.01 | **64.25** |
| **StackOverflow** | | | | | | | | | | | | |
| nDCG@10 | 79.93 | 87.14 | 80.54 | **91.17** | 87.76 | 85.36 | 78.93 | 81.31 | 80.55 | 85.76 | 88.71 | 90.26 |
| Recall@100 | 96.59 | 97.54 | 97.64 | 98.79 | 97.54 | 98.09 | 96.28 | 97.44 | 96.54 | 98.09 | 98.09 | **99.40** |
| AUC@10 | 69.59 | 84.00 | 68.28 | 88.61 | 85.06 | 17.14 | 69.69 | 73.89 | 71.27 | 75.51 | 82.43 | 89.83 |
| **Avg. nDCG@10 ↑** | 75.52 | 77.04 | 76.39 | 77.33 | 77.48 | 77.11 | 75.90 | 75.69 | 76.19 | 77.76 | 78.49 | **79.58** |
| **Avg. Recall@100 ↑** | 89.31 | 87.74 | 90.06 | 88.60 | 89.34 | 89.75 | 88.96 | 89.59 | 89.61 | 90.12 | 89.61 | **90.69** |
| **Avg. AUC@10 ↑** | 54.65 | 64.70 | 60.74 | 64.07 | 65.16 | 34.85 | 60.52 | 60.75 | 60.76 | 60.30 | 63.13 | **67.61** |

**Results** As shown in Table 1, UEC attains the highest average performance across both retrieval and uncertainty estimation, surpassing even the strong GTE-MB baseline by ensembling three weaker

models. These results demonstrate that our uncertainty-aware ensemble framework effectively integrates heterogeneous embeddings, achieving robust and accurate retrieval across diverse datasets.

### 4.2.2 CLASSIFICATION

**Setting** We evaluate on five classification datasets covering finance, law, poetry, multilingual intent, and social media: FinancialPhrasebankClassification (Malo et al., 2014), SwissJudgementClassification (Niklaus et al., 2021), PoemSentimentClassification (Sheng & Uthus, 2020), MassiveIntentClassification (FitzGerald et al., 2022), and TweetTopicSingleClassification (Antypas et al., 2022). Accuracy, F1 score, and AUROC are reported as evaluation metrics, where AUROC (Kuhn et al., 2023) serves as a key quantitative measure to assess how well the model estimates the degree of uncertainty in distinguishing between classes.

Table 2: Comparison of classification performance. The best and second-best results per row are **bolded** and underlined. UEC achieves the highest average scores, *demonstrating robust performance across diverse classification tasks* (Uni.=Uniform, Wt.=Weighted, TA=Task arithmetic).

| Dataset / Metric | Individual Models | | | | | Model Merging | | | Ensemble | | |
|---|---|---|---|---|---|---|---|---|---|---|---|
| | BGE | E5 | GTE | GTE-MB | GroVE | Unif. | Wt. | TA | Unif. | Wt. | UEC |
| **Financial** | | | | | | | | | | | |
| Accuracy | 80.47 | 82.92 | 81.89 | 75.07 | 76.73 | 77.64 | 79.39 | 79.53 | 82.15 | 82.64 | **83.02** |
| F1 | 78.38 | 79.90 | 79.10 | 73.50 | 75.65 | 75.53 | 77.85 | 77.93 | 79.76 | 80.18 | **80.30** |
| AUROC | 92.46 | 93.04 | 92.02 | 90.62 | 91.13 | 89.63 | 91.92 | 92.38 | 92.86 | 93.16 | **94.07** |
| **Judgement** | | | | | | | | | | | |
| Accuracy | 57.16 | **58.58** | 56.37 | 55.45 | 55.44 | 56.82 | 55.90 | 55.44 | 57.09 | 57.71 | 58.36 |
| F1 | 46.63 | 47.25 | 46.34 | 46.41 | 45.82 | 46.55 | 45.87 | 45.73 | 46.61 | 47.02 | **48.52** |
| AUROC | 50.48 | 50.22 | 50.65 | 50.67 | 50.48 | 50.13 | 50.24 | 50.37 | 50.43 | 50.51 | **51.02** |
| **Poem** | | | | | | | | | | | |
| Accuracy | 51.92 | 53.07 | 47.88 | **56.92** | 53.13 | 50.28 | 49.90 | 51.82 | 52.30 | 52.01 | 56.81 |
| F1 | 40.76 | 41.43 | 37.53 | **45.53** | 41.82 | 39.27 | 39.36 | 40.93 | 40.96 | 41.00 | 44.46 |
| AUROC | 56.70 | 57.38 | 53.41 | **62.21** | 57.31 | 55.13 | 54.36 | 55.76 | 56.71 | 56.27 | 61.39 |
| **Intent** | | | | | | | | | | | |
| Accuracy | 73.46 | 68.05 | 65.09 | 75.19 | 74.29 | 72.95 | 74.07 | 75.33 | 68.78 | 68.31 | **77.08** |
| F1 | 70.01 | 63.00 | 60.62 | 72.79 | 72.80 | 71.01 | 72.10 | 73.65 | 66.43 | 65.46 | **74.78** |
| AUROC | 66.74 | 64.54 | 63.09 | 67.23 | 66.98 | 64.07 | 67.15 | 67.02 | 65.26 | 65.11 | **67.91** |
| **Topic** | | | | | | | | | | | |
| Accuracy | 71.75 | 69.96 | 71.97 | 70.99 | 71.24 | 71.18 | 70.59 | 70.53 | 73.60 | 73.47 | **74.20** |
| F1 | 55.04 | 53.09 | 54.65 | 54.85 | 54.93 | 55.48 | 54.78 | 54.82 | 56.32 | 56.13 | **57.14** |
| AUROC | 87.02 | 87.83 | 88.54 | 89.76 | 87.95 | 88.64 | 88.12 | 88.33 | 90.07 | 90.12 | **90.73** |
| **Avg. Accuracy ↑** | 66.95 | 66.52 | 64.64 | 66.73 | 66.16 | 65.77 | 65.97 | 66.53 | 66.79 | 66.83 | **68.89** |
| **Avg. F1 ↑** | 58.16 | 56.93 | 55.65 | 58.62 | 58.20 | 57.57 | 57.99 | 58.61 | 58.02 | 57.96 | **61.04** |
| **Avg. AUROC ↑** | 70.68 | 70.60 | 69.54 | 72.09 | 70.77 | 69.52 | 70.35 | 70.77 | 71.06 | 71.03 | **73.02** |

**Results** As shown in Table 2, UEC attains the highest average accuracy, F1 score, and AUROC across five classification datasets. Notably, UEC also yields the best average AUROC, indicating that the model effectively captures the degree of uncertainty. These results demonstrate the effectiveness of UEC in classification scenarios where embedding-level uncertainty plays a crucial role.

### 4.2.3 SEMANTIC TEXTUAL SIMILARITY

**Setting** We evaluate STS performance on 10 datasets, including STSBenchmark, FinParaSTS, SICK-R Marelli et al. (2014), STS22.v2 Chen et al. (2022), SemRel24STS (Ousidhoum et al., 2024), and STS12–STS17 (Agirre et al., 2012; 2013; Bandhakavi et al., 2014; Biçici, 2015; Cer et al., 2017) . Spearman correlation is used to measure alignment with human-annotated similarity scores.

**Results** As shown in Table 3, UEC achieves the highest average performance across the STS tasks. It ranks first on 8 out of the 10 datasets. These results demonstrate that our uncertainty-aware aggregation not only enhances robustness across diverse semantic similarity tasks but also outperforms existing deterministic and ensemble baselines.

Table 3: Comparison of STS performance. The best and second-best results per column are **bolded** and underlined. UEC achieves the highest average performance, *ranking first on 8 out of the 10 datasets evaluated* (Uni.=Uniform, Wt.=Weighted, TA=Task arithmetic).

| Models | | STSB | FinPara | SICK-R | SemRel24 | STS12 | STS13 | STS14 | STS15 | STS17 | STS22 | Avg. |
|---|---|---|---|---|---|---|---|---|---|---|---|---|
| **Individual** | **BGE** | 86.41 | 9.43 | 80.30 | 79.52 | 78.02 | 84.18 | 82.27 | 87.95 | 86.41 | 66.53 | 74.10 |
| | **E5** | 85.47 | **15.99** | 78.39 | 81.37 | 73.49 | 82.99 | 80.44 | 88.18 | 88.89 | 66.51 | 74.17 |
| | **GTE** | 85.73 | 12.95 | 78.85 | 77.79 | 75.70 | 85.72 | 81.51 | 88.80 | 87.88 | 67.65 | 74.26 |
| | **GTE-MB** | 85.41 | 15.18 | 75.71 | 75.02 | 74.82 | 86.63 | 78.75 | 85.39 | 88.31 | **72.28** | 73.75 |
| | **GroVE** | 85.93 | 11.53 | 80.02 | 78.99 | 76.26 | 83.33 | 79.03 | 85.27 | 88.89 | 66.98 | 73.62 |
| **Model Merging** | **Uni.** | 86.78 | 11.75 | 79.85 | 79.49 | 75.43 | 86.79 | 82.07 | 89.12 | 88.71 | 68.42 | 74.84 |
| | **Wt.** | 87.22 | 11.64 | 80.16 | 78.80 | 77.36 | 87.08 | 83.39 | 89.49 | 88.43 | 67.51 | 75.11 |
| | **TA** | 86.83 | 10.22 | 80.25 | 78.61 | 78.40 | 85.85 | 83.18 | 88.67 | 87.41 | 66.31 | 74.57 |
| **Ensemble** | **Uni.** | 87.44 | 12.03 | 80.73 | 80.24 | 78.00 | 85.90 | 83.62 | 89.07 | 88.21 | 68.15 | 75.34 |
| | **Wt.** | 87.44 | 14.13 | 80.41 | 81.08 | 77.07 | 85.76 | 83.45 | 89.17 | 89.18 | 68.68 | 75.64 |
| | **UEC** | **87.55** | 15.53 | **81.37** | **81.89** | **78.85** | **87.62** | **83.75** | **89.50** | **89.24** | 69.55 | **76.49** |

## 4.3 UNCERTAINTY ESTIMATION DIAGNOSIS

To assess whether the post-hoc probabilistic embeddings and the resulting similarity variances provide meaningful uncertainty estimates, we evaluate two aspects: (1) calibration of the Laplace-based probabilistic embedding model used in UEC, and (2) whether the similarity variance $\sigma_s^2$ faithfully reflects predictive uncertainty. For all experiments, we fit LA following the same configuration with Section 4.2, and evaluate uncertainty using a binary classification proxy task on MS MARCO.

### 4.3.1 CALIBRATION OF PROBABILISTIC EMBEDDINGS

We first measure how well the Laplace-based probabilistic embeddings capture epistemic uncertainty at the *model* level. Across all models, LA consistently improves calibration, and UEC yields the best-calibrated predictions. These findings demonstrate that the post-hoc probabilistic embedding model provides a stable and meaningful estimate of parameter uncertainty. Further analyses—covariance structure, number of layers included in LA, and data sparsity—are provided in Appendix C, where we observe consistent robustness across all factors.

Table 4: Calibration of probabilistic embeddings derived via LA. Probabilistic embeddings outperform the deterministic ones consistently.

| ECE ↓ | Deter. | Prob. |
|---|---|---|
| **E5** | 0.063 | **0.036** |
| **BGE** | 0.067 | **0.042** |
| **GTE** | 0.077 | **0.043** |
| **UEC** | 0.075 | **0.032** |

### 4.3.2 VARIANCE CALIBRATION OF THE SIMILARITY SCORE

We next assess whether UEC's similarity variance $\sigma_s^2$ meaningfully reflects predictive uncertainty.

**Var-ECE** Since standard ECE does not apply to continuous similarity values, we introduce a variance–error metric (Var-ECE) that measures the alignment between predicted variance and empirical squared error. For each example, higher predicted variance should correspond to a larger squared residual. We group examples into variance-quantile bins and compute the absolute discrepancy between predicted variance and empirical squared error. The full derivation and extended diagnostics remain in Appendix C.2.

**Results** On the multilingual SEMREL24STS benchmark, UEC achieves the lowest Var-ECE (Table 5), showing that $\sigma_s^2$ provides a numerically faithful estimate of similarity uncertainty.

Table 5: Variance calibration on SEMREL24STS. UEC achieves the lowest Var-ECE, indicating that $\sigma_s^2$ faithful estimate of similarity uncertainty.

| Model | Var-ECE ↓ |
|---|---|
| **E5** | 0.035 |
| **BGE** | 0.051 |
| **GTE** | 0.038 |
| **UEC** | **0.028** |

## 4.4 EFFICIENCY AND COMPUTATIONAL CHARACTERISTICS

Table 6 compares the computational characteristics of UEC against common ensemble baselines. Three key capabilities are considered: automatic coefficient selection, data-wise coefficient estimation, and uncertainty-aware similarity estimation. Among all methods, only UEC supports all three features (✔), while others lack adaptability or require manual tuning (✗, -).

Despite its added functionality, UEC introduces negligible overhead. It maintains the same asymptotic time and memory complexity $\mathcal{O}(KD)$, where $D$ is the embedding dimension. UEC increases only

Table 6: Comparison of computational characteristics. UEC supports automatic coefficient selection and uncertainty-aware similarity while preserving efficiency. "Rel." and "Comp." denote relative time and complexity, respectively. Symbols: ✓ (Yes), ✗ (No), – (Not applicable).

| Method | Auto Coeff. Selection | Data-wise Coeff. | Uncertainty-aware Similarity | Memory Complex. | Time Rel. | Time Complex. |
|---|---|---|---|---|---|---|
| Uniform Ensemble | - | ✗ | ✗ | $\mathcal{O}(KD)$ | 1.000 | $\mathcal{O}(KD)$ |
| Weighted Ensemble | ✗ | ✗ | ✗ | $\mathcal{O}(KD)$ | 1.000 | $\mathcal{O}(KD)$ |
| Task Arithmetic | ✗ | ✗ | ✗ | $\mathcal{O}(KD)$ | 1.000 | $\mathcal{O}(KD)$ |
| **UEC (Ours)** | ✓ | ✓ | ✓ | $\mathcal{O}(KD)$ | 1.006 | $\mathcal{O}(KD)$ |

0.6% in actual computation time for similarity estimation, demonstrating strong adaptivity without sacrificing efficiency and indicating its suitability for real-time deployment.

## 4.5 ABLATION STUDY

We perform an ablation study to evaluate each module's contribution. Following the protocol in Section 4.1, we replace (i) the uncertainty-aware similarity with an uncalibrated one (- Unc Sim), (ii) the uncertainty-driven convolution coefficient with a uniform one (- Unc Conv), and (iii) both (- Unc Sim, Conv). Table 7 shows that removing any component noticeably drops performance, highlighting their individual and complementary contributions.

Table 7: Ablation for UEC's components: uncertainty-aware similarity (Unc Sim) and adaptive convolution (Unc Conv). Removing either degrades performance, with the joint removal yielding the worst results.

| | nDCG@10 | Recall@10 | AUC@10 |
|---|---|---|---|
| **UEC** | **59.65**% | **80.07**% | **91.04**% |
| - Unc Sim | 58.72% | 78.13% | 82.48% |
| | (↓ 0.93%) | (↓1.94%) | (↓8.56%) |
| - Unc Conv | 48.45% | 66.69% | 10.30% |
| | (↓ 11.20%) | (↓ 13.38%) | (↓ 80.74%) |
| - Unc Sim, Conv | 46.78% | 62.66% | 4.01% |
| | (↓ 22.87%) | (↓ 17.41%) | (↓ 87.03%) |

## 4.6 CHALLENGING CASE

Figure 5 shows a qualitative example from the setting in Section 4.1. Although the query is in English and E5 is expected to perform well, all individual models fail to retrieve the correct passage, which appears at ranks 12, 28, and 37. In contrast, UEC retrieves it at rank 6, thereby correctly retrieving the answer under both nDCG@10 and Recall@10 metrics, which are evaluated over the top 10 candidates. This example highlights UEC's ability to handle model limitations by using uncertainty-aware ensembling and to generalize beyond language-specific encoders.

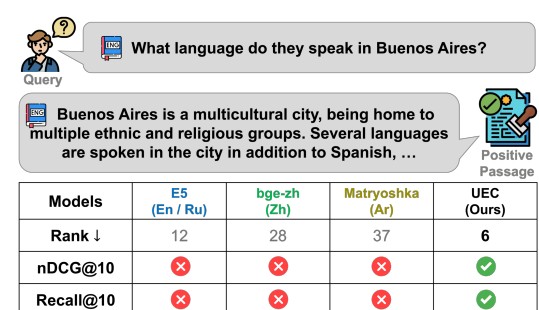

Figure 5: Challenging case where only UEC correctly retrieves the positive passage, demonstrating its ability to leverage uncertainty.

## 5 CONCLUSION AND DISCUSSION

We propose the Uncertainty-driven Embedding Convolution (UEC) framework, which converts pre-trained deterministic embeddings into probabilistic representations and adaptively ensembles them via Bayesian coefficient estimation and an uncertainty-aware similarity function. To the best of our knowledge, this introduces the first fully post-hoc, uncertainty-calibrated ensemble approach for embedding models, yielding consistent and reliable gains across diverse tasks. While UEC currently models only epistemic uncertainty and relies on diagonal LA, it does not yet capture aleatoric or full predictive uncertainty. Extending UEC to incorporate these additional uncertainty components and to explore richer posterior structures with efficient computation represents a promising direction for enhanced robustness. Furthermore, UEC assumes that all embeddings share the same dimensionality and may inherit biases from the underlying models. Relaxing the dimensionality constraint and developing uncertainty mechanisms that explicitly account for bias and fairness offer important avenues for responsible real-world deployment. Finally, applying UEC to multimodal settings, in which heterogeneous sources of uncertainty interact across vision, speech, and text, presents an exciting opportunity to broaden the framework's impact.

ACKNOWLEDGEMENT

This work was supported by the National Research Foundation of Korea(NRF) grant funded by the Korea government(MSIT)(RS-2024-00457216).

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

# A  DERIVATION AND PROOF

## A.1  DETAILED DERIVATION OF THE LAPLACE APPROXIMATION

Let $\mathcal{D} = \{(\mathbf{x}_i, y_i)\}_{i=1}^N$ denote the training set and $\mathbf{W}^{(L)}$ the parameters of the final linear layer of the embedding model. The posterior over $\mathbf{W}^{(L)}$ is given by Bayes' rule:

$$p(\mathbf{W}^{(L)} \mid \mathcal{D}) = \frac{p(\mathcal{D} \mid \mathbf{W}^{(L)})p(\mathbf{W}^{(L)})}{p(\mathcal{D})}. \tag{7}$$

The exact posterior is typically intractable. The Laplace Approximation (LA) expands the negative log-posterior around its maximizer (the MAP estimate)

$$\widehat{\mathbf{W}}^{(L)} = \arg \max_{\mathbf{W}^{(L)}} p(\mathbf{W}^{(L)} \mid \mathcal{D}).$$

**Quadratic Expansion of the Log-Posterior**  Define the log-posterior

$$\ell(\mathbf{W}^{(L)}) = \log p(\mathcal{D} \mid \mathbf{W}^{(L)}) + \log p(\mathbf{W}^{(L)}).$$

A second-order Taylor expansion around $\widehat{\mathbf{W}}^{(L)}$ gives

$$\ell(\mathbf{W}^{(L)}) \approx \ell(\widehat{\mathbf{W}}^{(L)}) + \frac{1}{2}(\mathbf{W}^{(L)} - \widehat{\mathbf{W}}^{(L)})^\top \nabla^2 \ell(\widehat{\mathbf{W}}^{(L)})(\mathbf{W}^{(L)} - \widehat{\mathbf{W}}^{(L)}). \tag{8}$$

Since $\widehat{\mathbf{W}}^{(L)}$ is a maximizer,

$$\nabla \ell(\widehat{\mathbf{W}}^{(L)}) = \mathbf{0}.$$

Define the negative Hessian (precision matrix)

$$\mathbf{H}_{\widehat{\mathbf{W}}^{(L)}} = -\nabla^2 \ell(\widehat{\mathbf{W}}^{(L)}) = \nabla^2 \big[ -\log p(\mathcal{D} \mid \mathbf{W}^{(L)}) - \log p(\mathbf{W}^{(L)}) \big]\Big|_{\mathbf{W}^{(L)} = \widehat{\mathbf{W}}^{(L)}}.$$

Thus the posterior is approximated by

$$p(\mathbf{W}^{(L)} \mid \mathcal{D}) \approx \mathcal{N}\left(\widehat{\mathbf{W}}^{(L)}, \mathbf{H}_{\widehat{\mathbf{W}}^{(L)}}^{-1}\right).$$

**Posterior over Embeddings**  Following Section 3.1, let $f(\mathbf{x})$ be an embedding model with last-layer parameters $\mathbf{W}^{(L)}$ and penultimate representation $\mathbf{h}^{(L-1)}(\mathbf{x})$. The embedding is given by the final linear projection

$$\mathbf{z}(\mathbf{x}) = \mathbf{W}^{(L)}\mathbf{h}^{(L-1)}(\mathbf{x}).$$

Under the Gaussian posterior

$$\mathbf{W}^{(L)} \sim \mathcal{N}\left(\widehat{\mathbf{W}}^{(L)}, \mathbf{H}_{\widehat{\mathbf{W}}^{(L)}}^{-1}\right),$$

the embedding $\mathbf{z}(\mathbf{x})$ becomes a Gaussian random variable since it is a linear transformation of $\mathbf{W}^{(L)}$.

Using standard results on affine transformations of Gaussian variables, the predictive distribution of the embedding is given by

$$\mathbf{z}(\mathbf{x}) \sim \mathcal{N}\left(\widehat{\mathbf{W}}^{(L)}\mathbf{h}^{(L-1)}(\mathbf{x}),\ \mathbf{h}^{(L-1)}(\mathbf{x})^\top \mathbf{H}_{\widehat{\mathbf{W}}^{(L)}}^{-1}\mathbf{h}^{(L-1)}(\mathbf{x})\right). \tag{9}$$

Therefore, the deterministic embedding model is converted into a probabilistic one that outputs a Gaussian embedding for each input $\mathbf{x}$, where the mean corresponds to the original embedding and the covariance captures epistemic uncertainty induced by the last-layer parameter posterior.

**Diagonal Precision Approximation**  In practice, computing the full Hessian is computationally expensive. We therefore adopt a diagonal approximation of the precision matrix:

$$\mathbf{H}_{\widehat{\mathbf{W}}^{(L)}} \approx \mathrm{diag}\big(\mathbf{H}_{\widehat{\mathbf{W}}^{(L)}}\big).$$

This approximation preserves tractability while retaining model-specific epistemic uncertainty information, yielding a closed-form Gaussian embedding distribution used in the UEC framework.

A.2 DERIVATION OF UNCERTAINTY-DRIVEN COEFFICIENT FOR EMBEDDING CONVOLUTION

In this section, we provide a detailed derivation of the uncertainty-driven coefficients used in the embedding convolution described in Section 3.2. In contrast to a hard selection induced by a linear objective, we derive a temperature-controlled, closed-form solution by introducing entropy regularization, which yields stable and query-adaptive coefficients.

**Problem Setting**  Let there be $K$ independent embedding models. Each model $k \in \{1, \ldots, K\}$ produces a probabilistic embedding

$$\mathbf{z}_k(\mathbf{x}) \sim \mathcal{N}(\boldsymbol{\mu}_k(\mathbf{x}), \boldsymbol{\Sigma}_k(\mathbf{x}))$$

for an input $\mathbf{x}$. Similarly, for a positive example $\mathbf{x}'$, we define

$$\mathbf{z}_k(\mathbf{x}') \sim \mathcal{N}(\boldsymbol{\mu}_k(\mathbf{x}'), \boldsymbol{\Sigma}_k(\mathbf{x}')).$$

The ensembled embedding is defined as a convex combination:

$$\mathbf{z}(\mathbf{x}) = \sum_{k=1}^{K} \boldsymbol{\pi}_k(\mathbf{x}) \, \mathbf{z}_k(\mathbf{x}), \quad \sum_{k=1}^{K} \boldsymbol{\pi}_k(\mathbf{x}) = 1, \quad \boldsymbol{\pi}_k(\mathbf{x}) \geq 0.$$

**Surrogate Loss**  Following prior analyses of contrastive representation learning (Wang & Isola, 2020), we adopt the squared Euclidean distance as a surrogate objective. For $\ell_2$-normalized features, minimizing squared distance is equivalent to maximizing cosine similarity, which captures the alignment property of contrastive learning while avoiding explicit negative sampling.

For a positive pair $(\mathbf{x}, \mathbf{x}')$, we define the uncertainty-aware surrogate loss as

$$\mathcal{L}_{\mathrm{sur}}(\boldsymbol{\pi}; \mathbf{x}, \mathbf{x}') = \sum_{k=1}^{K} \boldsymbol{\pi}_k(\mathbf{x}) \, \mathbb{E}\big[\|\mathbf{z}_k(\mathbf{x}) - \mathbf{z}_k(\mathbf{x}')\|^2\big].$$

**Expected Distance Between Gaussian Embeddings**  For independent Gaussian variables $\mathbf{z}_k(\mathbf{x}) \sim \mathcal{N}(\boldsymbol{\mu}_k(\mathbf{x}), \boldsymbol{\Sigma}_k(\mathbf{x}))$ and $\mathbf{z}_k(\mathbf{x}') \sim \mathcal{N}(\boldsymbol{\mu}_k(\mathbf{x}'), \boldsymbol{\Sigma}_k(\mathbf{x}'))$, the expected squared distance admits the closed-form expression:

$$\mathbb{E}\big[\|\mathbf{z}_k(\mathbf{x}) - \mathbf{z}_k(\mathbf{x}')\|^2\big] = \|\boldsymbol{\mu}_k(\mathbf{x}) - \boldsymbol{\mu}_k(\mathbf{x}')\|^2 + \mathrm{tr}(\boldsymbol{\Sigma}_k(\mathbf{x})) + \mathrm{tr}(\boldsymbol{\Sigma}_k(\mathbf{x}')).$$

This decomposition separates a fidelity term, capturing semantic alignment, and an epistemic uncertainty term, reflecting model confidence.

**Retrieval-feasible Approximation**  In large-scale retrieval systems, document embeddings corresponding to $\mathbf{x}'$ are precomputed and fixed, while the query $\mathbf{x}$ arrives only at inference time. To respect this constraint, we discard all document-dependent terms and retain only the query-dependent components, yielding the following query-time objective:

$$\min_{\boldsymbol{\pi}} \sum_{k=1}^{K} \boldsymbol{\pi}_k(\mathbf{x}) \Big( \mathrm{tr}(\boldsymbol{\Sigma}_k(\mathbf{x})) + \|\boldsymbol{\mu}_k(\mathbf{x})\|^2 \Big) - T \, H(\boldsymbol{\pi}(\mathbf{x})), \quad \text{s.t.} \sum_k \boldsymbol{\pi}_k(\mathbf{x}) = 1, \, \boldsymbol{\pi}_k(\mathbf{x}) \geq 0, \quad (10)$$

where $H(\boldsymbol{\pi}(\mathbf{x})) := -\sum_k \boldsymbol{\pi}_k(\mathbf{x}) \log \boldsymbol{\pi}_k(\mathbf{x})$ denotes the Shannon entropy.

The entropy term prevents degenerate solutions that collapse onto a single model and can be interpreted as a KL divergence to a uniform prior over models, thereby promoting robust aggregation under noisy, query-specific uncertainty.

**Closed-form Solution**  The optimization problem in Eq. 10 is strictly convex. Introducing a Lagrange multiplier $\lambda$ for the simplex constraint, the Lagrangian is

$$\mathcal{L}(\boldsymbol{\pi}, \lambda) = \sum_k \boldsymbol{\pi}_k c_k + T \sum_k \boldsymbol{\pi}_k \log \boldsymbol{\pi}_k + \lambda \left( \sum_k \boldsymbol{\pi}_k - 1 \right),$$

where $c_k := \mathrm{tr}(\boldsymbol{\Sigma}_k(\mathbf{x})) + \|\boldsymbol{\mu}_k(\mathbf{x})\|^2$.

Setting $\partial \mathcal{L}/\partial \boldsymbol{\pi}_k = 0$ yields

$$\log \boldsymbol{\pi}_k = -\frac{1}{T}c_k - \frac{\lambda}{T} - 1,$$

which implies

$$\boldsymbol{\pi}_k \propto \exp\left(-\frac{c_k}{T}\right).$$

Normalizing over $k$ gives the closed-form solution

$$\boldsymbol{\pi}_k^\star(\mathbf{x}) = \frac{\exp(-c_k/T)}{\sum_{j=1}^K \exp(-c_j/T)}. \tag{11}$$

**Approximation under $\ell_2$ Normalization**   In practice, embeddings are $\ell_2$-normalized, implying $\|\boldsymbol{\mu}_k(\mathbf{x})\|^2 \approx 1$ across models. Treating this term as approximately constant, Eq. 11 reduces to

$$\boldsymbol{\pi}_k^\star(\mathbf{x}) \approx \frac{\exp(-\mathrm{tr}(\boldsymbol{\Sigma}_k(\mathbf{x}))/T)}{\sum_j \exp(-\mathrm{tr}(\boldsymbol{\Sigma}_j(\mathbf{x}))/T)},$$

which is equivalent to the power-law form in Eq. 4 of the main text.

**Interpretation**   The resulting coefficients assign higher weight to models with lower epistemic uncertainty for the given query, while entropy regularization ensures smooth, query-adaptive aggregation. This yields a principled uncertainty-aware embedding convolution that is robust to heterogeneous model reliability and distributional shift.

## A.3   Derivation of Uncertainty-aware Similarity Estimation

In this section, we provide a detailed derivation of the uncertainty-aware similarity estimation method introduced in Section 3.3. When embeddings are modeled as multivariate Gaussian distributions, the similarity between them becomes a random variable. We aim to derive the distribution of this similarity score and propose a calibrated approximation that accounts for predictive uncertainty.

**Problem Setup**   Let $\mathbf{q} = \mathbf{z}(\mathbf{x}) \sim \mathcal{N}(\boldsymbol{\mu_q}, \boldsymbol{\Sigma_q})$ and $\mathbf{c} = \mathbf{z}(\mathbf{x}') \sim \mathcal{N}(\boldsymbol{\mu_c}, \boldsymbol{\Sigma_c})$ denote two ensembled probabilistic embeddings generated by the proposed UEC method. Each embedding is represented as a Gaussian distribution with a mean vector $\boldsymbol{\mu} \in \mathbb{R}^d$ and a covariance matrix $\boldsymbol{\Sigma} \in \mathbb{R}^{d \times d}$.

We define the cosine similarity between $\mathbf{q}$ and $\mathbf{c}$ as:

$$s := \cos(\mathbf{q}, \mathbf{c}) = \frac{\mathbf{q}^\top \mathbf{c}}{\|\mathbf{q}\| \cdot \|\mathbf{c}\|},$$

which itself is a random variable, since both $\mathbf{q}$ and $\mathbf{c}$ are random. In our implementation, both $\mathbf{q}$ and $\mathbf{c}$ are $\ell_2$-normalized in expectation (i.e., $\|\boldsymbol{\mu_q}\| \approx \|\boldsymbol{\mu_c}\| \approx 1$), and their variances are relatively small. Therefore, we simplify cosine similarity to the inner product:

$$s \approx \mathbf{q}^\top \mathbf{c},$$

as is common in prior work (Mallik & Sagias, 2011; Randone et al., 2024).

**Distribution of the Inner Product via Moment Matching**   To characterize the distribution of $s = \mathbf{q}^\top \mathbf{c}$, we approximate it as a Gaussian random variable using moment-matching (Randone et al., 2024). Since $\mathbf{q}$ and $\mathbf{c}$ are independent Gaussian vectors, their inner product $s$ follows an approximately Gaussian distribution whose mean and variance can be computed in closed form.

**Mean** By linearity of expectation:

$$\mu_s := \mathbb{E}[\mathbf{q}^\top \mathbf{c}] = \mathbb{E}[\mathbf{q}]^\top \mathbb{E}[\mathbf{c}] = \boldsymbol{\mu_q}^\top \boldsymbol{\mu_c}. \tag{12}$$

**Variance** Using the law of total variance:

$$\mathrm{Var}(\mathbf{q}^\top \mathbf{c}) = \mathbb{E}_{\mathbf{q}}\left[\mathrm{Var}(\mathbf{q}^\top \mathbf{c} \mid \mathbf{q})\right] + \mathrm{Var}_{\mathbf{q}}\left(\mathbb{E}[\mathbf{q}^\top \mathbf{c} \mid \mathbf{q}]\right)$$
$$= \mathbb{E}_{\mathbf{q}}[\mathbf{q}^\top \boldsymbol{\Sigma_c} \mathbf{q}] + \mathrm{Var}_{\mathbf{q}}(\mathbf{q}^\top \boldsymbol{\mu_c}),$$

where we assume $\mathbf{q}$ and $\mathbf{c}$ are independent.

We evaluate each term as follows. For the first term, using the identity $\mathbb{E}[\mathbf{x}^\top \mathbf{A}\mathbf{x}] = \mathrm{tr}(\mathbf{A}\boldsymbol{\Sigma}) + \boldsymbol{\mu}^\top \mathbf{A}\boldsymbol{\mu}$ for $\mathbf{x} \sim \mathcal{N}(\boldsymbol{\mu}, \boldsymbol{\Sigma})$ (Shao, 2008):

$$\mathbb{E}_{\mathbf{q}}[\mathbf{q}^\top \boldsymbol{\Sigma}_{\mathbf{c}} \mathbf{q}] = \mathrm{tr}(\boldsymbol{\Sigma}_{\mathbf{c}} \boldsymbol{\Sigma}_{\mathbf{q}}) + \boldsymbol{\mu}_{\mathbf{q}}^\top \boldsymbol{\Sigma}_{\mathbf{c}} \boldsymbol{\mu}_{\mathbf{q}}.$$

For the second term, using the variance of a linear form in a Gaussian:

$$\mathrm{Var}_{\mathbf{q}}(\mathbf{q}^\top \boldsymbol{\mu}_{\mathbf{c}}) = \boldsymbol{\mu}_{\mathbf{c}}^\top \boldsymbol{\Sigma}_{\mathbf{q}} \boldsymbol{\mu}_{\mathbf{c}}.$$

Combining both, we obtain:

$$\sigma_s^2 := \mathrm{Var}(\mathbf{q}^\top \mathbf{c}) = \boldsymbol{\mu}_{\mathbf{q}}^\top \boldsymbol{\Sigma}_{\mathbf{c}} \boldsymbol{\mu}_{\mathbf{q}} + \boldsymbol{\mu}_{\mathbf{c}}^\top \boldsymbol{\Sigma}_{\mathbf{q}} \boldsymbol{\mu}_{\mathbf{c}} + \mathrm{tr}(\boldsymbol{\Sigma}_{\mathbf{q}} \boldsymbol{\Sigma}_{\mathbf{c}}).$$

Therefore, under moment-matching, we approximate the distribution of similarity as:

$$s = \mathbf{q}^\top \mathbf{c} \sim \mathcal{N}(\mu_s, \sigma_s^2).$$

**Probit-based Calibration of Similarity Scores**    The similarity distribution provides a way to quantify uncertainty, but we still need a scalar value to rank candidates. To account for this uncertainty in a principled yet computationally efficient way, we apply a probit-based calibration to the mean similarity score:

$$\hat{s} \approx \frac{\mu_s}{\sqrt{1 + \frac{\pi}{8}\sigma_s^2}}. \tag{13}$$

This formula arises from approximating the expectation of a sigmoid over a Gaussian random variable (Gibbs, 1998; Eschenhagen et al., 2021), i.e., $\mathbb{E}[\sigma(\mathbf{z})] \approx \sigma\left(\mu/\sqrt{1 + \frac{\pi}{8}\sigma^2}\right)$ when $z \sim \mathcal{N}(\mu, \sigma^2)$ and $\sigma(\cdot)$ is the logistic sigmoid function.

**Interpretation and Robustness**    The numerator $\mu_s$ captures alignment between the means, while the denominator serves as a soft penalty for high uncertainty. As $\sigma_s^2$ increases, the similarity score $\hat{s}$ is downscaled, reflecting reduced confidence in the similarity estimate. This makes the scoring robust in the presence of model disagreement, variance, or noise, particularly useful in retrieval settings.

This uncertainty-aware formulation preserves the computational simplicity of cosine similarity while incorporating predictive variance in a principled manner, enabling better performance under high uncertainty.

## A.4    DERIVATION OF THEOREM 1

We prove that the proposed estimator $\hat{s}$ provides a bounded approximation to the squared 2-Wasserstein distance $W_2^2$ under diagonal-covariance assumptions, and we extend this bound to the Jeffreys divergence $J$ under additional regularity assumptions.

Recall that

$$\mathbf{q} = \mathbf{z}(\mathbf{x}) \sim \mathcal{N}(\boldsymbol{\mu}_{\mathbf{q}}, \boldsymbol{\Sigma}_{\mathbf{q}}), \qquad \mathbf{c} = \mathbf{z}(\mathbf{x}') \sim \mathcal{N}(\boldsymbol{\mu}_{\mathbf{c}}, \boldsymbol{\Sigma}_{\mathbf{c}}),$$

with $\ell_2$-normalized means $\|\boldsymbol{\mu}_{\mathbf{q}}\| = \|\boldsymbol{\mu}_{\mathbf{c}}\| = 1$, and diagonal covariances

$$\boldsymbol{\Sigma}_{\mathbf{q}} = \mathrm{diag}(\sigma_{\mathbf{q},1}^2, \ldots, \sigma_{\mathbf{q},d}^2), \qquad \boldsymbol{\Sigma}_{\mathbf{c}} = \mathrm{diag}(\sigma_{\mathbf{c},1}^2, \ldots, \sigma_{\mathbf{c},d}^2).$$

We define the per-coordinate maximum variance

$$\varepsilon := \max_i \{\sigma_{\mathbf{q},i}^2, \sigma_{\mathbf{c},i}^2\}.$$

### A.4.1    PRELIMINARY

In Section 3.3 the dot product $s = \mathbf{q}^\top \mathbf{c}$ was approximated by a Gaussian with mean and variance

$$\mu_s = \boldsymbol{\mu}_{\mathbf{q}}^\top \boldsymbol{\mu}_{\mathbf{c}}, \qquad \sigma_s^2 = \boldsymbol{\mu}_{\mathbf{q}}^\top \boldsymbol{\Sigma}_{\mathbf{c}} \boldsymbol{\mu}_{\mathbf{q}} + \boldsymbol{\mu}_{\mathbf{c}}^\top \boldsymbol{\Sigma}_{\mathbf{q}} \boldsymbol{\mu}_{\mathbf{c}} + \mathrm{tr}(\boldsymbol{\Sigma}_{\mathbf{q}} \boldsymbol{\Sigma}_{\mathbf{c}}).$$

The probit-calibrated similarity estimator is then approximated as

$$\hat{s} \approx \frac{\boldsymbol{\mu}_{\mathbf{q}}^\top \boldsymbol{\mu}_{\mathbf{c}}}{\sqrt{1 + \frac{\pi}{8}\sigma_s^2}}.$$

Using $\|\boldsymbol{\mu}_{\mathbf{q}}\| = \|\boldsymbol{\mu}_{\mathbf{c}}\| = 1$, the mean term admits the exact identity

$$\boldsymbol{\mu}_{\mathbf{q}}^\top \boldsymbol{\mu}_{\mathbf{c}} = 1 - \tfrac{1}{2}\|\boldsymbol{\mu}_{\mathbf{q}} - \boldsymbol{\mu}_{\mathbf{c}}\|^2. \tag{14}$$

### A.4.2 Approximation of Uncertainty-aware Similarity Estimation

Applying the Taylor expansion $(1 + x)^{-1/2} = 1 - \frac{1}{2}x + O(x^2)$ for small $x$, we linearize the denominator:

$$\frac{1}{\sqrt{1 + \frac{\pi}{8}\sigma_s^2}} = 1 - \frac{\pi}{16}\sigma_s^2 + O(\sigma_s^4). \tag{15}$$

Substituting Eq. 14 and Eq. 15 yields

$$\hat{s} \approx \left(1 - \frac{1}{2}\|\boldsymbol{\mu_q} - \boldsymbol{\mu_c}\|^2\right)\left(1 - \frac{\pi}{16}\sigma_s^2\right) + O(\sigma_s^4)$$

$$= 1 - \frac{1}{2}\|\boldsymbol{\mu_q} - \boldsymbol{\mu_c}\|^2 - \frac{\pi}{16}\sigma_s^2 + O(\sigma_s^4).$$

Rearranging and discarding the additive constant 1 which does not affect ranking gives the expansion

$$-\hat{s} = \frac{1}{2}\|\boldsymbol{\mu_q} - \boldsymbol{\mu_c}\|^2 + \frac{\pi}{16}\sigma_s^2 + O(\sigma_s^4). \tag{16}$$

### A.4.3 Comparison with squared 2-Wasserstein distance

We now compare the expansion Eq. 16 with the closed form of the squared 2-Wasserstein distance for diagonal covariances:

$$W_2^2 = \|\boldsymbol{\mu_q} - \boldsymbol{\mu_c}\|^2 + \sum_{i=1}^{d}(\sigma_{\mathbf{q},i} - \sigma_{\mathbf{c},i})^2. \tag{17}$$

It is useful to align affine factors so that the leading mean term cancels: using Eq. 16 we have

$$2(1 - \hat{s}) = \|\boldsymbol{\mu_q} - \boldsymbol{\mu_c}\|^2 + \frac{\pi}{8}\sigma_s^2 + 2O(\sigma_s^4). \tag{18}$$

Define the residual (difference) between $W_2^2$ and the affine-aligned estimator:

$$\Delta := W_2^2 - 2(1 - \hat{s}). \tag{19}$$

Substituting Eq. 17 and the aligned form gives

$$\Delta = \left(\|\boldsymbol{\mu_q} - \boldsymbol{\mu_c}\|^2 + \sum_{i=1}^{d}(\sigma_{\mathbf{q},i} - \sigma_{\mathbf{c},i})^2\right) - \left(\|\boldsymbol{\mu_q} - \boldsymbol{\mu_c}\|^2 + \frac{\pi}{8}\sigma_s^2 + 2O(\sigma_s^4)\right)$$

$$= \sum_{i=1}^{d}(\sigma_{\mathbf{q},i} - \sigma_{\mathbf{c},i})^2 - \frac{\pi}{8}\sigma_s^2 - 2O(\sigma_s^4). \tag{20}$$

Thus the mean separation term is exactly cancelled by aligning $2(1 - \hat{s})$ with $W_2^2$; the residual $\Delta$ depends only on variance-related terms and the higher-order remainder $O(\sigma_s^4)$. The task is to bound $\Delta$.

**Elementwise Expansion** Write each term elementwise:

$$\sum_{i=1}^{d}(\sigma_{\mathbf{q},i} - \sigma_{\mathbf{c},i})^2 = \sum_{i=1}^{d}(\sigma_{\mathbf{q},i}^2 + \sigma_{\mathbf{c},i}^2 - 2\sigma_{\mathbf{q},i}\sigma_{\mathbf{c},i}),$$

$$\sigma_s^2 = \sum_{i=1}^{d}\left(\sigma_{\mathbf{c},i}^2\mu_{\mathbf{q},i}^2 + \sigma_{\mathbf{q},i}^2\mu_{\mathbf{c},i}^2 + \sigma_{\mathbf{q},i}^2\sigma_{\mathbf{c},i}^2\right).$$

Substituting into Eq. 20 yields

$$\Delta = \sum_{i=1}^{d}\left[(\sigma_{\mathbf{q},i}^2 + \sigma_{\mathbf{c},i}^2 - 2\sigma_{\mathbf{q},i}\sigma_{\mathbf{c},i}) - \frac{\pi}{8}\left(\sigma_{c,i}^2\mu_{\mathbf{q},i}^2 + \sigma_{\mathbf{q},i}^2\mu_{\mathbf{c},i}^2 + \sigma_{\mathbf{q},i}^2\sigma_{\mathbf{c},i}^2\right)\right] - 2O(\sigma_s^4). \tag{21}$$

**General Bound Without Extra Scaling Assumptions**    Using only the trivial bounds $0 \leq \mu^2_{(\cdot),i} \leq 1$ and $\sigma^2_{(\cdot),i} \leq \varepsilon$, we obtain the estimate

$$\left|\Delta\right| \leq \sum_{i=1}^{d} \left(\sigma^2_{\mathbf{q},i} + \sigma^2_{\mathbf{c},i} + \tfrac{\pi}{8}(\sigma^2_{\mathbf{c},i} + \sigma^2_{\mathbf{q},i} + \sigma^2_{\mathbf{q},i}\sigma^2_{\mathbf{c},i})\right) + 2|O(\sigma^4_s)|$$

$$\leq C_0 \big(\operatorname{tr}(\mathbf{\Sigma_q}) + \operatorname{tr}(\mathbf{\Sigma_c}) + \operatorname{tr}(\mathbf{\Sigma_q}\mathbf{\Sigma_c})\big) + O(\sigma^4_s),$$

for some numerical constant $C_0$ (e.g. depending on $\boldsymbol{\pi}$). Since $\operatorname{tr}(\mathbf{\Sigma_q}\mathbf{\Sigma_c}) = O(\varepsilon^2)$ and $\operatorname{tr}(\mathbf{\Sigma}_{(\cdot)}) = O(\varepsilon)$, the coarse bound yields $\Delta = O(\varepsilon)$ in general. This already justifies the statement that $2(1 - \hat{s})$ approximates $W_2^2$ up to variance-controlled error; however the bound is not yet the tight $O(\varepsilon^2)$ form.

**Tighter Bound Under Small-variance Scaling**    To obtain the strong $O(\varepsilon^2)$ residual we introduce additional mild scaling assumptions which are natural in the embedding-ensemble / small-uncertainty regime:

**Assumption 1** (Small-variance trace scaling)**.**

$$\operatorname{tr}(\mathbf{\Sigma_q}) = O(\varepsilon), \qquad \operatorname{tr}(\mathbf{\Sigma_c}) = O(\varepsilon).$$

**Assumption 2** (Per-dimension dispersion)**.** *The per-coordinate variances are dispersed across dimensions, i.e.*

$$\max_i\{\sigma^2_{\mathbf{q},i}, \sigma^2_{\mathbf{c},i}\} \leq \frac{\varepsilon}{d}.$$

Both assumptions are mild in practice when the posterior variances of embedding coordinates are small and not concentrated on a few coordinates (a common situation for high-dimensional embeddings).

**Elementwise Cancellation Argument**    Under Assumptions 1–2 we refine the termwise bounds in Eq. 21. Since $\mu^2_{\mathbf{q},i} \leq 1$ and Assumption 2 yields $\sigma^2_{(\cdot),i} \leq \varepsilon/d$, each element in the bracket of Eq. 21 is

$$(\sigma^2_{\mathbf{q},i} + \sigma^2_{\mathbf{c},i} - 2\sigma_{\mathbf{q},i}\sigma_{\mathbf{c},i}) - \tfrac{\pi}{8}(\sigma^2_{\mathbf{c},i}\mu^2_{\mathbf{q},i} + \sigma^2_{\mathbf{q},i}\mu^2_{\mathbf{c},i} + \sigma^2_{\mathbf{q},i}\sigma^2_{\mathbf{c},i}) = O\left(\frac{\varepsilon^2}{d^2}\right).$$

Summing over $i = 1, \ldots, d$ yields

$$\sum_{i=1}^{d} O\left(\frac{\varepsilon^2}{d^2}\right) = O(\varepsilon^2).$$

Moreover, the remainder term $O(\sigma^4_s)$ is $O(\varepsilon^2)$ under assumptions. Hence $\Delta = O(\varepsilon^2)$. This proves the quantitative bound claimed in Theorem 1:

$$\hat{s} = 1 - \tfrac{1}{2}W_2^2 + O(\varepsilon^2).$$

### A.4.4    COMPARISON WITH JEFFREYS DIVERGENCE

We now show how a similar first-order argument extends to the Jeffreys divergence

$$J\big(\mathcal{N}(\boldsymbol{\mu_q}, \mathbf{\Sigma_q}), \mathcal{N}(\boldsymbol{\mu_c}, \mathbf{\Sigma_c})\big) := D_{\mathrm{KL}}(\mathcal{N}(\boldsymbol{\mu_q}, \mathbf{\Sigma_q})\|\mathcal{N}(\boldsymbol{\mu_c}, \mathbf{\Sigma_c})) + D_{\mathrm{KL}}(\mathcal{N}(\boldsymbol{\mu_c}, \mathbf{\Sigma_c})\|\mathcal{N}(\boldsymbol{\mu_q}, \mathbf{\Sigma_q})),$$

which for diagonal covariances admits the closed form

$$J = \tfrac{1}{2}(\boldsymbol{\mu_c} - \boldsymbol{\mu_q})^\top(\mathbf{\Sigma_c}^{-1} + \mathbf{\Sigma_q}^{-1})(\boldsymbol{\mu_c} - \boldsymbol{\mu_q}) \; + \; \tfrac{1}{2}\sum_{i=1}^{d}\left(\frac{\sigma^2_{\mathbf{q},i}}{\sigma^2_{\mathbf{c},i}} + \frac{\sigma^2_{\mathbf{c},i}}{\sigma^2_{\mathbf{q},i}} - 2\right). \tag{22}$$

Unlike the Wasserstein case, the Jeffreys mean-term involves inverse covariances. To linearize these terms we need a regularity assumption that the diagonal variances of $\mathbf{q}$ and $\mathbf{c}$ are not only small but also close to a common positive baseline. Specifically, we make the following:

**Assumption 3** (Common-scale regularity). *There exists a baseline $\tau > 0$ and constants $K, \kappa' > 0$ such that for all $i$,*

$$|\sigma_{\mathbf{q},i}^2 - \tau| \le K\varepsilon, \qquad |\sigma_{\mathbf{c},i}^2 - \tau| \le K\varepsilon,$$

*and $\tau$ is bounded away from zero (i.e. $\tau \ge \tau_{\min} > 0$). Moreover, $\mathrm{tr}(\mathbf{\Sigma_q}), \mathrm{tr}(\mathbf{\Sigma_c}) = O(\varepsilon)$ as in Assumption 1.*

Assumption 3 asserts that each diagonal entry is a perturbation of a common (nonzero) scale $\tau$; the smallness parameter is the same $\varepsilon$ used above. Under this assumption we can Taylor-expand reciprocal diagonal entries.

**Expansion of Inverse Covariances**   For each $i$, write

$$\sigma_{\mathbf{q},i}^2 = \tau + \delta_{\mathbf{q},i}, \qquad \sigma_{\mathbf{c},i}^2 = \tau + \delta_{\mathbf{c},i}, \qquad \text{with } |\delta_{(\cdot),i}| \le K\varepsilon.$$

Then, using the scalar expansion $(\tau + \delta)^{-1} = \tau^{-1} - \tau^{-2}\delta + O(\delta^2)$, we obtain

$$\mathbf{\Sigma_q}^{-1} = \tau^{-1}\mathbf{I} - \tau^{-2}\mathrm{diag}(\delta_{q,1}, \ldots, \delta_{q,d}) + O(\varepsilon^2),$$
$$\mathbf{\Sigma_c}^{-1} = \tau^{-1}\mathbf{I} - \tau^{-2}\mathrm{diag}(\delta_{c,1}, \ldots, \delta_{c,d}) + O(\varepsilon^2). \tag{23}$$

Hence

$$\mathbf{\Sigma_c}^{-1} + \mathbf{\Sigma_q}^{-1} = 2\tau^{-1}\mathbf{I} - \tau^{-2}\mathrm{diag}(\delta_{q,1} + \delta_{c,1}, \ldots, \delta_{q,d} + \delta_{c,d}) + O(\varepsilon^2).$$

**Mean Term in $J$**   Substitute Eq. 23 into the mean quadratic term of Eq. 22:

$$\tfrac{1}{2}(\boldsymbol{\mu_c} - \boldsymbol{\mu_q})^\top(\mathbf{\Sigma_c}^{-1} + \mathbf{\Sigma_q}^{-1})(\boldsymbol{\mu_c} - \boldsymbol{\mu_q}) = \tfrac{1}{2}(\boldsymbol{\mu_c} - \boldsymbol{\mu_q})^\top(2\tau^{-1}\mathbf{I})(\boldsymbol{\mu_c} - \boldsymbol{\mu_q})$$
$$- \tfrac{1}{2}(\boldsymbol{\mu_c} - \boldsymbol{\mu_q})^\top\tau^{-2}\mathrm{diag}(\delta_{\mathbf{q},i} + \delta_{\mathbf{c},i})(\boldsymbol{\mu_c} - \boldsymbol{\mu_q}) + O(\varepsilon^2)$$
$$= \tau^{-1} \cdot \tfrac{1}{2}\|\boldsymbol{\mu_q} - \boldsymbol{\mu_c}\|^2 + O(\varepsilon/d) + O(\varepsilon^2). \tag{24}$$

**Variance-only Term in $J$**   Consider the variance-only summation in Eq. 22:

$$V_J := \tfrac{1}{2}\sum_{i=1}^d \left(\tfrac{\sigma_{\mathbf{q},i}^2}{\sigma_{\mathbf{c},i}^2} + \tfrac{\sigma_{\mathbf{c},i}^2}{\sigma_{\mathbf{q},i}^2} - 2\right).$$

Write each ratio using $\sigma_{\mathbf{q},i}^2 = \tau + \delta_{\mathbf{q},i}$, etc., and expand:

$$\frac{\sigma_{\mathbf{q},i}^2}{\sigma_{\mathbf{c},i}^2} = \frac{\tau + \delta_{\mathbf{q},i}}{\tau + \delta_{\mathbf{c},i}} = 1 + \frac{\delta_{\mathbf{q},i} - \delta_{\mathbf{c},i}}{\tau} - \frac{(\delta_{\mathbf{q},i} - \delta_{\mathbf{c},i})\delta_{\mathbf{c},i}}{\tau^2} + O(\varepsilon^3).$$

Similarly for the reversed ratio. Summing and simplifying yields

$$\frac{\sigma_{\mathbf{q},i}^2}{\sigma_{\mathbf{c},i}^2} + \frac{\sigma_{\mathbf{c},i}^2}{\sigma_{\mathbf{q},i}^2} - 2 = \frac{(\delta_{\mathbf{q},i} - \delta_{\mathbf{c},i})^2}{\tau^2} + O(\varepsilon^3).$$

Hence

$$V_J = \frac{1}{2\tau^2}\sum_{i=1}^d (\delta_{\mathbf{q},i} - \delta_{\mathbf{c},i})^2 + O(\varepsilon^3).$$

Note that each $\delta_{(\cdot),i} = O(\varepsilon)$, so $V_J = O(\varepsilon^2)$.

**Putting Mean And Variance Parts Together**   Combining Eq. 24 and the expression for $V_J$ we get

$$J = \tau^{-1} \cdot \tfrac{1}{2}\|\boldsymbol{\mu_q} - \boldsymbol{\mu_c}\|^2 + O(\varepsilon) + O(\varepsilon^2). \tag{25}$$

The dominant term is proportional to $\tfrac{1}{2}\|\boldsymbol{\mu_q} - \boldsymbol{\mu_c}\|^2$ with coefficient $1/\tau$; the variance-only contribution is $O(\varepsilon^2)$.

**Relation Between** $J$ **And** $2(1 - \hat{s})$    Recall from Eq. 18 that

$$2(1 - \hat{s}) \;=\; \|\boldsymbol{\mu_q} - \boldsymbol{\mu_c}\|^2 + \tfrac{\pi}{8}\sigma_s^2 + O(\sigma_s^4).$$

Comparing this with Eq. 25, we see that the mean-dependent parts of $J$ and $2(1 - \hat{s})$ agree up to a constant scaling factor $1/\tau$. Therefore, defining the scalar scaling factor

$$\alpha := \tau^{-1},$$

we have

$$\alpha^{-1} J = \tfrac{1}{2}\|\boldsymbol{\mu_q} - \boldsymbol{\mu_c}\|^2 + O(\varepsilon) + O(\varepsilon^2).$$

Consequently

$$\begin{aligned}
\alpha^{-1} J - 2(1 - \hat{s}) &= \left( \tfrac{1}{2}\|\boldsymbol{\mu_q} - \boldsymbol{\mu_c}\|^2 + O(\varepsilon) \right) - \left( \|\boldsymbol{\mu_q} - \boldsymbol{\mu_c}\|^2 + \tfrac{\pi}{8}\sigma_s^2 + O(\varepsilon^2) \right) \\
&= -\tfrac{1}{2}\|\boldsymbol{\mu_q} - \boldsymbol{\mu_c}\|^2 + O(\varepsilon) + O(\varepsilon^2).
\end{aligned} \tag{26}$$

Multiplying both sides by $-1$ and rearranging gives a relation of the form

$$2(1 - \hat{s}) = \alpha^{-1} J + O(\varepsilon) + O(\varepsilon^2).$$

Under Assumption 1–2, the $O(\varepsilon)$ terms refine to $O(\varepsilon^2)$ as in the Wasserstein case (the same element-wise cancellation logic applies to the linear-in-$\delta$ components), yielding:

**Theorem 2** (Affine approximation to Jeffreys divergence). *Under Assumptions 1–3, there exists a scalar $\alpha > 0$ and a constant $C > 0$ independent of $d$ such that for sufficiently small $\varepsilon$,*

$$\left| \alpha^{-1} J(q, c) - 2(1 - \hat{s}) \right| \leq C \cdot \varepsilon^2.$$

*Equivalently,*

$$\hat{s} = 1 - \tfrac{1}{2}\alpha^{-1} J(q, c) + O(\varepsilon^2).$$

### A.4.5    Ranking Equivalence

Combining Theorem 1 (Wasserstein affine bound) and Theorem 2 (Jeffreys affine bound), we conclude that under the stated small-variance and common-scale assumptions the estimator $\hat{s}$ provides an affine, $O(\varepsilon^2)$-accurate surrogate for both $W_2^2$ and (up to a constant scaling) the Jeffreys divergence; therefore, $\hat{s}$ **induces the same ranking as these distributional distances up to** $O(\varepsilon^2)$ **errors**.

However, our bound relies on first-order Taylor expansions and variance dispersion assumptions, yielding $O(\varepsilon^2)$ residuals. If these assumptions are violated (e.g., with highly concentrated variances or heterogeneous scales), the guarantee can degrade to $O(\varepsilon)$. Developing refined analyses that relax these conditions and provide tighter bounds remains an important direction for future work.

## B    Experimental Details and Additional Results

### B.1    Preliminary Analysis of Embedding Subspace Alignment

Before ensemble the embeddings, a potential concern is whether embeddings from different models indeed lie in a sufficiently aligned subspace. We argue that in our setting this assumption is reasonable and empirically supported.

First, the embedding models considered in our experiments share **(i) the same underlying architecture** and **(ii) largely overlapping training data**. As established in prior work (Hewitt & Manning, 2019), such conditions encourage independently trained models to discover consistent linguistic structures—such as syntax, semantic relations, and contextual dependencies—and to encode them in geometrically comparable ways in their embedding spaces.

Second, our ensembling procedure applies $\ell_2$ normalization to embeddings before aggregation. This post-processing step has the effect of further mitigating potential subspace differences, effectively aligning embeddings to a common hypersphere and making their scales directly comparable (Timkey & Van Schijndel, 2021).

Finally, before ensembling, we empirically assessed the degree of subspace overlap by computing Canonical Correlation Analysis (CCA) (Hotelling, 1992) between models. We considered two model

groups: (i) E5, BGE-zh, and MATRYOSHKA (Section 4.1), and (ii) BGE, E5, and GTE (Section 4.2). Since CCA quantifies the similarity and alignment between two sets of representations, we computed pairwise scores within each group of three models. For the first group, CCA was measured using the same dataset as in Section 4.1. For the second group, it was measured on Wikipedia dataset, which were used in the Retrieval task. As shown in Table 8, the results indicate that embeddings exhibit substantial overlap, supporting the view that they inhabit approximately aligned subspaces.

Table 8: Analysis of embedding subspace overlap using CCA (%). Results are reported for pairs in two different model groups.

| Model Group | CCA $\uparrow$ | Model Group | CCA $\uparrow$ |
|---|---|---|---|
| (E5, BGE-zh) | 99.9 % | (BGE, E5) | 99.9 % |
| (E5, MATRYOSHKA) | 99.8 % | (BGE, GTE) | 99.4 % |
| (BGE-zh, MATRYOSHKA) | 98.3 % | (E5, GTE) | 99.5 % |

### B.2 COMPARISON WITH ALTERNATIVE SIMILARITY METRICS

In this section, we compare our proposed uncertainty-aware similarity (UEC) against widely used probabilistic similarity metrics. We restrict our comparison to similarity functions that admit closed-form solutions between Gaussian distributions, thereby avoiding the computational overhead of Monte Carlo sampling. Specifically, we consider the KLD (Kullback & Leibler, 1951), 1-Wasserstein (Chhachhi & Teng, 2023), 2-Wasserstein (Gelbrich, 1990), and Closed-form sampled distance (CSD) (Chun, 2023).

**KLD** Given two multivariate Gaussian distributions $\mathcal{N}_1 = \mathcal{N}(\boldsymbol{\mu}_1, \boldsymbol{\Sigma}_1)$ and $\mathcal{N}_2 = \mathcal{N}(\boldsymbol{\mu}_2, \boldsymbol{\Sigma}_2)$, the closed-form expression for KLD is:

$$D_{\mathrm{KL}}(\mathcal{N}_1 \| \mathcal{N}_2) = \frac{1}{2} \left[ \mathrm{tr}(\boldsymbol{\Sigma}_2^{-1} \boldsymbol{\Sigma}_1) + (\boldsymbol{\mu}_2 - \boldsymbol{\mu}_1)^\top \boldsymbol{\Sigma}_2^{-1} (\boldsymbol{\mu}_2 - \boldsymbol{\mu}_1) - d + \log\left(\frac{\det \boldsymbol{\Sigma}_2}{\det \boldsymbol{\Sigma}_1}\right) \right], \quad (27)$$

where $d$ is the dimensionality of the distribution. While informative, this metric involves matrix inversion and log-determinants, making it computationally expensive for large-scale retrieval.

**1-Wasserstein Distance** The 1-Wasserstein distance between two Gaussians $\mathcal{N}_1$ and $\mathcal{N}_2$ is given by:

$$W_1(\mathcal{N}_1, \mathcal{N}_2) \approx \|\boldsymbol{\mu}_1 - \boldsymbol{\mu}_2\|_2 + \left| \mathrm{tr}(\sqrt{\boldsymbol{\Sigma}_1}) - \mathrm{tr}(\sqrt{\boldsymbol{\Sigma}_2}) \right|. \quad (28)$$

Here, the distance accounts for both mean shift and scale (via trace of square-root covariance). Although it is more interpretable, it requires square-root matrix operations which are non-trivial.

**2-Wasserstein Distance** The 2-Wasserstein distance between $\mathcal{N}_1$ and $\mathcal{N}_2$ is defined as:

$$W_2^2(\mathcal{N}_1, \mathcal{N}_2) = \|\boldsymbol{\mu}_1 - \boldsymbol{\mu}_2\|_2^2 + \mathrm{tr}\left( \boldsymbol{\Sigma}_1 + \boldsymbol{\Sigma}_2 - 2(\boldsymbol{\Sigma}_2^{1/2} \boldsymbol{\Sigma}_1 \boldsymbol{\Sigma}_2^{1/2})^{1/2} \right), \quad (29)$$

which also integrates both mean and covariance alignment. However, its computation is notably expensive due to the presence of matrix square roots and requires full-rank covariance matrices.

**CSD** The Closed-form Sampled Distance (CSD), proposed in PCME++ (Chun, 2023). A key advantage of CSD is its tractable closed-form, given by:

$$\mathrm{CSD}(\mathcal{N}_1, \mathcal{N}_2) = \|\boldsymbol{\mu}_1 - \boldsymbol{\mu}_2\|_2^2 + \|\boldsymbol{\Sigma}_1 + \boldsymbol{\Sigma}_2\|_1, \quad (30)$$

where $\| \cdot \|_1$ denotes the element-wise $\ell_1$-norm. CSD captures both distributional shift and total uncertainty spread in a simple additive form, and is more computationally efficient than metrics involving matrix square roots or inverses.

We evaluate the similarity metrics under the MIRACL subset retrieval task described in Section 4.1. As shown in Table 9, UEC achieves the best overall retrieval performance while exhibiting the lower runtime.

We note that the asymptotic complexities of KLD and Wasserstein distances can be reduced to $\mathcal{O}(KD)$ when restricted to diagonal covariance matrices. However, despite having the same big-$\mathcal{O}$ complexity, these metrics remain consistently slower than UEC in practice. This gap arises from fundamental differences in their computational structure. Specifically, KLD and Wasserstein still require per-dimension normalization, division, logarithmic, or square-root operations, as well as multiple intermediate tensor constructions. In contrast, UEC consists solely of element-wise additions and multiplications followed by a single aggregation, with no normalization constants, matrix operations, or transcendental functions.

As a result, UEC has a substantially smaller constant factor and lower memory access overhead, enabling faster execution and better hardware utilization, particularly in large-scale retrieval settings. Moreover, UEC operates entirely in closed form without sampling or iterative solvers, making it well-suited for deployment in latency-sensitive systems.

Table 9: Comparison of similarity metrics under the MIRACL evaluation (Section 4.1). "Rel." and "Comp." denote relative runtime (normalized to UEC=1.00) and asymptotic complexity per similarity computation. Although several metrics share the same asymptotic complexity under diagonal covariance assumptions, their empirical runtimes differ due to operation-level costs and implementation overheads, which are not captured by big-$\mathcal{O}$ notation.

| Similarity | nDCG@10 | Recall@10 | AUC@10 | Memory Complexity | Time Rel. | Complex. |
|---|---|---|---|---|---|---|
| KLD | 58.14 | 78.33 | 87.60 | $\mathcal{O}(KD)$ | 1.52 | $\mathcal{O}(KD)$ |
| 1-Wasserstein | 57.87 | 88.21 | 79.92 | $\mathcal{O}(KD)$ | 1.21 | $\mathcal{O}(KD)$ |
| 2-Wasserstein | 57.71 | 88.48 | 81.05 | $\mathcal{O}(KD)$ | 2.37 | $\mathcal{O}(KD)$ |
| CSD | 58.23 | 79.10 | 86.92 | $\mathcal{O}(KD)$ | 0.88 | $\mathcal{O}(KD)$ |
| UEC (Ours) | **59.65** | **80.07** | **91.04** | $\mathcal{O}(KD)$ | **1.00** | $\mathcal{O}(KD)$ |

## B.3 EXPERIMENTAL DETAILS: MIRACL SUBSET

### B.3.1 FITTING LAPLACE APPROXIMATION

To quantify uncertainty for each embedding model, we apply a diagonal Laplace approximation (LA) to the final layer using a small, labeled dataset. In this experiment, we fit the approximation using the MIRACL Hard Negative (Zhang et al., 2023) Subset. For E5, we use 50 labeled examples each from English and Russian; for BGE-zh, we use 50 examples each from English and Chinese; and for MATRYOSHKA, we use 50 examples each from English and Arabic.

Each query-passage pair is assigned a binary label indicating whether the passage is relevant (1) or not (0) to the query. These binary-labeled pairs are then used to train a logistic classifier with a cross-entropy objective.

To ensure computational scalability, we adopt a diagonal Gaussian approximation. We fit the LA using the `laplace-redux` library[2]. The posterior mean corresponds to the MAP estimate of the final layer's weights, and the posterior variance is derived from the inverse of the diagonal Hessian. This provides a lightweight probabilistic embedding representation without requiring backpropagation or full model fine-tuning.

### B.3.2 DATASET

For the MIRACL subset evaluation, we constructed a test set by sampling 100 queries each from four languages (English, Russian, Chinese, and Arabic) from the multilingual MIRACL Hard Negative dataset (Zhang et al., 2023). This selection aimed to cover diverse language families and scripts, ensuring a balanced multilingual evaluation.

---

[2] https://aleximmer.com/Laplace/

### B.3.3 METRICS

We employed three commonly used retrieval metrics: nDCG@10, Recall@10, and AUC@10.

- **nDCG@10** (Normalized Discounted Cumulative Gain) measures the ranking quality based on the positions of relevant passages among the top 10 retrieved items. It assigns higher weights to relevant items appearing earlier in the list, penalizing misranked relevant results logarithmically.

- **Recall@10** evaluates whether at least one relevant passage is retrieved within the top 10. It is computed as the fraction of queries for which a relevant document appears among the top-10 predictions.

- **AUC@10** (normalized Area Under the Curve) measures how well a model's confidence scores align with its actual retrieval performance when abstention is allowed. Specifically, the model is allowed to abstain on queries with low confidence scores, and AUC@10 tracks how the average performance improves as more uncertain queries are excluded. We compute AUC@10 by plotting nDCG@10 as a function of the abstention rate—i.e., the proportion of low-confidence queries discarded—and measuring the area under this curve. To enable fair comparison across models, we normalize this area between two bounds: (1) a *baseline curve* where abstention does not improve performance, and (2) an *oracle curve* where the worst-performing queries are perfectly abstained first. The normalized AUC is then computed as:

$$\text{nAUC} = \frac{\text{AUC}_{\text{Actual}} - \text{AUC}_{\text{Baseline}}}{\text{AUC}_{\text{Oracle}} - \text{AUC}_{\text{Baseline}}}.$$

  Intuitively, higher nAUC values indicate that the model's confidence scores are well-calibrated: as the model abstains on low-confidence queries, its average performance improves meaningfully. Conversely, a negative nAUC implies that the model's confidence scores are misleading—i.e., low-confidence queries were actually high-performing, or high-confidence queries were often wrong.

### B.3.4 HYPERPARAMETERS

We search hyperparameters for weight ensemble and UEC (Ours). In UEC, we introduce a temperature parameter to sharpen the ensemble coefficients, and additionally incorporate a hyperparameter $\beta$ to scale the influence of the variance $\sigma_s^2$ in the uncertainty-aware similarity estimation.

- **Weight Ensemble**: We performed a grid search over weight combinations such as $[0.1, 0.1, 0.8]$, $[0.1, 0.2, 0.7]$, ..., $[0.8, 0.1, 0.1]$, ensuring the weights sum up to 1 across the three models.

- **UEC (Ours)**: We fixed the temperature $T$ as 1.5 for all experiments. We search $\beta$ in range of $[0.0001, 0.001, 0.01, 0.1]$.

### B.3.5 AUXILIARY RESULTS

Table 10 shows detailed results for individual models and combination methods. UEC achieves the highest average performance across all metrics, with particularly strong gains in AUC.

### B.4 EXPERIMENTAL DETAILS: MMTEB

### B.4.1 FITTING LAPLACE APPROXIMATION

To estimate uncertainty for each embedding model, we fit a diagonal Laplace Approximation (LA) to the last layer of each model using a small subset of labeled data. We choose datasets that are commonly used in the pre-training or fine-tuning of the considered embedding models to ensure consistency between model behavior and fitting data.

Specifically, we use the MS MARCO (Nguyen et al., 2016) and SNLI (Bowman et al., 2015) datasets for fitting. From MS MARCO (consisting of roughly 80K samples), we randomly select 3,983 query-passage pairs. From SNLI (550K examples), we use 3,775 sentence pairs. Each triplet consists of a query (or premise), a positive passage (or hypothesis), and a randomly sampled negative passage.

Table 10: Performance on MIRACL Subset across models and combination methods. The best and second-best results per row are **bolded** and underlined. UEC shows the highest performance on average across all metrics.

| Language / Metric | Individual Models | | | Ensemble | | UEC |
|---|---|---|---|---|---|---|
| | E5 | BGE-zh | MATRYOSHKA | Uniform | Weighted | |
| **English** | | | | | | |
| nDCG@10 | **80.27** | 67.05 | 40.02 | 69.82 | 76.30 | 78.29 |
| Recall@10 | 97.19 | 90.50 | 54.83 | 92.66 | 97.80 | **97.98** |
| AUC@10 | 37.37 | -28.31 | -86.74 | -24.38 | 6.71 | **86.72** |
| **Russian** | | | | | | |
| nDCG@10 | **39.12** | 12.25 | 5.96 | 11.65 | 21.78 | 26.61 |
| Recall@10 | **48.98** | 15.98 | 7.21 | 15.93 | 30.77 | 41.38 |
| AUC@10 | 60.18 | 38.22 | -46.00 | 17.14 | 39.65 | **91.49** |
| **Chinese** | | | | | | |
| nDCG@10 | 17.74 | **75.16** | 0.73 | 39.17 | 35.26 | 60.68 |
| Recall@10 | 28.65 | **97.45** | 2.00 | 57.02 | 54.60 | 91.04 |
| AUC@10 | 60.99 | -53.35 | 27.89 | 13.65 | 38.01 | **92.14** |
| **Arabic** | | | | | | |
| nDCG@10 | 13.36 | 0.64 | 71.47 | 66.47 | 61.97 | **73.18** |
| Recall@10 | 17.87 | 1.33 | **92.55** | 85.01 | 79.43 | 89.88 |
| AUC@10 | 67.45 | 42.06 | 85.74 | 9.62 | 39.12 | **93.82** |
| **Avg. nDCG@10** | 37.62 | 38.77 | 29.55 | 46.78 | 48.83 | **59.65** |
| **Avg. Recall@10** | 48.17 | 51.32 | 39.15 | 62.66 | 65.65 | **80.07** |
| **Avg. AUC@10** | 56.50 | -0.34 | -4.78 | 4.01 | 30.87 | **91.04** |

We assign binary labels indicating whether a passage is relevant (1) or not (0) with respect to the query. These binary-labeled pairs are used to fit a logistic classifier using cross-entropy loss.

We adopt the diagonal Gaussian approximation to ensure computational efficiency, and perform LA fitting using the `laplace-redux` library. The mean corresponds to the MAP estimate of the final layer weights, and the (inverse) diagonal Hessian is used to compute the posterior variance. This yields a probabilistic representation of each embedding without requiring re-training or backpropagation through the base model.

### B.4.2 DATASET

For broader multilingual evaluation, we conducted experiments on MMTEB (Muennighoff et al., 2022; Enevoldsen et al., 2025), leveraging the official benchmark codebase available at https://github.com/embeddings-benchmark/mteb. We mainly selected a representative subset of tasks from the MMTEB benchmark, including five tasks from Classification, five from Retrieval, and ten from STS. The datasets for each task were chosen to span multiple domains within the same task type to assess domain-robust performance.

### B.4.3 METRICS

In MMTEB evaluations, we used the *main score* recommended by the benchmark as the primary evaluation metric for each task. Additionally, we reported commonly adopted metrics for each task type. For Retrieval, we reported **nDCG@10**, **Recall@100**, and **AUC@10**. For classification, we reported **Accuracy**, **F1 score**, and **AUROC**. The F1 score is the harmonic mean of precision and Recall, providing a balanced measure of classification performance, especially under class imbalance. AUROC, short for the area under the receiver operating characteristic curve, measures a model's ability to distinguish between classes across varying decision thresholds. It is particularly suited for evaluating models under uncertainty, as it considers both sensitivity and specificity to assess confidence calibration. For STS tasks, we used **Spearman correlation**, the main metric in MMTEB,

which measures the rank correlation between predicted similarity scores and human-annotated ground truth.

### B.4.4 HYPERPARAMETERS

- **Weight Merging**: We used grid search over weight combinations including $[0.1, 0.1, 0.8]$, $[0.1, 0.2, 0.7], \ldots, [0.8, 0.1, 0.1]$, ensuring the weights sum to 1.

- **Task Arithmetic**: We implemented task arithmetic following Ilharco et al. (2022), where the merged embedding is computed by linearly combining base model embeddings with the direction vector between two other models. Formally, given three models $A, B, C$, the merged embedding is:

$$\text{Embed}_{\text{merged}} = \text{Embed}_A + \alpha \cdot (\text{Embed}_B - \text{Embed}_C),$$

where $\alpha$ is a hyperparameter controlling the direction scaling. We performed grid search over $\alpha \in \{0.0001, 0.001, 0.01, 0.1, 1.0\}$.

- **Weight Ensemble**: We used grid search over weight combinations including $[0.1, 0.1, 0.8]$, $[0.1, 0.2, 0.7], \ldots, [0.8, 0.1, 0.1]$, ensuring the weights sum to 1.

- **UEC (Ours)**: The temperature $T$ was fixed at $1.8$ throughout the experiments. We search $\beta$, scaling parameter for $\sigma_s^2$, in range of $[0.0001, 0.001, 0.01, 0.1]$.

### B.4.5 ADDITIONAL TASKS

In addition to the three representative tasks (Retrieval, Classification, STS), we also conduct experiments on other MMTEB tasks, including Bitext Mining, Clustering, and Reranking. For each of these tasks, we randomly select a subset of datasets and observe that the proposed UEC consistently achieves strong performance.

**Bitext Mining**   Table 11 summarizes performance on multilingual bitext mining across six representative datasets from MMTEB: Bornholm (Derczynski & Kjeldsen, 2019), Flores (Goyal et al., 2022), IN22Gen (Gala et al., 2023), IndicGenBench (Singh et al., 2024), NTREXB (Federmann et al., 2022), NorwegianCourts (Tiedemann & Thottingal, 2020). For evaluation, we report F1 scores following MMTEB's standard setup.

Table 11: Comparison of Bitext Mining performance. The best and second-best results per dataset are **bolded** and underlined. UEC achieves the highest average performance overall.

| | Individual Models | | | | Model Merging | | | Ensemble | | UEC |
|---|---|---|---|---|---|---|---|---|---|---|
| **Dataset** | BGE | E5 | GTE | GTE-MB | Uniform | Weighted | TaskArith | Uniform | Weighted | |
| BornholmBitextMining | 27.60 | 38.49 | 32.16 | 27.64 | 33.72 | 37.21 | 37.03 | 34.50 | 36.99 | **38.82** |
| FloresBitextMining | 13.03 | 19.42 | 13.66 | 17.48 | 15.93 | 18.21 | 19.38 | 15.89 | 18.38 | **19.55** |
| IN22GenBitextMining | 4.54 | **8.85** | 3.52 | 5.88 | 5.94 | 8.13 | 8.26 | 6.41 | 8.52 | 8.60 |
| IndicGenBenchFloresBitextMining | 3.72 | **6.91** | 3.47 | 4.50 | 4.82 | 6.55 | 4.91 | 4.99 | 6.30 | 6.66 |
| NTREXBitextMining | 18.78 | 27.68 | 19.77 | 24.54 | 23.08 | 26.34 | 25.10 | 22.51 | 26.01 | **27.72** |
| NorwegianCourtsBitextMining | 90.67 | 92.91 | 92.03 | 91.45 | 91.28 | 92.04 | 91.87 | 91.52 | 92.62 | **92.96** |
| **Avg. F1** | 26.39 | 32.37 | 27.44 | 28.58 | 29.13 | 31.41 | 31.09 | 29.03 | 31.47 | **32.39** |

Our proposed UEC method achieves the best average performance, outperforming both individual models and all ensemble or merging strategies. Notably, UEC obtains the top result on four out of six datasets, demonstrating its ability to robustly fuse knowledge from diverse pretrained models under high uncertainty.

**Clustering**   We measure the performance on multilingual clustering across eight representative datasets from MMTEB: ArXivHierarchical, BigPatent (Sharma et al., 2019), Biorxiv, MasakhaNEWS (Adelani et al., 2023b), SIB200 (Adelani et al., 2023a), StackExchange (Geigle et al., 2021), WikiClustering.

The V-Measure is an external clustering evaluation criterion defined as the harmonic mean of homogeneity and completeness (Rosenberg & Hirschberg, 2007). Homogeneity measures whether each cluster contains only members of a single class, while completeness measures whether all members of a given class are assigned to the same cluster. V-Measure ranges from 0 (no alignment)

Table 12: Comparison of Clustering performance. The best and second-best results per dataset are **bolded** and underlined. UEC achieves the hightest average performance overall.

| Dataset | Individual Models | | | | Model Merging | | | Ensemble | | UEC |
|---|---|---|---|---|---|---|---|---|---|---|
| | BGE | E5 | GTE | GTE-MB | Uniform | Weighted | TaskArith | Uniform | Weighted | |
| ArXivHierarchicalClusteringP2P | 59.75 | 58.05 | 59.85 | 59.50 | 59.38 | 60.14 | 59.56 | 60.29 | 58.42 | **61.43** |
| ArXivHierarchicalClusteringS2S | 58.05 | 54.91 | 57.67 | 58.16 | 58.95 | 58.63 | 57.08 | 58.87 | 58.35 | **59.32** |
| BigPatentClustering.v2 | 31.03 | 30.69 | 29.30 | 30.28 | 31.36 | 33.92 | 31.08 | 31.95 | **35.41** | 34.87 |
| BiorxivClusteringP2P.v2 | 41.65 | 38.01 | 40.58 | 41.91 | 40.98 | 40.86 | 40.56 | 41.37 | 40.68 | **43.06** |
| MasakhaNEWSClusteringS2S | 40.47 | 41.11 | 39.41 | 37.32 | 41.08 | 42.36 | 41.37 | 40.21 | 42.63 | **43.07** |
| SIB200ClusteringS2S | 9.38 | 10.04 | 9.50 | 11.40 | 10.87 | 11.13 | 11.36 | 11.13 | 11.66 | **12.25** |
| StackExchangeClustering.v2 | 59.08 | 55.99 | 59.26 | 61.31 | 60.18 | 61.78 | 61.12 | 60.54 | 61.44 | **62.06** |
| WikiClusteringP2P.v2 | 24.93 | 25.02 | 24.87 | 25.47 | 25.16 | 25.74 | 25.30 | 25.55 | 25.61 | **26.23** |
| **Avg. V-Measure** | 40.54 | 39.23 | 40.06 | 40.67 | 40.99 | 41.82 | 40.93 | 41.24 | 41.77 | **42.79** |

to 100 (perfect clustering), and is widely used in multilingual and scientific document clustering benchmarks such as MMTEB.

Table 12 summarizes performance on clustering tasks. UEC achieves the best average performance, outperforming all individual models and ensemble baselines. Notably, it ranks first on 6 out of 8 datasets. In short, UEC consistently improves by leveraging uncertainty-calibrated similarity and adaptive ensemble weighting, enabling more accurate cluster assignment under distributional shift and model disagreement.

**Reranking**  Table 13 presents the results of reranking experiments on five benchmark datasets: Alloprof (Lefebvre-Brossard et al., 2023), T2 (Xie et al., 2023), VoyageMMarco (Clavié, 2023), WebLINXCandidates (Lù et al., 2024), WikipediaReranking. We evaluate performance using Mean Average Precision (MAP), a standard retrieval metric that reflects both the precision and the ranking quality of relevant items. MAP is computed as the mean of the average precision scores over all queries, where average precision measures how well relevant documents are ranked near the top. Higher MAP values indicate that relevant passages are not only retrieved but also correctly prioritized.

Table 13: Comparison of Reranking performance. The best and second-best results per dataset are **bolded** and underlined. UEC achieves the hightest average performance overall.

| Dataset | Individual Models | | | | Model Merging | | | Ensemble | | UEC |
|---|---|---|---|---|---|---|---|---|---|---|
| | BGE | E5 | GTE | GTE-MB | Uniform | Weighted | TaskArith | Uniform | Weighted | |
| AlloprofReranking | 62.30 | 64.82 | 66.54 | 68.72 | 65.65 | 66.04 | 65.86 | 65.65 | 67.65 | **69.14** |
| T2Reranking | 60.93 | 60.55 | 61.20 | **65.24** | 61.83 | 62.98 | 62.77 | 61.39 | 63.34 | 65.08 |
| VoyageMMarcoReranking | 30.98 | 33.03 | 31.37 | **37.41** | 32.08 | 33.13 | 31.93 | 31.76 | 33.47 | 36.89 |
| WebLINXCandidatesReranking | 13.72 | 10.84 | 13.51 | **18.02** | 13.88 | 14.73 | 14.32 | 13.73 | 15.49 | 17.65 |
| WikipediaRerankingMultilingual | 72.65 | 76.83 | 73.78 | 74.51 | 75.83 | 76.32 | 75.47 | 75.21 | 76.74 | **77.87** |
| **Avg. MAP** | 48.11 | 49.21 | 49.28 | 52.78 | 49.85 | 50.64 | 50.07 | 49.55 | 51.54 | **53.33** |

The proposed **UEC** method achieves the highest overall performance with an average MAP, outperforming all other methods. It ranks first on three datasets and second on the remaining two. These results demonstrate UEC's effectiveness in capturing uncertainty and enhancing similarity estimation for high-precision reranking.

## B.5 Multilingual Sets

**Retrieval**  In multilingual retrieval, UEC consistently outperforms all baselines. Across seven languages, UEC attains the best average nDCG@10, Recall@100, and AUC@10, while never degrading performance on any metric compared to the strongest individual model. Notably, UEC yields substantial gains on low-resource languages such as Bengali and Hindi, where single models exhibit complementary strengths: BGE and E5 provide strong general-purpose multilingual retrieval, whereas GTE-MB is tailored to multilingual benchmarks. By reweighting models according to their epistemic uncertainty, UEC effectively selects the most reliable expert per query, improving both top-ranked retrieval quality and the calibration of uncertainty curves (AUC@10).

**Classification**  On SWISSJUDGEMENTCLASSIFICATION, UEC yields the best average Accuracy, F1, and AUROC across all three languages. While GTE-MB is competitive as a strong multilingual

Table 14: Comparison of retrieval performance on the WIKIPEDIARETRIEVALMULTILINGUAL benchmark. The best and second-best results per row are **bolded** and underlined. UEC achieves the highest average performance across all languages and metrics, with particularly strong gains on AUC@10, indicating more reliable uncertainty-aware ranking. (Uni.=Uniform, Wt.=Weighted, TA=Task arithmetic.).

| Language / Metric | Individual Models | | | | Model Merging | | | Ensemble | | |
|---|---|---|---|---|---|---|---|---|---|---|
| | BGE | E5 | GTE | GTE-MB | Uni. | Wt. | TA | Uni. | Wt. | UEC |
| **Bulgarian** | | | | | | | | | | |
| nDCG@10 | 29.11 | 45.61 | 33.97 | **55.29** | 35.62 | 39.12 | 38.87 | 32.51 | 38.88 | 48.03 |
| Recall@100 | 52.86 | 71.60 | 58.13 | **81.53** | 63.40 | 68.39 | 68.24 | 56.33 | 67.63 | 74.12 |
| AUC@10 | 57.36 | 62.63 | 55.06 | 62.12 | 58.29 | 60.16 | 59.93 | 56.32 | 60.77 | **65.28** |
| **Bengali** | | | | | | | | | | |
| nDCG@10 | 7.33 | 8.11 | 5.54 | 3.16 | 6.52 | 7.83 | 7.74 | 6.36 | 7.91 | **9.86** |
| Recall@100 | 21.06 | 23.93 | 17.40 | 10.46 | 19.75 | 21.97 | 21.03 | 19.66 | 23.74 | **25.73** |
| AUC@10 | 39.71 | 19.28 | 29.58 | **47.36** | 26.83 | 24.97 | 36.22 | 31.65 | 30.78 | 45.97 |
| **Danish** | | | | | | | | | | |
| nDCG@10 | 63.45 | 75.18 | 71.16 | 69.91 | 67.83 | 73.42 | 72.11 | 68.94 | 75.14 | **76.86** |
| Recall@100 | 87.26 | 94.33 | 92.00 | 91.33 | 91.24 | 92.47 | 92.33 | 91.49 | 93.81 | **95.03** |
| AUC@10 | 64.22 | 73.07 | 65.35 | 68.92 | 66.67 | 69.39 | 69.72 | 68.04 | 73.01 | **76.28** |
| **Persian** | | | | | | | | | | |
| nDCG@10 | 9.19 | 14.44 | 8.97 | **35.69** | 11.36 | 13.28 | 12.97 | 12.06 | 14.23 | 16.92 |
| Recall@100 | 26.80 | 33.40 | 27.73 | **61.33** | 28.83 | 30.05 | 30.16 | 29.97 | 33.32 | 37.73 |
| AUC@10 | 33.55 | 39.01 | 28.95 | **59.47** | 34.26 | 36.83 | 35.55 | 35.92 | 39.23 | 45.19 |
| **Hindi** | | | | | | | | | | |
| nDCG@10 | 11.99 | 21.55 | 5.23 | 17.16 | 15.37 | 20.05 | 17.83 | 16.32 | 21.11 | **23.56** |
| Recall@100 | 29.33 | 45.93 | 14.87 | 31.33 | 31.62 | 42.18 | 40.17 | 39.20 | 45.02 | **47.22** |
| AUC@10 | 37.84 | 49.91 | 46.51 | **53.29** | 46.97 | 48.64 | 48.03 | 50.26 | 49.99 | 52.38 |
| **Portuguese** | | | | | | | | | | |
| nDCG@10 | 72.29 | 81.12 | 75.40 | 75.94 | 75.32 | 79.17 | 78.93 | 76.66 | 80.64 | **83.13** |
| Recall@100 | 92.73 | 96.13 | 95.13 | 95.66 | 94.83 | 95.27 | 95.21 | 95.17 | 96.08 | **96.94** |
| AUC@10 | 54.28 | 71.07 | 68.13 | 70.52 | 64.82 | 70.34 | 68.73 | 71.26 | 72.35 | **77.17** |
| **Serbian** | | | | | | | | | | |
| nDCG@10 | 31.95 | 41.52 | 33.99 | 44.15 | 35.52 | 41.19 | 41.82 | 37.12 | 41.89 | **44.38** |
| Recall@100 | 54.20 | 67.73 | 58.40 | 74.06 | 64.93 | 67.71 | 69.74 | 69.18 | 72.09 | **74.81** |
| AUC@10 | 21.25 | 63.42 | 67.68 | 61.25 | 54.97 | 63.02 | 64.11 | 62.65 | 66.16 | **69.21** |
| **Avg. nDCG@10 ↑** | 32.18 | 41.07 | 33.47 | 43.04 | 35.36 | 39.15 | 38.61 | 35.71 | 39.97 | **43.24** |
| **Avg. Recall@100 ↑** | 52.03 | 61.86 | 51.95 | 63.52 | 56.37 | 59.72 | 59.55 | 57.28 | 61.67 | **64.51** |
| **Avg. AUC@10 ↑** | 44.03 | 54.05 | 51.61 | 60.41 | 50.40 | 53.34 | 54.61 | 53.73 | 56.05 | **61.64** |

encoder, it is dominated by UEC once epistemic uncertainty is taken into account at the ensemble level. Crucially, UEC does not trade off performance between language groups: it matches or improves upon the strongest baseline in German and French, and achieves the largest gains on Italian, the smallest and most imbalanced language split. This suggests that UEC remains robust under multilingual label imbalance and does not sacrifice minority-language performance when down-weighting uncertain embeddings.

**STS** On SEMREL24STS, UEC further improves over strong multilingual encoders on sentence-level similarity. While GTE-MB and E5 already achieve competitive performance on high-resource languages such as English and Afrikaans, UEC attains the best average score and ranks first on 9 out of 12 languages, including several low-resource African and Indic languages (e.g., Hausa, Hindi, Marathi). This indicates that uncertainty-driven ensembling not only preserves the strengths of specialist models but also yields more stable semantic similarity estimates in challenging multilingual and low-resource regimes.

Table 15: Comparison of multilingual legal judgment classification performance on SwissJudge-mentClassification. The best and second-best results per row are **bolded** and underlined. UEC achieves the highest average scores over German, French, and Italian, demonstrating robust performance across languages and label distributions. (Uni.=Uniform, Wt.=Weighted, TA=Task arithmetic.)

| Language / Metric | Individual Models | | | | Model Merging | | | Ensemble | | |
|---|---|---|---|---|---|---|---|---|---|---|
| | BGE | E5 | GTE | GTE-MB | Uni. | Wt. | TA | Uni. | Wt. | UEC |
| **German** | | | | | | | | | | |
| Accuracy | 50.28 | 47.85 | 47.73 | 53.08 | 48.26 | 49.73 | 48.08 | 49.14 | 51.75 | **53.13** |
| F1 | 43.86 | 41.96 | 43.01 | **46.10** | 42.67 | 44.61 | 44.43 | 43.24 | 44.61 | 45.97 |
| AUROC | 50.32 | 50.17 | 50.86 | 51.12 | 50.17 | 50.21 | 50.83 | 50.72 | 50.26 | **51.83** |
| **French** | | | | | | | | | | |
| Accuracy | 57.16 | **58.58** | 56.37 | 55.45 | 56.82 | 55.90 | 55.44 | 57.09 | 57.71 | 58.36 |
| F1 | 46.63 | 47.25 | 46.34 | 46.41 | 46.55 | 45.87 | 45.73 | 46.61 | 47.02 | **48.52** |
| AUROC | 50.48 | 50.22 | 50.65 | 50.67 | 50.13 | 50.24 | 50.37 | 50.53 | 50.51 | **51.02** |
| **Italian** | | | | | | | | | | |
| Accuracy | 53.47 | 57.61 | 53.00 | 58.36 | 54.76 | 55.89 | 54.83 | 54.92 | 57.65 | **58.72** |
| F1 | 45.64 | 46.09 | 45.72 | 47.79 | 45.88 | 46.03 | 45.31 | 45.62 | 46.72 | **48.03** |
| AUROC | 50.73 | 50.39 | 50.42 | 51.36 | 50.89 | 51.25 | 51.08 | 50.56 | 50.84 | **51.89** |
| **Avg. Accuracy ↑** | 53.64 | 54.68 | 52.37 | 55.63 | 53.28 | 53.84 | 52.79 | 53.72 | 55.65 | **56.74** |
| **Avg. F1 ↑** | 47.38 | 45.10 | 45.02 | 46.77 | 45.03 | 45.50 | 45.16 | 45.16 | 46.12 | **47.51** |
| **Avg. AUROC ↑** | 50.51 | 50.26 | 50.64 | 51.05 | 50.40 | 50.57 | 50.76 | 50.60 | 50.54 | **51.58** |

Table 16: Comparison of multilingual STS performance on SemRel24STS. The best and second-best results per column are **bolded** and underlined. UEC attains the highest average Spearman correlation, ranking first on 9 out of 12 languages and consistently outperforming both individual encoders and heuristic ensembles. We evaluate on 12 languages: Afrikaans (Afr.), Amharic (Amh.), Modern Standard Arabic (Arb.), Algerian Arabic (Arq.), Moroccan Arabic (Ary.), English (Eng.), Hausa (Hau.), Hindi (Hin.), Indonesian (Ind.), Kinyarwanda (Kin.), Marathi (Mar.), and Telugu (Tel.). (Uni.=Uniform, Wt.=Weighted, TA=Task arithmetic.).

| Model / Language | | Afr. | Amh. | Arb. | Arq. | Ary. | Eng. | Hau. | Hin. | Ind. | Kin. | Mar. | Tel. | Avg. |
|---|---|---|---|---|---|---|---|---|---|---|---|---|---|---|
| **Individual** | BGE | 72.13 | 12.28 | 30.88 | 37.80 | 22.93 | 79.52 | 33.69 | 48.91 | 46.99 | 45.26 | 55.00 | 27.43 | 42.73 |
| | E5 | 77.16 | 15.43 | 27.58 | 40.42 | 12.35 | 81.37 | 41.30 | 47.28 | 51.99 | 49.41 | 54.20 | 29.25 | 43.97 |
| | GTE | 77.29 | 15.21 | 22.01 | 34.19 | 12.88 | 77.79 | 35.25 | 40.69 | 45.20 | 47.33 | 45.66 | 32.36 | 40.48 |
| | GTE-MB | 77.03 | 18.95 | 28.42 | 42.34 | 31.32 | 75.02 | 43.70 | 24.11 | 42.71 | 49.95 | 44.30 | 35.81 | 42.81 |
| **Model Merging** | Uni. | 75.87 | 13.43 | 26.91 | 38.24 | 14.18 | 78.99 | 38.42 | 44.34 | 48.09 | 47.01 | 52.31 | 28.87 | 42.22 |
| | Wt. | 76.34 | 14.17 | 27.94 | 38.43 | 13.98 | 80.05 | 38.79 | 46.26 | 49.73 | 48.28 | 52.44 | 28.49 | 42.91 |
| | TA | 76.21 | 13.78 | 27.33 | 38.32 | 14.08 | 79.43 | 38.51 | 45.58 | 49.22 | 47.49 | 52.33 | 28.54 | 42.57 |
| **Ensemble** | Uni. | 75.79 | 13.82 | 27.18 | 38.19 | 14.25 | 79.16 | 38.19 | 44.85 | 48.34 | 46.91 | 52.42 | 28.99 | 42.34 |
| | Wt. | 76.46 | 15.12 | 28.06 | 39.18 | 13.83 | 80.86 | 39.44 | 46.52 | 50.93 | 48.88 | 53.02 | 28.26 | 43.38 |
| | UEC | **77.84** | 16.77 | **31.02** | 40.88 | 23.88 | **82.28** | **43.81** | **50.11** | **52.46** | 50.56 | **55.48** | 34.17 | **46.61** |

## C  Uncertainty Estimation Diagnosis

### C.1  Uncertainty Estimation of Post-hoc Probabilistic Embedding Model

We evaluate the behavior of the Laplace-based post-hoc probabilistic embedding model under several factors: covariance structure, the number of layers included in the curvature approximation, and the amount of data used for fitting. For all experiments, we fit LA following the same configuration in Appendix B.4, and evaluate uncertainty using a binary classification proxy task on MS MARCO.

**Covariance Structure**  We first compare diagonal and KFAC covariance structures applied to the same penultimate-layer parameters with E5 model. KFAC (Kronecker-Factored Approximate Curvature) approximates the full Fisher or Hessian matrix using a Kronecker factorization of layerwise curvature, enabling structured (second-order) uncertainty modeling that is substantially richer than a diagonal approximation. In principle, this allows correlations between parameters within the same linear layer to be captured, providing a more expressive posterior.

Table 17: Diagonal vs. KFAC covariance for LA. $m$, $d$, and $N$ denote penultimate-layer width, embedding dimension, and number of LA fitting samples, respectively. Diagonal covariance achieved near identical calibration performance without extra computational overhead compared to KFAC.

| Cov Type | ACC ↑ | ECE ↓ | Time Comp. | Mem. Comp. |
|---|---|---|---|---|
| **Diag** | 83.57 | 0.036 | $\mathcal{O}(Nmd)$ | $\mathcal{O}(md)$ |
| **KFAC** | 83.57 | 0.028 | $\mathcal{O}\big(N(m^2+d^2) + m^3+d^3\big)$ | $\mathcal{O}(m^2+d^2)$ |

Table 17 shows that both predictive accuracy and calibration (as measured by ECE) are nearly identical across the two settings. Despite capturing a higher-fidelity curvature structure, the KFAC variant does not yield measurable improvements in uncertainty quality for this setting while incurring significantly larger computational overhead. Given this trade-off, we adopt the diagonal covariance in the main experiments.

**Number of Layers** To examine whether deeper curvature modeling affects post-hoc uncertainty, we vary the number of layers included in the LA from only the final linear projection layer (the standard configuration) up to six layers.

Table 18: Effect of increasing the number of layers included in LA. Extending the curvature approximation to multiple layers yields negligible gains while significantly increasing computational cost.

| # Layers | 1 (Last) | 2 | 4 | 6 |
|---|---|---|---|---|
| **ACC ↑** | 83.57 | 83.57 | 83.56 | 83.57 |
| **ECE ↓** | 0.036 | 0.035 | 0.038 | 0.035 |

As shown in Table 18, both accuracy and ECE remain stable across all configurations. This behavior aligns with recent findings in Daxberger et al. (2021) and Sharma et al. (2023), which show that last-layer LA already provide strong and computationally efficient uncertainty estimates for large pretrained encoders. Our observations support this conclusion: extending the curvature approximation to multiple layers yields negligible gains while significantly increasing computational cost.

**Data Sparsity** We further examine how the amount of data used for fitting the Laplace Approximation affects uncertainty estimation. We subsample the fitting set to $50\%$, $10\%$, and $1\%$ of the original 7.5K samples, and additionally include a $0\%$ **condition** corresponding to purely deterministic embeddings without LA to isolate the contribution of the posterior variance.

Table 19: Effect of Laplace fitting data sparsity on uncertainty estimation. Calibration error increases gradually under extreme sparsity, but remains well-behaved for moderate subsampling.

| Sparsity | 100% (Original) | 50% | 10% | 1% | 0% (Det.) |
|---|---|---|---|---|---|
| **ACC ↑** | 83.57 | 83.57 | 83.56 | 83.56 | 83.57 |
| **ECE ↓** | 0.036 | 0.038 | 0.051 | 0.061 | 0.063 |

Table 19 shows that accuracy is unaffected by data sparsity, confirming that LA does not harm predictive quality. More importantly, all LA–based settings outperform the deterministic baseline (0%) in calibration. Even with only $1\%$ of the fitting data ($\sim$75 samples), LA yields mild but consistent calibration gains. With $10\%$ of the data ($\sim$750 samples), the calibration error nearly matches the full-data setting, indicating that LA requires only a modest amount of data to produce stable variance estimates. Overall, these results demonstrate that LA is not particularly sensitive to noise or sparsity, and that uncertainty estimation remains reliable even under substantial subsampling.

## C.2 Variance Calibration of $\sigma_s^2$

To directly assess the quality of $\sigma_s^2$ as a variance estimator, we perform a variance–error calibration analysis, as mentioned in Section 4.3.2.

**Setup**   Standard calibration metrics such as ECE are designed for probability outputs and are not directly applicable to variance-based confidence measures such as $\sigma_s^2$. To address this limitation, we introduce Var-ECE, a variance-oriented calibration metric that evaluates whether lower predicted variance corresponds to higher empirical correctness. For each sentence pair $(x_1, x_2)$ in SEMREL24STS, a human similarity score $y \in [0, 1]$ is provided. Under UEC, the similarity is modeled as

$$s(x_1, x_2) \sim \mathcal{N}\big(\mu_s(x_1, x_2), \, \sigma_s^2(x_1, x_2)\big),$$

where $\mu_s$ is the predicted mean similarity and $\sigma_s^2$ is the epistemic variance. If $\sigma_s^2$ is an accurate estimate of predictive variance, then examples with large predicted variance should on average incur larger squared error. We therefore compute the empirical squared error

$$r(x_1, x_2) \;:=\; \big(\mu_s(x_1, x_2) - y(x_1, x_2)\big)^2,$$

and directly compare $\sigma_s^2$ and $r$ across groups of examples.

**Variance–Error Binning**   To reduce noise and obtain stable estimates, we partition the predicted variances into $M$ quantile bins $\{B_m\}_{m=1}^M$ based on $\sigma_s^2(x_1, x_2)$ (we use $M = 10$ in all experiments). For each bin, we compute the average predicted variance and the average squared error:

$$\overline{\sigma^2}_m = \frac{1}{|B_m|} \sum_{(x_1, x_2) \in B_m} \sigma_s^2(x_1, x_2), \qquad \overline{r}_m = \frac{1}{|B_m|} \sum_{(x_1, x_2) \in B_m} r(x_1, x_2).$$

If $\sigma_s^2$ is well estimated, then $\overline{\sigma^2}_m$ should be close to $\overline{r}_m$ for every bin: examples assigned higher predicted variance should indeed exhibit higher empirical error.

**Variance Calibration Error (Var-ECE)**   We measure the discrepancy between predicted and empirical variances via

$$\text{Var-ECE} \;=\; \sum_{m=1}^M \frac{|B_m|}{N} \left| \overline{r}_m - \overline{\sigma^2}_m \right|, \tag{31}$$

where $N$ is the total number of evaluated examples. A small Var-ECE indicates that the predicted variance $\sigma_s^2$ numerically matches the observed squared error across uncertainty levels, meaning the UEC provides an accurate estimate of predictive variance.

## D   SENSITIVITY TO $\beta$ AND TEMPERATURE $T$

We analyze the sensitivity of UEC to two key hyperparameters: (1) the coefficient $\beta$ that determines how strongly the estimated variance influences the uncertainty-aware similarity, and (2) the temperature parameter $T$ that sharpens the ensemble coefficients.

**MIRACL Subset**   Figure 6 summarizes the effect of varying $\beta$ and $T$ on MIRACL Subset. While the curves exhibit noticeable movement, UEC maintains stable behavior across a broad range of values. This reflects that both hyperparameters meaningfully control how uncertainty modulates similarity and coefficient sharpening. Performance consistently peaks near $\beta = 10^{-2}$ and $T = 1.5$, which provides a balanced trade-off across all retrieval metrics.

**MMTEB**   A similar analysis on the StackOverflow task (MMTEB) is presented in Figure 7. Remarkably, the optimal region remains highly consistent with the MIRACL case: the best performance again emerges around $\beta = 10^{-2}$ and requires only mild temperature sharpening (here, $T \approx 1.8$).

## E   STATEMENT ON THE USE OF LARGE LANGUAGE MODELS

In this study, we use LLMs in a limited, supporting capacity. The usage was restricted to polish writing, specifically for grammatical corrections, sentence refinement, and improving overall writing consistency, and experimental codes.

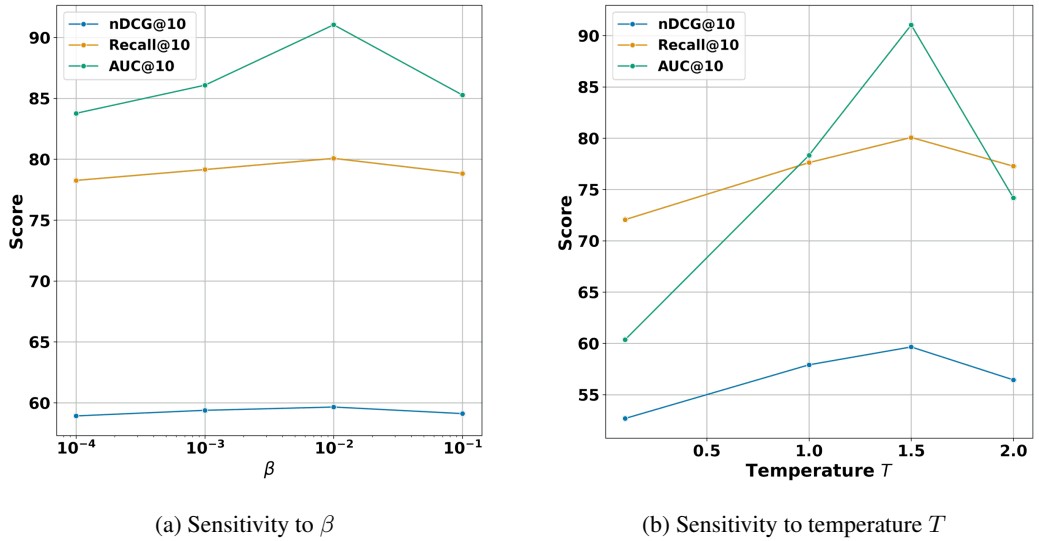

(a) Sensitivity to $\beta$

(b) Sensitivity to temperature $T$

Figure 6: Sensitivity analysis of UEC on MIRACL. Both $\beta$ and $T$ influence performance in interpretable ways controlling variance contribution and sharpening, respectively. UEC exhibits a stable optimum region around $(\beta = 10^{-2}, T = 1.5)$, making tuning straightforward in practice.

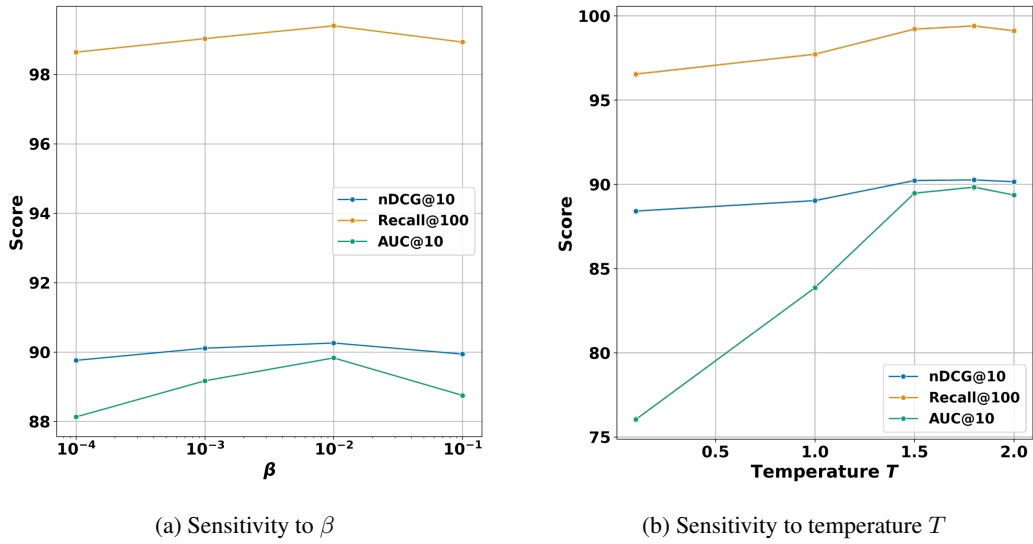

(a) Sensitivity to $\beta$

(b) Sensitivity to temperature $T$

Figure 7: Sensitivity analysis of UEC on StackOverflow (MMTEB). Despite dataset differences, the optimal hyperparameter region closely matches that of MIRACL, indicating that UEC's hyperparameter tuning is stable and transferable across tasks.

