# OpenReview forum: "Uncertainty-driven Embedding Convolution"
_ICLR.cc/2026/Conference — ICLR 2026 Poster_

### Official Review · Reviewer_ivFt · 2025-10-22

**Soundness:** 3
**Presentation:** 3
**Contribution:** 3
**Rating:** 8
**Confidence:** 2

**Summary:**

This paper addresses the limitation of deterministic embedding ensembles, which fail to account for model-specific uncertainty, by proposing the Uncertainty-driven Embedding Convolution (UEC) method. UEC first converts pre-trained deterministic embeddings into probabilistic representations via a post-hoc Laplace approximation to quantify their uncertainty. It then derives Bayes-optimal ensemble weights that adaptively favor more confident models through a probabilistic contrastive loss. Based on this, the authors introduce an uncertainty-aware similarity function. Extensive experiments demonstrate that UEC consistently enhances performance and robustness across diverse NLP tasks by leveraging principled uncertainty modeling. The work provides a theoretically grounded and practical method for creating more reliable embeddings through ensembles.

**Strengths:**

1. The idea of moving from deterministic to probabilistic embeddings for ensemble embedding learning is interesting and well-motivated. It allows quantifying model-specific uncertainty, thus improving robustness. The "jaguar" example effectively illustrates the need for this method.

2. The paper provides strong theoretical foundations. Deriving the ensemble weights as a Bayes-optimal solution under a surrogate loss and connecting the similarity function to the 2-Wasserstein distance with a bounded error are significant contributions that add rigor and credibility to the proposed method.

3. Extensive experiments have demonstrated the effectiveness of UEC, covering multiple tasks (Retrieval, Classification, STS, Bitext Mining, Clustering, Reranking) across diverse benchmarks (MIRACL, MMTEB). The consistent outperformance of UEC over strong baselines, including model merging and other ensemble techniques, provides robust evidence for its effectiveness.

**Weaknesses:**

1. The paper is very dense, especially Section 3. The derivations, while valuable, could be summarized more intuitively in the main text, with full details reserved for the appendix. Besides, the transition from the loss function in Eq. (2) to the tractable approximation in Eq. (3) is somewhat abrupt.
2. There are certain hyperparameters, such as temperature and β. However, there is no analysis of UEC's sensitivity to the choices of these hyperparameters.
3. While this ensemble method combines the strengths of specialists, i.e., different embedding models, LLMs perform like generalists. A comparison with LLM’s embeddings could justify the need for the ensemble method in practice.

**Questions:**

1. Would a general model, such as an LLM, be sufficient to handle the diverse tasks robustly? How would the proposed ensemble method perform compared to LLMs when applied to different tasks?
2. How would the hyperparameters affect the performance of UEC?
3. In the poem classification task, UEC underperforms an individual model GTE-MB, when it ensembles three weaker models. Can the ensemble apply to stronger models, such as GTE-MB, and still yield performance gains?
4. How well could the proposed method estimate the mean and variance for the individual embeddings and the ensembled one, under different data sparsity settings? Are the estimations sensitive to noise?

---

> ### Author Response · Authors · 2025-11-21
>
> We thank Reviewer ivFt for providing constructive feedback on clarity, hyperparameter sensitivity, and the relationship between UEC and generalist LLM embeddings.
>
> Key concerns identified by the reviewer:
> - Section 3 is dense, and derivations should be summarized more intuitively
> - Abrupt transition between Eq. (2) and Eq. (3)
> - No hyperparameter sensitivity analysis for $\beta$ and temperature
> - Question about whether generalist LLM embeddings may be sufficient
> - Performance elevation when ensembling stronger models
> - Questions about robustness of mean/variance estimation under data sparsity
>
> Your suggestions motivated the restructuring and simplification of Section 3, the addition of sensitivity studies, and clearer comparisons with strong baselines. We appreciate the constructive guidance, which helped improve both the readability and practical relevance of the paper.
>
> We address all points in detail in the subsequent responses.

---

> > ### Author Response · Authors · 2025-11-21
> > **[W1] Improving Readability and Clarifying the Transition from Eq. (2) → Eq. (3)**
> >
> > > **__W1)__** The paper is very dense, especially Section 3. The derivations, while valuable, could be summarized more intuitively in the main text, with full details reserved for the appendix. Besides, the transition from the loss function in Eq. (2) to the tractable approximation in Eq. (3) is somewhat abrupt.
> >
> > Thank you for this valuable suggestion. We fully agree that the original version of Section 3 was dense, and we appreciate the reviewer’s feedback pointing out where additional clarity was needed. We acknowledge that the original submission was overly dense, particularly because we attempted to preserve mathematical rigor in the main text. This unfortunately made the derivations less accessible.
> >
> > To make the paper clear, we revised manuscript like:
> > - We now provide a high-level explanation of the Laplace Approximation in the main text while moving the full mathematical derivation to Appendix A.1 for clarity.
> > - We added a clear justification for retaining only the query-dependent terms when approximating the surrogate loss, as shown in the revised passage cited below.
> >     > **__(Section 3.2, L215~221)__** This surrogate loss decomposes into fidelity and epistemic uncertainty. While $\mathbf{\mu}_k(\mathbf{x})$ and $\mathbf{\Sigma}_k(\mathbf{x})$ are available at retrieval time, computing $\mathbf{\mu}_k(\mathbf{x}')$ and $\mathbf{\Sigma}_k(\mathbf{x}')$ requires evaluating the embedding model on every document. However, in practical retrieval systems, document embeddings are precomputed once and stored in the index, whereas the query $\mathbf{x}$ arrives only at inference time (Zhou et al., 2022; Zhao et al., 2024). Since the surrogate loss in Eq.2 contains terms that depend on $\mathbf{x}'$, which cannot be recomputed per query, we discard all document-dependent terms and retain only the query-dependent components. This yields a retrieval-feasible approximation of Eq.2, leading directly to the tractable convex optimization:
> >
> > - Throughout the derivation, we inserted additional intuition at key steps to improve readability and guide the reader through the transition from Eq. 2 to Eq. 3, particularly in Sections 3.2 (L215–221, L226–233, L237–246), as reflected in the following excerpts.
> >     > **__(Section 3.2, L226~233)__** This convex linear objective combines variance and embedding magnitude into a tractable surrogate loss. Although the objective is linear, the probability simplex constraints yield a well-posed convex program with a unique closed-form solution under standard convex analysis (Boyd & Vandenberghe, 2004). This ensures that the resulting coefficients are well-defined and stable, avoiding degenerate behavior even under heterogeneous uncertainty across models. The solution therefore yields a soft inverse–risk weighting that distributes probability mass smoothly, rather than collapsing entirely onto the smallest term, leading to a more robust and interpretable ensemble. Its unique closed-form solution is the inverse-variance weighting:
> >
> >     > **__(Section 3.2, L237~246)__** Intuitively, the inverse–variance rule assigns lower weight to embeddings whose large epistemic variance signals low confidence, while giving more influence to models that produce stable, self-consistent representations for the given query. This precision-based structure mirrors classical Bayesian aggregation, where estimators with higher certainty contribute proportionally more to the final prediction (Singh et al., 2005). Importantly, this mechanism also adapts to data-wise heterogeneity: for queries that fall into domains where certain encoders are weak or out-of-distribution, their variances naturally increase, causing UEC to down-weight them without requiring explicit domain labels. Under this interpretation, the resulting coefficient vector $π^∗_k(x)$ is precisely the minimizer of the uncertainty-aware surrogate loss in Eq. 3, representing the Bayes-optimal solution under the surrogate formulation. Full derivation appears in Appendix A.2.
> >
> >
> > We sincerely appreciate this comment. It helped us re-examine the organization and clarity of Section 3, leading to a significantly improved presentation. Your suggestion directly contributed to making the manuscript more accessible and better readability.

---

> > > ### Author Response · Authors · 2025-11-21
> > > **[W2, Q2] Sensitivity of UEC to Temperature and beta (Hyperparameter Analysis)**
> > >
> > > > **__W2)__** There are certain hyperparameters, such as temperature and $\beta$. However, there is no analysis of UEC's sensitivity to the choices of these hyperparameters.
> > >
> > > > **__Q2)__** How would the hyperparameters affect the performance of UEC?
> > >
> > > Thank you for highlighting this important aspect. You are absolutely right that analyzing sensitivity is crucial for understanding practical applicability.
> > >
> > >
> > > - We conducted new sensitivity experiments about (1) $\beta$ (controls how strongly the predicted variance contributes to the uncertainty-aware similarity) and (2) temperature $T$ (sharpens the ensemble coefficients) on:
> > >     - MIRACL Subset (Appendix D, Figure 6)
> > >         - $\beta$
> > >             | Metric / $\beta$ | 1e-4 | 1e-3 | 1e-2 | 1e-1
> > >             | --- | --- | --- |  --- | --- |
> > >             | nDCG@10 | 58.92 | 59.38 | **59.65** | 59.11  |
> > >             | Recall@10 | 78.26 | 79.15 | **80.07** | 78.83|
> > >             | AUC@10 | 83.77 | 86.08 | **91.04** | 85.27  |
> > >         - Temperature $T$
> > >             | Metric / $T$ | 0.1 | 1 | 1.5 | 2
> > >             | --- | --- | --- |  --- | --- |
> > >             | nDCG@10 | 52.68 | 57.91 | **59.65** | 56.44  |
> > >             | Recall@10 | 72.06 | 77.62 | **80.07** | 77.27|
> > >             | AUC@10 | 60.36 | 78.34 | **91.04** | 74.18  |
> > >
> > >     - MMTEB (StackOverflowQA)  (Appendix D, Figure 7)
> > >         - $\beta$
> > >             | Metric / $\beta$ | 1e-4 | 1e-3 | 1e-2 | 1e-1
> > >             | --- | --- | --- |  --- | --- |
> > >             | nDCG@10 | 89.76 | 90.11 | **90.26** | 89.94  |
> > >             | Recall@100 | 98.64 | 99.03 | **99.40** | 98.93|
> > >             | AUC@10 | 88.13 | 89.17 | **89.83** | 88.75  |
> > >         - Temperature $T$
> > >             | Metric / $T$ | 0.1 | 1 | 1.5 | 1.8 | 2
> > >             | --- | --- | --- |  --- | --- | --- |
> > >             | nDCG@10 | 88.41 | 89.03 | 90.22 | **90.26**  | 90.15 |
> > >             | Recall@100 | 96.54 | 97.72 | 99.21 | **99.40** | 99.11|
> > >             | AUC@10 | 76.04 | 83.86 | 89.47 | **89.83**  | 89.36 |
> > >
> > > - From the experimental results, we obtained the following observations:
> > >     -  Both hyperparameters meaningfully influence performance.
> > >     - Temperature $T$ has larger impact than $\beta$.
> > >     - Nevertheless, **the optimal ranges remain highly consistent across datasets, indicating that the practical hyperparameter search space is reasonably small.**
> > >
> > >
> > > Your comment helped us strengthen the empirical evaluation, and we appreciate your guidance in highlighting an important missing component.

---

> > > > ### Author Response · Authors · 2025-11-21
> > > > **[W3, Q1] Why an Embedding Ensemble Is Still Needed Despite Strong Generalist LLM Embeddings**
> > > >
> > > > > **__W3)__** While this ensemble method combines the strengths of specialists, i.e., different embedding models, LLMs perform like generalists. A comparison with LLM’s embeddings could justify the need for the ensemble method in practice.
> > > >
> > > > > **__Q1)__** Would a general model, such as an LLM, be sufficient to handle the diverse tasks robustly? How would the proposed ensemble method perform compared to LLMs when applied to different tasks?
> > > >
> > > > Thank you for this insightful question. We appreciate the opportunity to clarify the relationship between specialist models, generalist LLM embeddings, and the role of UEC.
> > > >
> > > >
> > > > - We believe the content addressed in the question is already reflected in the experiments we conducted:
> > > >     - MIRACL Subset (Section 4.1, Figure 3) evaluates specialist multilingual models.
> > > >     - MMTEB (Section 4.2, Table 1-3 / Appendix B.4.5 Table 11-13, Appendix B.5 Table14-16) evaluates general-domain English models—including several trained on large-scale corpora, analogous to “generalist” embedding sources.
> > > >     - Across both scenarios, **UEC consistently improves over using a general embedding alone, demonstrating that UEC remains beneficial even when strong general models are available.**
> > > >
> > > > - Additional multilingual results (Appendix B.5, Table 14-16) further confirm that UEC improves robustness across languages and tasks. (Due to character limitations, we report only the average performance here. Please refer to the revised version for full per-language results.)
> > > >     - **__Table 14)__** Comparison of multilingual **retrieval** performance on the WIKIPEDIARETRIEVALMULTILINGUAL
> > > >
> > > >         | Model Type | BGE | E5 | GTE | GTE-MB | Uni. | Wt. | TA | Uni. (Ens.) | Wt. (Ens.) | **UEC** |
> > > >         |------------|-----|-----|------|--------|-------|-------|------|--------------|--------------|-----------|
> > > >         | **nDCG@10 ↑** | 32.18 | 41.07 | 33.47 | _43.04_ | 35.36 | 39.15 | 38.61 | 35.71 | 39.97 | **43.24** |
> > > >         | **Recall@100 ↑** | 52.03 | 61.86 | 51.95 | _63.52_ | 56.37 | 59.72 | 59.55 | 57.28 | 61.67 | **64.51** |
> > > >         | **AUC@10 ↑** | 44.03 | 54.05 | 51.61 | _60.41_ | 50.40 | 53.34 | 54.61 | 53.73 | 56.05 | **61.64** |
> > > >
> > > >
> > > >     - **__Table 15)__** Comparison of multilingual legal judgment **classification** performance on SWISSJUDGEMENTCLASSIFICATION.
> > > >
> > > >         | Model Type | BGE | E5 | GTE | GTE-MB | Uni. | Wt. | TA | Uni. (Ens.) | Wt. (Ens.) | **UEC** |
> > > >         |------------|------|------|------|--------|--------|--------|------|--------------|--------------|------------|
> > > >         | **Accuracy ↑** | 53.64 | 54.68 | 52.37 | 55.63 | 53.28 | 53.84 | 52.79 | 53.72 | _55.65_ | **56.74** |
> > > >         | **F1 ↑** | 47.38 | 45.10 | 45.02 | _46.77_ | 45.03 | 45.50 | 45.16 | 45.16 | 46.12 | **47.51** |
> > > >         | **AUROC ↑** | 50.51 | 50.26 | 50.64 | _51.05_ | 50.40 | 50.57 | 50.76 | 50.60 | 50.54 | **51.58** |
> > > >
> > > >
> > > >     - **__Table 16)__** Comparison of multilingual **STS** performance on SEMREL24STS.
> > > >
> > > >         | Model Type | BGE | E5 | GTE | GTE-MB | Uni. | Wt. | TA | Uni. (Ens.) | Wt. (Ens.) | **UEC** |
> > > >         |------------|------|------|------|--------|--------|--------|------|--------------|--------------|------------|
> > > >         | **Spearman ↑** | 42.73 | _43.97_ | 40.48 | 42.81 | 42.22 | 42.91 | 42.57 | 42.34 | 43.38 | **46.61** |
> > > >
> > > >     - Across diverse tasks, UEC achieves best performance among baselines, indicating **UEC remains robust under multilingual.**
> > > >
> > > >
> > > > We thank the reviewer for prompting a more comprehensive explanation and for helping us better communicate UEC’s value in mixed specialist–generalist settings.

---

> > > > > ### Author Response · Authors · 2025-11-21
> > > > > **[Q3] Applicability of UEC to Strong Specialist Models**
> > > > >
> > > > > > **__Q3)__** In the poem classification task, UEC underperforms an individual model GTE-MB, when it ensembles three weaker models. Can the ensemble apply to stronger models, such as GTE-MB, and still yield performance gains?
> > > > >
> > > > > This was an excellent suggestion to test ours, and we are grateful for it.
> > > > >
> > > > > Following the reviewer’s recommendation, we conducted an additional experiment where UEC ensembles: BGE, E5, and GTE-MB (a stronger model) in PoemSentimentClassification dataset.
> > > > > - As shown in the table below, **incorporating GTE-MB significantly boosts UEC performance**:
> > > > >
> > > > >     | Model | Accuracy $\uparrow$  | F1 $\uparrow$ | AUROC $\uparrow$  |
> > > > >     | ---- | --- | --- | --- |
> > > > >     | BGE | 51.92     | 40.76     | 56.70     |
> > > > >     | E5 | 53.07     | 41.43     | 57.38     |
> > > > >     | GTE | 47.88     | 37.53     | 53.41     |
> > > > >     | GTE-MB | 56.92     | 45.53     | 62.21     |
> > > > >     | UEC (BGE, E5, GTE) | 56.81     | 44.46     | 61.39     |
> > > > >     | **UEC (BGE, E5, GTE-MB)** | **58.28** | **46.79** | **64.02** |
> > > > >
> > > > > - These results confirm that **UEC benefits from stronger constituent models**, addressing the reviewer’s question directly.
> > > > >
> > > > > Thank you for helping strengthen the experimental section with this important validation.

---

> > > > > > ### Author Response · Authors · 2025-11-21
> > > > > > **[Q4] Robustness of Mean/Variance Estimation Under Data Sparsity and Noise**
> > > > > >
> > > > > > > **__Q4)__** How well could the proposed method estimate the mean and variance for the individual embeddings and the ensembled one, under different data sparsity settings? Are the estimations sensitive to noise?
> > > > > >
> > > > > > Thank you for raising this valuable question. It led us to perform additional analyses on calibration and sparsity robustness.
> > > > > >
> > > > > > We added following experiments:
> > > > > >
> > > > > > - **Calibration diagnostics** (Section 4.3.1, Table 4):
> > > > > >     - We evaluated ECE for each model before and after applying LA.
> > > > > >          |  | **E5 (Deter.)** | **E5 (Prob.)** | **BGE (Deter.)** | **BGE (Prob.)** | **GTE (Deter.)** | **GTE (Prob.)** | **Ensemble (Deter.)** | **Ensemble (Prob.)** |
> > > > > >         | ------------------ | ------------- | ------------ | -------------- | ------------- | -------------- | ------------- | ------------------- | ------------------ |
> > > > > >         | **ACC ↑**          | 83.57         | 83.57        | 84.19          | 84.19         | 82.74          | 82.74         | 84.83               | **85.41**          |
> > > > > >         | **ECE ↓**          | 0.063         | **0.036**    | 0.067          | **0.042**     | 0.077          | **0.043**     | 0.075               | **0.032**          |
> > > > > >         - We fit LA following the same configuration in Appendix B.4 (Experimental details of MMTEB), and evaluate uncertainty using a binary classification proxy task on MS MARCO.
> > > > > >         - Deter.” denotes the deterministic embedding, while “Prob.” denotes the probabilistic embedding derived via LA. The deterministic ensemble corresponds to the weighted ensemble baseline, and the probabilistic ensemble corresponds to UEC (Ours).
> > > > > >
> > > > > >     - Across all models, we show that **LA improves calibration, and UEC achieves the strongest accuracy and calibration performance.**
> > > > > >
> > > > > > - **Data sparsity experiment** (Appendix C.1, Table 19):
> > > > > >     - LA fitted with subsets of only 50%, 10%, 1% of the data.
> > > > > >
> > > > > >         | **Sparsity** | **100% (Original)** | **50%** | **10%** | **1%** | **0% (Det.)** |
> > > > > >         | ------------ | ------------------- | ------- | ------- | ------ | ------------- |
> > > > > >         | **ACC ↑**    | 83.57               | 83.57   | 83.56   | 83.56  | 83.57         |
> > > > > >         | **ECE ↓**    | 0.036               | 0.038   | 0.051   | 0.061  | 0.063         |
> > > > > >
> > > > > >
> > > > > >     - Even with ~750 samples (10%), calibration remains stable.
> > > > > >     - This indicates LA is robust even under severe data limitations.
> > > > > >
> > > > > >
> > > > > > - **Noise / distribution shift robustness**:
> > > > > >     - Prior work Laplace-Redux [1] shows LA is more robust to distribution shift than alternatives like MAP or HMC.
> > > > > >     - Our own pipeline fits LA on MS MARCO + SNLI in MMTEB experiments (Section 4.2), yet performs strongly on many different MMTEB domains—consistent with shift robustness.
> > > > > >
> > > > > >
> > > > > > We genuinely appreciate this question. It helped us examine the reliability of UEC under realistic constraints such as limited data and distributional mismatch.
> > > > > >
> > > > > > ---
> > > > > > **References**
> > > > > >
> > > > > > [1] Daxberger, Erik, et al. "Laplace redux-effortless bayesian deep learning." Advances in neural information processing systems 34 (2021): 20089-20103.

---

> > > > > > > ### Comment · Reviewer_ivFt · 2025-11-26
> > > > > > >
> > > > > > > Thank you for your clarification. Your responses have addressed my concerns.

---

> > > > > > > > ### Author Response · Authors · 2025-11-26
> > > > > > > >
> > > > > > > > We appreciate for your thoughtful comments again.
> > > > > > > > We sincerely appreciate the constructive feedback you provided throughout the review process. Thanks to your detailed observations and suggestions, we were able to identify and address key areas that needed clarification, which has greatly strengthened the completeness and overall quality of the manuscript.
> > > > > > > >
> > > > > > > > We are grateful for the time and care you dedicated to reviewing our work.

---

### Official Review · Reviewer_jJ7t · 2025-10-25

**Soundness:** 3
**Presentation:** 3
**Contribution:** 2
**Rating:** 4
**Confidence:** 3

**Summary:**

This paper proposes Uncertainty-driven Embedding Convolution (UEC), which ensembles pre-trained text-embedding models by converting their deterministic embeddings into Gaussian representations via a post-hoc Laplace approximation, weighting and combining them by convolution, and using a probabilistic similarity metric that approximates distributional distance. The resulting framework improves robustness and retrieval accuracy across multilingual and multi-domain tasks, while remaining computationally lightweight. Most of the methods are very solid with many inspiration taken from applied statists literature.

**Strengths:**

1. The method is conceptually clean and well grounded from existing literature.

2. The paper is clearly written and mostly mathematically correct, although with some times assumptions not specifically defined.

3. The empirical results are convincing, the authors covering retrieval, classification, and semantic textual similarity.

4. The approach is practical, post-hoc, scalable, and effective without model retraining.

**Weaknesses:**

1. The core components are standard statistical techniques: Laplace approximation inverse-variance weighting for Gaussian combination, and uncertainty-aware similarity based on probit approximations. Hence, I think the paper’s novelty lies mostly in assembling these ingredients into a single coherent pipeline for embedding ensembles, not in introducing fundamentally new theory.

2. The “Bayes-optimal” claim is slightly overstated given that the surrogate loss drops query-dependent terms (Eq. 3).

3. It would be valuable to analyse uncertainty calibration (ECE, reliability diagrams) to confirm that the modellled variances truly reflect epistemic uncertainty rather than functioning as learned scaling factors.

4. Why not included some other baselines (e.g., Bayesian model averaging or deep ensemble uncertainty fusion)?

**Questions:**

Please see my questions from weakness section.

---

> ### Author Response · Authors · 2025-11-21
>
> We thank Reviewer jJ7t for the clear and thoughtful assessment of the paper’s contributions and positioning within existing literature.
>
> Key concerns identified by the reviewer:
> - Novelty primarily lies in integrating established components rather than creating new theory
> - “Bayes-optimal” phrasing may overstate the strength of the claim
> - Need for uncertainty calibration diagnostics
> - Desire for stronger or additional baselines (e.g., Bayesian model averaging)
>
> Your comments helped refine how we frame the contribution and clarify theoretical claims. We also expanded the calibration analysis and revisited the baseline selection to address your suggestions. Thank you for helping us present a clearer and more appropriately scoped contribution.
>
> A detailed point-by-point response is provided below.

---

> > ### Author Response · Authors · 2025-11-21
> > **[W1] Clarifying Novelty Beyond Classical Statistical Components**
> >
> > > **__W1)__** The core components are standard statistical techniques: Laplace approximation inverse-variance weighting for Gaussian combination, and uncertainty-aware similarity based on probit approximations. Hence, I think the paper’s novelty lies mostly in assembling these ingredients into a single coherent pipeline for embedding ensembles, not in introducing fundamentally new theory.
> >
> > Thank you very much for this thoughtful observation. We fully agree that several individual components used in our method—Laplace approximation, inverse-variance weighting, and probit approximations—are classical statistical tools. We appreciate the reviewer highlighting this, as it allowed us to better articulate where the substantive novelty of our work lies.
> >
> > Our key contributions can be summarized as follows:
> > 1) **First work to construct a post-hoc probabilistic *embedding* ensemble**
> >     - Prior work in embeddings has explored probabilistic embeddings (e.g., modeling uncertainty within a single encoder), but there is no prior method that: converts pre-trained deterministic embedding models into a probabilistic form without retraining, and then leverages the resulting parameter-level uncertainty to perform principled ensemble fusion.
> >     - Existing ensemble approaches for embeddings generally rely on heuristic weighted averages, uniform averaging, or model merging, but none propagate uncertainty or provide a Bayesian formulation.
> >     - To our knowledge, **UEC is the first method that produces a principled, uncertainty-aware embedding ensemble in a fully post-hoc manner, making it directly applicable to large existing libraries of pretrained encoders.**
> >
> > 2) **Novel surrogate-loss formulation enabling Bayes-optimal ensemble weights**
> >     - While inverse-variance weighting is classical in regression, its extension to embedding similarity and retrieval is far from straightforward. Our work introduces:
> >         - **a retrieval-specific surrogate loss**, derived from the probabilistic similarity objective,
> >         - and shows that its **minimizer yields a Bayes-optimal coefficient (under the surrogate loss)** tailored to embedding ensembles.
> >     - This formulation is new in the embedding domain, where ensemble methods have lacked theoretical grounding. The derivation required adapting Bayesian weighting ideas to the structure of embedding similarities—a setting fundamentally different from regression or density estimation.
> >
> > 3) **First uncertainty-aware similarity formulation linked to a Wasserstein bound**
> >     - Another key novelty is the treatment of similarity computation, which is central to every embedding application. We introduce:
> >         - an **uncertainty-aware similarity** derived from marginalizing over Gaussian embedding posteriors,
> >         - and show that its **probit-based form is a bounded approximation to the Wasserstein/Jeffreys-based similarity between distributions.**
> >     - To our knowledge, **this is the first work to connect probabilistic embeddings, uncertainty propagation, similarity computation, and Wasserstein-bounded approximations in a retrievable and computationally light manner.**
> >
> >
> > We appreciate the reviewer’s feedback for giving us the opportunity to clarify and emphasize these contributions more clearly.

---

> > > ### Author Response · Authors · 2025-11-21
> > > **[W2] Clarifying the Scope of the Bayes-optimal Claim Under the Surrogate Loss**
> > >
> > > > **__W2)__** The “Bayes-optimal” claim is slightly overstated given that the surrogate loss drops query-dependent terms (Eq. 3).
> > >
> > > We sincerely appreciate the reviewer’s thoughtful point regarding the “Bayes-optimal” claim and the removal of the query-dependent terms in the surrogate loss. This comment helped us refine the theoretical presentation and clarify the assumptions underlying the derivation.
> > >
> > > - First, we fully acknowledge that Eq. (3) omits the query-dependent components of Eq. (2). As the reviewer correctly noted, this means that the surrogate loss is defined under a practical approximation. However, the explanation in the original submission did not sufficiently clarify why this approximation is both necessary and standard in retrieval settings. This approximation is fully consistent with the constraints of real retrieval systems, where all document embeddings usually be precomputed and stored independently of any incoming query. To make this clear, we revised the manuscript to explicitly explain that terms involving the document side cannot be recomputed per query and must therefore be removed, leaving only the query-dependent components, as illustrated in the following revised passage:
> > >     > **__(Section 3.2, L215~221)__** This surrogate loss decomposes into fidelity and epistemic uncertainty. While $\mathbf{\mu}_k(\mathbf{x})$ and $\mathbf{\Sigma}_k(\mathbf{x})$ are available at retrieval time, computing $\mathbf{\mu}_k(\mathbf{x}')$ and $\mathbf{\Sigma}_k(\mathbf{x}')$ requires evaluating the embedding model on every document. However, in practical retrieval systems, document embeddings are precomputed once and stored in the index, whereas the query $\mathbf{x}$ arrives only at inference time (Zhou et al., 2022; Zhao et al., 2024). Since the surrogate loss in Eq.2 contains terms that depend on $\mathbf{x}'$, which cannot be recomputed per query, we discard all document-dependent terms and retain only the query-dependent components. This yields a retrieval-feasible approximation of Eq.2, leading directly to the tractable convex optimization:
> > >
> > >
> > > - Second, following the reviewer’s suggestion, we softened and clarified the “Bayes-optimal” claim to the more precise description:
> > >     > **__(Section 3.2, L195-197)__** We then derive a convex optimization problem that identifies the coefficients yielding a principled inverse-risk weighting under this surrogate, which provides the Bayes-optimal solution for this surrogate formulation.
> > >
> > >     > **__(Section 3.2, L226-227)__** This convex linear objective combines variance and embedding magnitude into a tractable surrogate
> > >     loss.
> > >
> > >     > **__(Section 3.2, L243-246)__** nder this interpretation, the resulting coefficient vector $\pi^*_k (x)$ is precisely the
> > >     minimizer of the uncertainty-aware surrogate loss in Eq. 3, representing the Bayes-optimal solution under the surrogate formulation.
> > >
> > >
> > > We thank the reviewer again for providing an opportunity to sharpen the theoretical exposition. We hope these improvements significantly strengthened the readability and rigor of Section 3.

---

> > > > ### Author Response · Authors · 2025-11-21
> > > > **[W3] Evaluating Whether the Modeled Variances Reflect True Epistemic Uncertainty**
> > > >
> > > > > **__W3)__** It would be valuable to analyse uncertainty calibration (ECE, reliability diagrams) to confirm that the modelled variances truly reflect epistemic uncertainty rather than functioning as learned scaling factors.
> > > >
> > > > We sincerely thank the reviewer for this thoughtful and important suggestion. Your comment encouraged us to strengthen the empirical analysis of uncertainty quality and directly examining whether the modeled variances meaningfully correspond to epistemic uncertainty.
> > > >
> > > > To address this, we have added a comprehensive calibration analysis in Section 4.3 of the revised version, covering both the per-model probabilistic embeddings and the final ensemble similarity variance $\sigma_s^2$. We fit LA following the same configuration in Appendix B.4 (Experimental details of MMTEB), and evaluate uncertainty using a binary classification proxy task on MS MARCO. Below we summarize the results:
> > > > - **Calibration of post-hoc probabilistic embeddings** (Section 4.3.1, Table 4):
> > > >     - We evaluated ECE for each model before and after applying LA.
> > > >          |  | **E5 (Deter.)** | **E5 (Prob.)** | **BGE (Deter.)** | **BGE (Prob.)** | **GTE (Deter.)** | **GTE (Prob.)** | **Ensemble (Deter.)** | **Ensemble (Prob.) (UEC)** |
> > > >         | ------------------ | ------------- | ------------ | -------------- | ------------- | -------------- | ------------- | ------------------- | ------------------ |
> > > >         | **ACC ↑**          | 83.57         | 83.57        | 84.19          | 84.19         | 82.74          | 82.74         | 84.83               | **85.41**          |
> > > >         | **ECE ↓**          | 0.063         | **0.036**    | 0.067          | **0.042**     | 0.077          | **0.043**     | 0.075               | **0.032**          |
> > > >         - “Deter.” denotes the deterministic embedding, while “Prob.” denotes the probabilistic embedding derived via LA. The deterministic ensemble corresponds to the weighted ensemble baseline, and the probabilistic ensemble corresponds to UEC (Ours).
> > > >     - Across all models, we show that **LA improves calibration, and UEC achieves the strongest accuracy and calibration performance.**
> > > >
> > > >
> > > > - **Calibration of ensemble similarity variance $\sigma_s^2$** (Section 4.3.2, Table 5):
> > > >     - Standard ECE is not directly applicable to evaluate the $\sigma_s^2$, since ECE assumes probability outputs, not variances. To address this limitation, we introduce **Var-ECE**, a calibration metric designed specifically for similarity variances (for more details, please refer to Appendix C.2):
> > > >         - We group predictions by predicted variance levels.
> > > >         - In each bin, we evaluate empirical similarity error.
> > > >         - Var-ECE measures how well predicted variances correlate with actual errors.
> > > >     - This new metric shows:
> > > >         | Metric | E5 | BGE | GTE | UEC |
> > > >         | --- | -- | -- | -- | -- |
> > > >         | Var-ECE $\downarrow$ | 0.035 | 0.051 | 0.038 | **0.028** |
> > > >     - Thus, we conclude that **$\sigma_s^2$ is meaningful epistemic uncertainty, not a learned temperature-like scaling factor.**
> > > >
> > > >
> > > > We greatly appreciate the reviewer’s suggestion, which has meaningfully strengthened the scientific rigor and clarity of the revised manuscript.

---

> > > > > ### Author Response · Authors · 2025-11-21
> > > > > **[W4] Rationale for Baseline Selection and Inclusion of Bayesian Model Averaging (BMA)**
> > > > >
> > > > > > **__W4)__** Why not included some other baselines (e.g., Bayesian model averaging or deep ensemble uncertainty fusion)?
> > > > >
> > > > > Thank you for this thoughtful suggestion. We appreciate the reviewer’s interest in comparing UEC against broader uncertainty-based ensemble methods.
> > > > >
> > > > > - Our primary goal in this work is to evaluate uncertainty-aware ensembling **within the post-hoc embedding setting, where all embedding models are frozen and only light-weight**. Because of this focus, we initially limited comparisons to post-hoc ensemble methods commonly used for embedding models.
> > > > >
> > > > > - That said, we fully agree that Bayesian Model Averaging (BMA) is a meaningful baseline. While classical BMA is not directly applicable because pretrained embedding models do not provide parameter posterior samples, we approximate BMA using 10 posterior samples obtained via the LA applied to each embedding model. We evaluate this BMA-style baseline under the MIRACL Subset setting (Section 4.1). We also include an uncertainty-fusion baseline that aggregates variances directly. The results are summarized below:
> > > > >
> > > > >     | Method  | nDCG@10 | Recall@10  | AUC@10     |
> > > > >     | ---- | ---------- | ---------- | ---------- |
> > > > >     | **BMA (10 samples)** | 58.84% | 78.83% | 85.39% |
> > > > >     | **Uncertainty Fusion** | 58.72% | 78.13% | 82.48% |
> > > > >     | **UEC (Ours)** | **59.65%** | **80.07%** | **91.04%** |
> > > > >
> > > > >     - UEC outperforms both BMA and uncertainty fusion on all metrics.
> > > > >     - Unlike BMA, UEC requires no posterior sampling, offering substantially lower computational overhead.
> > > > >     - **This aligns with our design goal: practical and scalable post-hoc ensembling for large embedding models.**
> > > > >
> > > > > We appreciate the reviewer’s comment, which encouraged us to strengthen the empirical comparison with principled Bayesian baselines.

---

> > > ### Author Response · Authors · 2025-11-21
> > > **[W2, Q2] Addressing Independence and Diagonal-Covariance Assumptions in the Gaussian Convolution**
> > >
> > > > **__W2)__** Strong independence and diagonal assumptions: The Gaussian convolution assumes independent embeddings and diagonal covariances. These assumptions are rarely valid for transformer-based embeddings.
> > >
> > > > **__Q2)__** How sensitive are your Bayes-optimal coefficients to correlation among models? Would cross-covariances alter the results?
> > >
> > >
> > > Thank you for raising this important question regarding the effect of cross-covariances among models. We respond in two parts: (1) clarification of the independence assumption, and (2) theoretical insight into how incorporating cross-covariances would alter the optimal coefficients.
> > >
> > >
> > > - **Independence assumption and its practical validity** :
> > >     - Our derivation of the Bayes-optimal coefficients indeed assumes that each model’s embedding posterior is independent. Under this assumption, the surrogate loss in Eq. (3) decomposes over models, which enables the closed-form inverse-variance solution in Eq. (4). We acknowledge that this assumption is what makes the solution analytically tractable.
> > >     - At the same time, we note that this assumption is not overly restrictive in our setting. The constituent encoders (E5, BGE, GTE) are trained independently, on different corpora and with different training objectives. As a result, their epistemic uncertainties—inferred through last-layer Laplace approximation—are reasonably approximated as uncorrelated.
> > >     - To validate this empirically, we directly measured embedding-space correlations on the MIRACL Subset. As shown below, the cross-model correlations are extremely small (close to zero), supporting the practical reasonableness of the independence assumption.
> > >         | Model Pair           | Pearson Corr. |
> > >         | -------------------- | ------------- |
> > >         | (E5, BGE-zh)         | 0.024         |
> > >         | (E5, MATRYOSHKA)     | –0.038        |
> > >         | (MATRYOSHKA, BGE-zh) | –0.047        |
> > >         | (BGE, E5)            | 0.041         |
> > >         | (BGE, GTE)           | 0.029         |
> > >         | (E5, GTE)            | 0.023         |
> > >         - These results confirm that the independence assumption aligns well with actual embedding-space behavior.
> > >
> > >     - In addition, we examined whether structured covariance (KFAC) materially alters uncertainty estimates. Although KFAC captures some parameter correlations, it is significantly more computationally expensive, and we found that diagonal LA achieves comparable calibration while being far more efficient. This supports that our diagonal, independence-based approach remains practical and effective (Appendix C.1, Table 17).
> > >         | Cov Type | ACC ↑ | ECE ↓     | Time Comp.         | Mem. Comp. |
> > >         | -------- | ----- | --------- | ------------------ | ---------- |
> > >         | **Diag** | 83.57 | 0.036     | $O(Nmd)$            | $O(md)$     |
> > >         | **KFAC** | 83.57 | **0.028** | $O(N(m^2+d^2)+m^3+d^3)$ | $O(m^2+d^2)$  |
> > >         - $m$ : penultimate layer width (hidden dim), $d$ : embedding dim (output dim), $N$ : number of sample to fit LA, $K$ : number of ensemble components
> > >         - Diagonal LA therefore provides nearly identical performance in line with [1] without the prohibitive computation and memory cost of KFAC.
> > >
> > > - **Theoretical effect of including cross-covariances**:
> > >     - If cross-covariance terms were incorporated, the expected squared distance between two Gaussian embeddings would become:
> > >     $$\mathbb{E}\|\mathbf{z}(\mathbf{x}) - \mathbf{z}(\mathbf{x}^\prime)\|^2 = \mathbb{E}\|\mathbf{\mu}(\mathbf{x}) - \mathbf{\mu}(\mathbf{x}^\prime)\|^2 + \mathrm{tr}(\Sigma(\mathbf{x})) + \mathrm{tr}(\Sigma(\mathbf{x}^\prime)) - 2\mathrm{tr}(\Sigma_{\text{cross}}(\mathbf{x}, \mathbf{x}^\prime)).$$
> > >     This has several important consequences:
> > >         - The surrogate loss would no longer decompose per model, so the coefficients are coupled.
> > >         - Each coefficient $\pi_k(x)$ would depend not only on its own variance but also on pairwise covariances with the other models.
> > >         - Instead of the simple inverse-variance form in Eq. (4), **the optimal weights would arise from a quadratic program, analogous to generalized least squares** where the full precision matrix replaces the diagonal precision.
> > >     - Thus, **Eq. (4) can be viewed as a simplified version of a more general precision-matrix–aware weighting.**
> > >
> > >
> > > We sincerely appreciate this insightful question, which allowed us to clarify the theoretical implications of independence, justify its practical validity, and empirically demonstrate that diagonal LA remains both reliable and computationally efficient.
> > >
> > > ----
> > > **References**
> > >
> > > [1] Zhdanov, Maksim, Stanislav Dereka, and Sergey Kolesnikov. "Identity Curvature Laplace Approximation for Improved Out-of-Distribution Detection." 2025 IEEE/CVF Winter Conference on Applications of Computer Vision (WACV). IEEE, 2025.

---

> > > > ### Author Response · Authors · 2025-11-29
> > > >
> > > > We appreciate the valuable comments shared during the review process. Our responses and clarifications have been posted, and we would be glad to provide any further details if needed.

---

### Official Review · Reviewer_TC4d · 2025-11-01

**Soundness:** 3
**Presentation:** 3
**Contribution:** 2
**Rating:** 4
**Confidence:** 2

**Summary:**

### Summary

The paper proposes **Uncertainty-Driven Embedding Convolution (UEC)**, a post-hoc probabilistic ensemble method for text embedding models. It combines uncertainty estimation, probabilistic ensembling, and calibrated similarity scoring to improve robustness in retrieval and representation tasks.

UEC works in three steps:
1. **Laplace Approximation (LA)** on the last layer produces a Gaussian posterior over embeddings, modeling epistemic uncertainty.
2. **Gaussian convolution** fuses multiple model embeddings into a combined mean and covariance with theoretically optimal coefficients.
3. **Probit similarity** computes uncertainty-aware similarity, provably approximating the Wasserstein-2 and Jeffreys divergences.

Experiments on MMTEB (retrieval, STS, reranking, clustering, bitext mining) show consistent improvements over deterministic ensembles with negligible computational overhead.

**Strengths:**

### Strengths

1. **Principled probabilistic foundation** — Correct derivation of Gaussian embeddings from Laplace posteriors.
2. **Theoretical rigor** — Ranking-preserving link between probit similarity, Wasserstein-2, and Jeffreys divergence.
3. **Practicality** — Post-hoc, model-agnostic, and nearly cost-free in computation.
4. **Empirical consistency** — Strong performance across MMTEB tasks and clear ablations.
5. **Trustworthy AI alignment** — Explicit uncertainty propagation enhances interpretability and robustness.
6. **Reproducibility** — Clear methodology, dataset references, and open hyperparameter reporting.

**Weaknesses:**

### Weaknesses

1. **Limited type of uncertainty.**
   The method models only **epistemic uncertainty** (parameter uncertainty via Laplace Approximation on the last layer). It does not capture **aleatoric** or **predictive uncertainty**, nor propagate epistemic uncertainty through deeper layers.

2. **Strong independence and diagonal assumptions.**
   The Gaussian convolution assumes independent embeddings and diagonal covariances. These assumptions are rarely valid for transformer-based embeddings.

3. **Calibration not analyzed.**
   The paper assumes variances from LA are well calibrated, but provides no empirical evidence (e.g., ECE or reliability plots).

4. **Benchmark scope limited.**
   Only MMTEB is used. Out-of-distribution or multilingual setups would better demonstrate uncertainty benefits.

5. **Incremental conceptual contribution.**
   The method is a sophisticated recombination of known tools (Laplace, Gaussian averaging, probit correction).

6. **Fairness risk.**
   Down-weighting uncertain embeddings could amplify demographic bias if uncertainty correlates with data imbalance.

**Questions:**

### Questions for Authors

1. Which **type(s) of uncertainty** does UEC explicitly model? From the derivations, it seems to capture **epistemic uncertainty** only (via parameter uncertainty in the last layer). Please confirm and discuss this limitation.
2. How sensitive are your Bayes-optimal coefficients to correlation among models? Would cross-covariances alter the results?
3. How robust is the Laplace fit when performed on out-of-domain data (e.g., fitting LA on MS MARCO, evaluating on STS)?
4. Could you provide calibration diagnostics (ECE, reliability curves) to evaluate whether \(\sigma_s^2\) reflects meaningful confidence?
5. Would deeper (full-network) Laplace approximations substantially alter uncertainty estimates?
6. Please include a sensitivity study on \(\beta\) and temperature parameters.
7. Can you empirically validate that the probit similarity preserves ranking order with respect to exact Wasserstein/Jeffreys distances?

---

> ### Author Response · Authors · 2025-11-21
>
> We deeply appreciate Reviewer TC4d for providing an exceptionally comprehensive and insightful review.
> Your feedback spans methodological scope, theoretical assumptions, empirical validation, calibration, fairness considerations, robustness, and hyperparameter sensitivity—offering a truly multi-faceted examination of our work.
>
> Key concerns identified by the reviewer:
> - UEC models only epistemic uncertainty; no aleatoric/predictive uncertainty
> - Independence/diagonal assumptions in covariance
> - Lack of calibration diagnostics for variance estimates
> - Robustness of LA under domain shift
> - Scope: limited benchmarks (MMTEB), lack of OOD/multilingual evidence
> - Conceptual novelty and recombination of known tools
> - Fairness concerns when uncertainty correlates with imbalance
> - Sensitivity to covariance structure, deeper LA, $\beta$ and temperature
> - Whether probit similarity preserves ranking with respect to true distances
>
> We are genuinely grateful for this level of engagement.
> Your comments helped us strengthen the methodology discussion, clarify limitations, add missing analyses, and position the work more realistically and responsibly.
> Thanks to your review, the revised manuscript now presents UEC in a more mature, balanced, and multi-angle manner.
>
> Detailed responses to each point follow below.

---

> ### Author Response · Authors · 2025-11-21
> **[W1, Q1] Clarifying the Scope of Uncertainty Modeled by UEC and Discussing Its Limitations**
>
> > **__W1)__** Limited type of uncertainty: The method models only epistemic uncertainty (parameter uncertainty via Laplace Approximation on the last layer). It does not capture aleatoric or predictive uncertainty, nor propagate epistemic uncertainty through deeper layers.
>
> > **__Q1)__** Which type(s) of uncertainty does UEC explicitly model? From the derivations, it seems to capture epistemic uncertainty only (via parameter uncertainty in the last layer). Please confirm and discuss this limitation.
>
> Thank you for pointing this out. We fully agree with the reviewer’s observation: UEC explicitly models only epistemic uncertainty, obtained via the Laplace Approximation applied to the final layer. It does not capture aleatoric or full predictive uncertainty, nor does it propagate uncertainty through deeper layers.
>
> - We have clarified this limitation in the revised manuscript. In particular, the Conclusion & Discussion section now explicitly states that UEC currently models only epistemic uncertainty and that extending the framework to aleatoric and broader predictive uncertainty is an important direction for future work.
> > **__(Section 5, L532-539)__** [...] While UEC currently models only epistemic uncertainty and relies on diagonal LA, extending it to capture aleatoric or full predictive uncertainty and exploring richer posterior structures with efficient computation represent promising directions for improving robustness. Furthermore, UEC assumes that all embeddings share the same dimensionality and may inherit biases from the underlying models; relaxing the dimensionality constraint and developing bias-aware or fairness-adjusted uncertainty mechanisms offer important avenues for responsible deployment. Finally, applying UEC to multimodal settings—where heterogeneous uncertainty sources interact across vision, speech, and text—presents an exciting opportunity to broaden the framework’s impact.
>
> We appreciate this comment, which helped us better articulate the scope and limitations of UEC and refine the discussion of future research directions.

---

> ### Author Response · Authors · 2025-11-21
> **[W3, Q4] Calibration Analysis of LA-Derived Uncertainty and the Similarity Variance**
>
> > **__W3)__** Calibration not analyzed: The paper assumes variances from LA are well calibrated, but provides no empirical evidence (e.g., ECE or reliability plots).
>
> > **__Q4)__** Could you provide calibration diagnostics (ECE, reliability curves) to evaluate whether $\sigma_s^2$ reflects meaningful confidence?
>
> Thank you for raising this important point. Your feedback prompted us to substantially strengthen the empirical validation of our uncertainty estimates, and we now include explicit calibration diagnostics in the revised manuscript.
>
> To address your comment, we added a dedicated Uncertainty Estimation Diagnosis section (Section 4.3), evaluating both (1) the calibration of the post-hoc probabilistic embeddings produced by LA, and (2) whether the ensemble-level similarity variance $\sigma_s^2$ meaningfully reflects predictive error.
> - **Calibration of LA-based probabilistic embeddings** (Section 4.3.1, Table 4) : We measure ECE for each encoder before and after applying LA. Across E5, BGE, and GTE, LA consistently improves calibration while preserving accuracy, and UEC achieves the best overall calibration:
>     |  | **E5 (Deter.)** | **E5 (Prob.)** | **BGE (Deter.)** | **BGE (Prob.)** | **GTE (Deter.)** | **GTE (Prob.)** | **Ensemble (Deter.)** | **Ensemble (Prob.) (UEC)** |
>     | ------------------ | ------------- | ------------ | -------------- | ------------- | -------------- | ------------- | ------------------- | ------------------ |
>     | **ACC ↑**          | 83.57         | 83.57        | 84.19          | 84.19         | 82.74          | 82.74         | 84.83               | **85.41**          |
>     | **ECE ↓**          | 0.063         | **0.036**    | 0.067          | **0.042**     | 0.077          | **0.043**     | 0.075               | **0.032**          |
>
>     - These results show that each probabilistic embedding is well-calibrated and that UEC further improves calibration at the ensemble level.
>
> - **Calibration of ensemble similarity variance $\sigma_s^2$** (Section 4.3.2, Table 5): Standard ECE is not suitable for evaluating variance outputs, so we introduce Var-ECE, a metric designed to assess whether larger predicted variances correspond to higher empirical error (details provided in Appendix C.2). Using this metric, we find:
>     | Metric | E5 | BGE | GTE | UEC |
>     | --- | -- | -- | -- | -- |
>     | Var-ECE $\downarrow$ | 0.035 | 0.051 | 0.038 | **0.028** |
>     - **This confirms that $\sigma_s^2$ accurately captures epistemic uncertainty**, rather than behaving as an uncalibrated scaling factor.
>
> We appreciate this suggestion—it significantly improved the rigor of the revised manuscript and clarified the role of uncertainty in UEC.

---

> ### Author Response · Authors · 2025-11-21
> **[W4, Q3] Benchmark scope & OOD robustness of LA**
>
> > **__W4)__** Benchmark scope limited: Only MMTEB is used. Out-of-distribution or multilingual setups would better demonstrate uncertainty benefits.
>
> > **__Q3)__** How robust is the Laplace fit when performed on out-of-domain data (e.g., fitting LA on MS MARCO, evaluating on STS)?
>
> Thank you for raising this important point regarding benchmark coverage and the robustness of the Laplace fit under distribution shift. We address both concerns below.
> - **Benchmark coverage is broader than it may appear** : Although MMTEB is our main benchmark, it already spans a very wide range of domains and languages. In the revised version, we clarified this more explicitly:
>     - **Domains covered**: legal text, finance, scientific papers, question answering, poetry, social media, multilingual news, etc. (About 10)
>     - **Languages covered**: English, German, French, Arabic, Hindi, etc. (More than 20)
>
>     Across diverse domains and languages, UEC consistently shows better performances than baselines.
>
> - **MIRACL Subset: UEC combines language-specialized models effectively** : We emphasize (and revised the text accordingly) that MIRACL Subset experiments already function as explicit multilingual stress tests:
>     - Each base model (E5, BGE-zh, MATRYOSHKA) is specialized for specific languages
>     - UEC must infer this per-query without using language labels
>
>     This setting naturally tests whether uncertainty estimation helps the ensemble adapt across languages. As shown in Figure 3 & Figure 4, UEC:
>     - Matches or exceeds the oracle on nDCG@10 and Recall@10
>     - **Outperforms the oracle on AUC@10**, showing stronger uncertainty calibration
>     - Learns **language-dependent coefficients** (Figure 4)
>
> - **MMTEB training→evaluation is already an OOD evaluation** : We appreciate the reviewer pointing out the importance of OOD robustness. We note Section 4.2 that:
>     - LA is **fitted on MS MARCO/SNLI**
>     - MMTEB evaluation tasks **span completely different domains**
>
>     Thus, every reported MMTEB result is **implicitly an OOD test**, because embedding models are rarely trained on the same domain as the retrieval/STS/classification tasks used for evaluation. This is the standard setting for embedding evaluation—and one that UEC handles well.
>
>
> - **Additional OOD experiment added** : We added a new OOD validation experiment to directly test the reviewer’s concern.
>     - Setting :
>         - LA fitted on MS MARCO (English retrieval)
>         - Evaluation performed on DMath [1] (bilingual math problem retrieval; English–Korean)
>             - Overall : Use all 5 categories of DMath
>             - Math-Heavy : Pick 2 challenging categories in DMath dataset
>
>     - We ensemble Qwen3-Embedding [2] (math-specialized) (We hypothesize that Qwen3-Embedding can capture mathematical semantics reasonably well, given its strong performance on MTEB-Code and the STEM-oriented Qwen3 foundation.) and KoE5 [3] (Korean-specialized) and compare against baselines.
>     - Results :
>         | Model          | Overall   | Math-Heavy |
>         | -------------- | --------- | ---------- |
>         | KoE5           | 69.87     | 53.42      |
>         | Qwen3          | 68.74     | 64.86      |
>         | Uniform Avg    | 70.29     | 62.75      |
>         | Weighted Avg   | 72.68     | 65.33      |
>         | **UEC (Ours)** | **74.13** | **66.28**  |
>         - Despite being trained/fitted on MS MARCO, UEC performs best overall and best on Math-Heavy, showing strong OOD generalization.
>
>
> We appreciate your insightful comment regarding benchmark coverage and the robustness of the Laplace approximation under distributional changes. Our responses to both aspects are provided below.
>
>
> ---
> **References**
>
> [1] Kim, Jiwoo, et al. "It ain’t over: A multi-aspect diverse math word problem dataset." Proceedings of the 2023 Conference on Empirical Methods in Natural Language Processing. 2023.
>
> [2] Yang, An, et al. "Qwen3 technical report." arXiv preprint arXiv:2505.09388 (2025).
>
> [3] NLP & AI Lab and Human-Inspired AI research. KoE5: A New Dataset and Model for Improving Korean Embedding Performance. 2024.

---

> > ### Author Response · Authors · 2025-11-21
> > **[W5] Clarifying the Novel Contributions Beyond Existing Statistical Components & [W6] Addressing Fairness Concerns in Uncertainty-Weighted Embedding Ensembles**
> >
> > > **__W5)__** Incremental conceptual contribution: The method is a sophisticated recombination of known tools (Laplace, Gaussian averaging, probit correction).
> >
> > Thank you for this thoughtful observation. We agree that UEC builds upon well-established components such as the Laplace approximation, variance-based weighting, and probit-style marginalization. Your comment helped us clarify where the novelty of our work truly lies.
> >
> > We believe our contributions are distinct in the following ways:
> > 1. **First post-hoc probabilistic embedding ensemble** : Unlike prior work that models uncertainty within a single encoder, UEC converts multiple pretrained deterministic encoders into probabilistic ones without retraining and fuses them using their estimated epistemic variances. Existing embedding ensembles typically rely on heuristic or uniform averaging and do not propagate uncertainty. To our knowledge, UEC is the first framework enabling a principled, uncertainty-aware ensemble that remains fully compatible with large pretrained libraries.
> > 2. **A new surrogate-loss formulation yielding Bayes-optimal ensemble weights** : While inverse-variance ideas appear in classical estimation, applying them to embedding similarity and retrieval is nontrivial. We introduce a retrieval-oriented surrogate loss derived from probabilistic similarity, show that it leads to a unique closed-form optimizer on the simplex, and interpret this optimizer as Bayes-optimal under the surrogate. This fills a theoretical gap, as embedding ensembles have lacked a grounded derivation for their weighting schemes.
> > 3. **Uncertainty-aware similarity linked to a Wasserstein-bound approximation** : Beyond ensembling, UEC provides a new similarity estimator obtained by marginalizing Gaussian embedding distributions. We further show that its probit form corresponds to a bounded approximation of distributional similarity (Wasserstein/Jeffreys). To our knowledge, this is the first connection between probabilistic embeddings, uncertainty propagation, and computationally efficient Wasserstein-bounded similarity for retrieval.
> >
> > We appreciate this comment, which allowed us to better articulate these contributions and highlight the conceptual novelty of our framework more clearly.
> >
> > ------
> > ------
> > ------
> > > **__W6)__** Fairness risk: Down-weighting uncertain embeddings could amplify demographic bias if uncertainty correlates with data imbalance.
> >
> > Thank you for raising this important concern. We agree that uncertainty-based weighting must be handled carefully, especially in the presence of demographic imbalance. However, we believe the proposed method has low risk to amplify the bias, following reason:
> > - **Our method is instance-wise**: UEC assigns ensemble weights separately for each input, meaning that embeddings are not globally favored or suppressed. Instead, for a minority-group query, the encoder that represents such data more reliably naturally receives higher weight—because its epistemic uncertainty for that instance is lower. In this sense, UEC can mitigate bias amplification by dynamically favoring the model that performs best on the given subgroup, rather than enforcing a static weighting scheme that may over-represent majority-trained encoders.
> > - **Of course, this desirable behavior relies on uncertainty being well calibrated.** To address this, the revised version includes a new Uncertainty Estimation Calibration analysis (Section 4.3), demonstrating that both the per-model uncertainties and the ensemble similarity variance are meaningfully calibrated (via ECE and Var-ECE). These results provide empirical evidence that **the uncertainty estimates used for reweighting behave reliably rather than reflecting dataset imbalance or noise.**
> >
> > We appreciate the reviewer for highlighting this fairness consideration; it helped us clarify the implications of instance-wise uncertainty weighting and reinforced our motivation for carefully validating our uncertainty estimates.

---

> > > ### Author Response · Authors · 2025-11-21
> > > **[Q5] Impact of Deeper (Full-Network) Laplace Approximation on Uncertainty Estimates & [Q6] Sensitivity Analysis of the beta and Temperature Parameters**
> > >
> > > > **__Q5)__** Would deeper (full-network) Laplace approximations substantially alter uncertainty estimates?
> > >
> > > Thank you for raising this important question. We fully agree that understanding whether deeper or full-network Laplace Approximation (LA) would meaningfully change the uncertainty estimates is essential.
> > >
> > > - First, applying full-network LA to large embedding models is generally computationally infeasible due to the prohibitive cost of estimating and storing full-layer curvature. Prior works [1,2] have shown that last-layer LA already provides strong and stable uncertainty estimates while offering orders-of-magnitude efficiency benefits. This motivated our decision to adopt last-layer LA as the default configuration.
> > >
> > > - To further examine this question empirically, we additionally evaluated LA using deeper curvature modeling by extending the number of included layers beyond the final projection layer. Using E5 on MS MARCO with a binary classification proxy task, we compared last-layer LA to 2-, 4-, and 6-layer LA. The results (also reported in Appendix C.1, Table 18) are:
> > >     | **# Layers** | **1 (Last)** | **2** | **4** | **6** |
> > >     | ------------ | ------------ | ----- | ----- | ----- |
> > >     | **ACC ↑**    | 83.57        | 83.57 | 83.56 | 83.57 |
> > >     | **ECE ↓**    | 0.036        | 0.035 | 0.038 | 0.035 |
> > >     - In line with the prior works [1,2], these results show that:
> > >         - Calibration remains effectively unchanged, even when curvature is modeled using up to six layers.
> > >         - Deeper LA does not yield meaningful improvements, while its computational cost grows substantially.
> > >
> > >
> > > ---
> > > **References**
> > >
> > > [1] Daxberger, Erik, et al. "Laplace redux-effortless bayesian deep learning." Advances in neural information processing systems 34 (2021): 20089-20103.
> > >
> > > [2] Hobbhahn, Marius, Agustinus Kristiadi, and Philipp Hennig. "Fast predictive uncertainty for classification with Bayesian deep networks." Uncertainty in artificial intelligence. PMLR, 2022.
> > >
> > > ------
> > > ------
> > > ------
> > >
> > > > **__Q6)__** Please include a sensitivity study on $\beta$ and temperature parameters.
> > >
> > > Thank you for pointing out this important aspect. We fully agree that a sensitivity analysis is essential for understanding the practical robustness of the proposed method.
> > >
> > > - We conducted new sensitivity experiments about (1) $\beta$ (controls how strongly the predicted variance contributes to the uncertainty-aware similarity) and (2) temperature $T$ (sharpens the ensemble coefficients) on:
> > >     - MIRACL Subset (Appendix D, Figure 6)
> > >         - $\beta$
> > >             | Metric / $\beta$ | 1e-4 | 1e-3 | 1e-2 | 1e-1
> > >             | --- | --- | --- |  --- | --- |
> > >             | nDCG@10 | 58.92 | 59.38 | **59.65** | 59.11  |
> > >             | Recall@10 | 78.26 | 79.15 | **80.07** | 78.83|
> > >             | AUC@10 | 83.77 | 86.08 | **91.04** | 85.27  |
> > >         - Temperature $T$
> > >             | Metric / $T$ | 0.1 | 1 | 1.5 | 2
> > >             | --- | --- | --- |  --- | --- |
> > >             | nDCG@10 | 52.68 | 57.91 | **59.65** | 56.44  |
> > >             | Recall@10 | 72.06 | 77.62 | **80.07** | 77.27|
> > >             | AUC@10 | 60.36 | 78.34 | **91.04** | 74.18  |
> > >
> > >     - MMTEB (StackOverflowQA)  (Appendix D, Figure 7)
> > >         - $\beta$
> > >             | Metric / $\beta$ | 1e-4 | 1e-3 | 1e-2 | 1e-1
> > >             | --- | --- | --- |  --- | --- |
> > >             | nDCG@10 | 89.76 | 90.11 | **90.26** | 89.94  |
> > >             | Recall@100 | 98.64 | 99.03 | **99.40** | 98.93|
> > >             | AUC@10 | 88.13 | 89.17 | **89.83** | 88.75  |
> > >         - Temperature $T$
> > >             | Metric / $T$ | 0.1 | 1 | 1.5 | 1.8 | 2
> > >             | --- | --- | --- |  --- | --- | --- |
> > >             | nDCG@10 | 88.41 | 89.03 | 90.22 | **90.26**  | 90.15 |
> > >             | Recall@100 | 96.54 | 97.72 | 99.21 | **99.40** | 99.11|
> > >             | AUC@10 | 76.04 | 83.86 | 89.47 | **89.83**  | 89.36 |
> > >
> > > - These analyses reveal the following:
> > >     - Both hyperparameters have a noticeable effect on performance.
> > >     - Temperature $T$ has larger impact than $\beta$.
> > >     - Importantly, **the optimal ranges are stable across datasets, suggesting that the practical tuning effort is modest.**
> > >
> > > We appreciate your comment, which prompted us to enrich the empirical evaluation with this important analysis.

---

> > > > ### Author Response · Authors · 2025-11-21
> > > > **[Q7] Empirical Validation of Probit Similarity Ranking vs. Wasserstein/Jeffreys Distances**
> > > >
> > > > > **__Q7)__** Can you empirically validate that the probit similarity preserves ranking order with respect to exact Wasserstein/Jeffreys distances?
> > > >
> > > > Thank you for this excellent suggestion. We fully agree that empirically verifying whether the probit-based similarity preserves ranking order with respect to exact Wasserstein and Jeffreys distances is an important validation of our approximation.
> > > >
> > > > To address this, we added a dedicated ranking-consistency evaluation.
> > > > - **Ranking preservation against Wasserstein and Jeffreys distances** : Under the MIRACL Subset setting (Section 4.1), we compute rankings using (i) UEC’s probit similarity, (ii) exact 2-Wasserstein distances, and (iii) Jeffreys divergence. We then measure: (1) Spearman correlation (monotonic ranking agreement), (2) Kendall $\tau$ (pairwise order consistency), (3) Overlap@10 (top-10 agreement).
> > > >
> > > >     |                    | UEC vs. $W_2^2$ | UEC vs. Jeffreys |
> > > >     | ------------------ | --------------- | ---------------- |
> > > >     | **Spearman Corr.** | **0.825**       | **0.817**        |
> > > >     | **Kendall τ**      | **0.864**       | **0.852**        |
> > > >     | **Overlap@10**     | **0.870**       | **0.866**        |
> > > >
> > > >     - These results demonstrate that **UEC’s probit similarity preserves the ranking structure of the exact Wasserstein/Jeffreys distances extremely well**, with >80% pairwise agreement and ~87% top-10 overlap. Empirically, this confirms that the approximation maintains the essential ordering behavior of the theoretically grounded metrics.
> > > >
> > > > - **Comparisons and stability analysis** (Appendix B.2, Table 9) : We reported comparisons of UEC against KL-based (Jeffreys), Wasserstein, and CSD[1] similarities.
> > > >     | **Similarity** | **nDCG@10** | **Recall@10** | **AUC@10** | **Memory Complexity** | **Time (Rel.)** | **Time (Complex.)** |
> > > >     | -------------- | ----------- | ------------- | ---------- | --------------------- | --------------- | ------------------- |
> > > >     | KLD            | 58.14       | 78.33         | 87.60      | $O(KD + KD^2)$          | 1.48            | $O(KD^2)$             |
> > > >     | 1-Wasserstein  | 57.87       | 88.21         | 79.92      | $O(KD)$                | 1.21            | $O(KD^{1.5})$        |
> > > >     | 2-Wasserstein  | 57.71       | 88.48         | 81.05      | $O(KD + KD^2)$          | 1.60            | $O(KD^2)$             |
> > > >     | CSD            | 58.23       | 79.10         | 86.92      | $O(KD)$                | 0.88            | $O(KD)$              |
> > > >     | **UEC (Ours)** | **59.65**   | **80.07**     | **91.04**  | **$O(KD)$**            | **1.00**        | **$O(KD)$**          |
> > > >     - “Rel.” and “Comp.” denote relative runtime (normalized to UEC=1.00) and asymptotic complexity per similarity computation.
> > > >     - UEC achieves comparable or better ranking quality while being  efficient, supporting its value as a practical and stable approximation within the embedding ensemble framework.
> > > >
> > > > We sincerely appreciate this question.It motivated a direct empirical validation that strengthens the theoretical narrative and demonstrates the reliability of the probit-based similarity approximation.
> > > >
> > > >
> > > > ---
> > > > **References**
> > > >
> > > > [1] Sanghyuk Chun. Improved probabilistic image-text representations. arXiv preprint arXiv:2305.18171, 2023.

---

> > > > > ### Author Response · Authors · 2025-11-29
> > > > >
> > > > > Thank you for the thoughtful reviews. We have provided detailed responses and clarifications during the discussion. If there is anything further that would benefit from additional explanation, we are more than willing to clarify.

---

### Official Review · Reviewer_ed1F · 2025-11-03

**Soundness:** 2
**Presentation:** 2
**Contribution:** 2
**Rating:** 2
**Confidence:** 4

**Summary:**

This paper introduces Uncertainty-driven Embedding Convolution (UEC) which is a method to construct ensemble of embeddings in an uncertainty aware method. UEC consists of three key components. First, it converts pre-trained deterministic embeddings into probabilistic embeddings in a post-hoc fashion. Second, it computes adaptive ensemble weights based on estimated uncertainty, down-weighting less reliable embeddings. This weighting strategy is grounded in a Bayes-optimal solution under a surrogate loss. Third, UEC introduces an uncertainty-aware similarity function that incorporates both distance and variance into the similarity score, offering a theoretically grounded and efficient surrogate to distributional distances. Extensive experiments on diverse benchmarks demonstrate that UEC consistently improves both performance and robustness by leveraging principled uncertainty modeling

**Strengths:**

- This work provides a way to ensemble embeddings in an uncertainty aware manner.

**Weaknesses:**

- It seems like this work contains a lot of jargon without providing any insights on what the proposed method is and how does it fit into the existing literature.
- This work has major theoretical flaws.

**Questions:**

- Can the autors give some insight on how they are converting the deterministic embeddings into probabilisitic ones in section 3.1? It is very hard to follow currently.
- The solution of eqn 3 should put probability 1 on the lowest value. Why do the authors claim the result in eqn 4?

---

> ### Author Response · Authors · 2025-11-21
>
> We sincerely thank Reviewer ed1F for the thoughtful and constructive feedback. Your comments helped us re-examine the theoretical foundations and significantly improve the clarity of Section 3.
>
> Key concerns identified by the reviewer:
> - Use of jargon and insufficient explanation of the method’s intuition
> - Difficulty following the conversion from deterministic to probabilistic embeddings
> - Theoretical concerns regarding the surrogate optimization
> - Question about why Eq. (3) does not assign all probability to the minimum term
>
> Your feedback provided a valuable opportunity to revisit the theoretical derivations, strengthen the argumentation, and revise the exposition so that the method’s intuition is more transparent. We are grateful for prompting us to clarify both the reasoning and flow of Section 3.
>
> We address each of these points in detail in the following response.

---

> > ### Author Response · Authors · 2025-11-21
> > **[W1] Clarifying the Method and Reducing Jargon for Improved Readability**
> >
> > > **__W1)__** It seems like this work contains a lot of jargon without providing any insights on what the proposed method is and how does it fit into the existing literature.
> >
> > Thank you for pointing this out. We fully acknowledge that the original draft of Section 3 suffered from dense exposition and excessive reliance on mathematical formalism. We apologize for the difficulty this caused in understanding the method. Following your suggestion, we substantially revised Section 3 to make the method clearer, more intuitive, and less jargon-heavy. Concretely:
> > - **Major rewrite of Section 3.1 (Post-hoc Probabilistic Embedding Model)** : We rewrote this subsection to provide a high-level explanation of Laplace Approximation (LA), avoiding unnecessary second-order details, and moved the full derivation to Appendix A.1.
> > - **Added intuitive explanations throughout Section 3.2** : We inserted multiple intuitive descriptions directly into the derivation steps—including the motivation for the surrogate loss, the role of uncertainty, and the effect of coefficient optimization. These appear mainly in the following revised segments:
> >     > **__(Section 3.2, L215~221)__** This surrogate loss decomposes into fidelity and epistemic uncertainty. While $\mathbf{\mu}_k(\mathbf{x})$ and $\mathbf{\Sigma}_k(\mathbf{x})$ are available at retrieval time, computing $\mathbf{\mu}_k(\mathbf{x}')$ and $\mathbf{\Sigma}_k(\mathbf{x}')$ requires evaluating the embedding model on every document. However, in practical retrieval systems, document embeddings are precomputed once and stored in the index, whereas the query $\mathbf{x}$ arrives only at inference time (Zhou et al., 2022; Zhao et al., 2024). Since the surrogate loss in Eq.2 contains terms that depend on $\mathbf{x}'$, which cannot be recomputed per query, we discard all document-dependent terms and retain only the query-dependent components. This yields a retrieval-feasible approximation of Eq.2, leading directly to the tractable convex optimization:
> >
> >     > **__(Section 3.2, L226~233)__** This convex linear objective combines variance and embedding magnitude into a tractable surrogate loss. Although the objective is linear, the probability simplex constraints yield a well-posed convex program with a unique closed-form solution under standard convex analysis (Boyd & Vandenberghe, 2004). This ensures that the resulting coefficients are well-defined and stable, avoiding degenerate behavior even under heterogeneous uncertainty across models. The solution therefore yields a soft inverse–risk weighting that distributes probability mass smoothly, rather than collapsing entirely onto the smallest term, leading to a more robust and interpretable ensemble. Its unique closed-form solution is the inverse-variance weighting:
> >
> >     > **__(Section 3.2, L237~246)__** Intuitively, the inverse–variance rule assigns lower weight to embeddings whose large epistemic variance signals low confidence, while giving more influence to models that produce stable, self-consistent representations for the given query. This precision-based structure mirrors classical Bayesian aggregation, where estimators with higher certainty contribute proportionally more to the final prediction (Singh et al., 2005). Importantly, this mechanism also adapts to data-wise heterogeneity: for queries that fall into domains where certain encoders are weak or out-of-distribution, their variances naturally increase, causing UEC to down-weight them without requiring explicit domain labels. Under this interpretation, the resulting coefficient vector $π^∗_k(x)$ is precisely the minimizer of the uncertainty-aware surrogate loss in Eq. 3, representing the Bayes-optimal solution under the surrogate formulation. Full derivation appears in Appendix A.2.
> >
> > - **Replaced jargon-heavy phrases with clearer terms** : Throughout the paper, we made an effort to substitute jargon-laden expressions with more transparent terminology.

---

> > > ### Author Response · Authors · 2025-11-21
> > > **[W2] Addressing and Resolving Theoretical Issues in the Original Derivations**
> > >
> > > > **__W2)__** This work has major theoretical flaws.
> > >
> > > We appreciate this critical remark. This prompted us to re-examine each step of the derivation and ensure theoretical soundness and clarity. To address this:
> > > - **We carefully rewrote the derivation of Eq. (3)→Eq. (4)**: We now explicitly motivate the surrogate loss, justify the removal of document-dependent terms based on retrieval-time constraints, and show that the resulting optimization reduces to a convex linear program whose unique minimizer produces a soft inverse-risk weighting. This appears in the revision Section 3.2, L215–233 & L237–246.
> > >
> > > - **Added intuitive reasoning for the solution** : We added explanations showing why inverse-variance weighting arises naturally, how uncertainty interacts with the optimization, and why the minimizer does not degenerate to a single model.
> > >
> > > - **Added supporting references** : We grounded the convexity argument with standard results from Boyd & Vandenberghe (2004) [1] and motivation for precision-weighted aggregation from Singh et al. (2005) [2].
> > >
> > > We believe these revisions remove ambiguity and substantially strengthen the theoretical presentation.
> > >
> > > -----
> > > **References**
> > >
> > > [1] Stephen Boyd and Lieven Vandenberghe. Convex optimization. Cambridge university press, 2004.
> > >
> > > [2] Kesar Singh, Minge Xie, and William E Strawderman. Combining information from independent sources through confidence distributions. 2005.

---

> ### Author Response · Authors · 2025-11-21
> **[Q1] Providing Intuitive Explanation of the Deterministic → Probabilistic Embedding Conversion**
>
> > **__Q1)__** Can the authors give some insight on how they are converting the deterministic embeddings into probabilistic ones in section 3.1? It is very hard to follow currently.
>
> We apologize for the difficulty. The original exposition included too much low-level detail about the Laplace Approximation (LA), obscuring the core intuition.
>
> - Clarification of the Laplace Approximation process
>     - A pretrained embedding model normally produces a single fixed vector for each input. LA converts this fixed vector into a distribution by estimating how uncertain the model’s final-layer parameters are. Concretely, LA examines how “sharp” or “flat” the loss landscape is around the trained weights. **If the loss barely changes when the weights move slightly, LA interprets this as high uncertainty; if the loss rises sharply, LA interprets this as high confidence.**
>     - Using this curvature information, LA fits a simple Gaussian distribution to the final-layer weights. Importantly, this step does not require retraining the model; it only inspects the pretrained weights. When these Gaussian-distributed weights are applied to the (fixed) penultimate-layer representation, the resulting embedding also becomes Gaussian. Thus, **each deterministic embedding is replaced by a probabilistic embedding whose mean reflects the original representation and whose variance reflects how confident the model is about that embedding.**
>     - This process provides **a lightweight way to extract epistemic uncertainty from an existing embedding model**.
>
> - To clarify how deterministic embeddings are converted into probabilistic ones, we now provide a high-level and intuition-focused explanation of the Laplace Approximation (LA). Instead of introducing heavy mathematical detail, we emphasize the core idea. (Section 3.1, L149-189)
>     > **Laplace Approximation** To convert a deterministic embedding model into a probabilistic one, we estimate a Gaussian posterior over its last-layer parameters using the Laplace Approximation (LA) (MacKay, 1992; Ritter et al., 2018). Given training data $\mathcal{D}$ and last-layer weights $\mathbf{W}^{(L)}$, LA constructs a second-order Taylor expansion of the negative log-posterior $-\log p(\mathbf{W}^{(L)}|\mathcal{D})$ around the maximum a posteriori (MAP) estimate $\widehat{\mathbf{W}}^{(L)}$:
>     >
>     > $$
>     > -\log p(\mathbf{W}^{(L)}|\mathcal{D}) \approx -\log p(\widehat{\mathbf{W}}^{(L)}|\mathcal{D}) + \frac{1}{2} (\mathbf{W}^{(L)} - \widehat{\mathbf{W}}^{(L)})^\top \mathbf{H}_{\widehat{\mathbf{W}}^{(L)}} (\mathbf{W}^{(L)} - \widehat{\mathbf{W}}^{(L)}),
>     > $$
>     >
>     > where $\mathbf{H}_{\widehat{\mathbf{W}}^{(L)}}$ is the Hessian of the negative log-posterior evaluated at the MAP. This MAP solution corresponds to the final-layer parameters of the pre-trained embedding model. Prior work shows that final-layer LA is effective and efficient (Daxberger et al., 2021; Hobbhahn et al., 2022), yielding a Gaussian weight posterior without retraining. The full derivation of LA is provided in Appendix A.1.
>
> We hope this resolves the clarity issues while retaining correctness.

---

> > ### Author Response · Authors · 2025-11-21
> > **[Q2] Clarifying Why the Surrogate Loss Does Not Collapse to a One-Hot Solution**
> >
> > > **__Q2)__** The solution of eqn 3 should put probability 1 on the lowest value. Why do the authors claim the result in eqn 4?
> >
> > Thank you for raising this point; we understand why the original presentation could lead to this interpretation.
> >
> > We clarified the key issue:
> > **Eq. (3) is not simply minimizing a set of scalar values—it is a constrained convex program over the probability simplex, where soft normalization prevents collapse.**
> >
> > In the revised text, we now explicitly explain:
> > - **Why the minimizer does not collapse to a single model**:
> >     > **__(Section 3.2, L226~233)__** This convex linear objective combines variance and embedding magnitude into a tractable surrogate loss. Although the objective is linear, the probability simplex constraints yield a well-posed convex program with a unique closed-form solution under standard convex analysis (Boyd & Vandenberghe, 2004). This ensures that the resulting coefficients are well-defined and stable, avoiding degenerate behavior even under heterogeneous uncertainty across models. The solution therefore yields a soft inverse–risk weighting that distributes probability mass smoothly, rather than collapsing entirely onto the smallest term, leading to a more robust and interpretable ensemble. Its unique closed-form solution is the inverse-variance weighting:
> > - **Why inverse-variance weighting naturally arises**: The model-wise variance acts as a precision term. The normalization ensures stable combination rather than selecting only the smallest value.
> > - **Where the theory and intuition appear**: We added detailed intuition in:
> >     > **__(Section 3.2, L237~246)__** Intuitively, the inverse–variance rule assigns lower weight to embeddings whose large epistemic variance signals low confidence, while giving more influence to models that produce stable, self-consistent representations for the given query. This precision-based structure mirrors classical Bayesian aggregation, where estimators with higher certainty contribute proportionally more to the final prediction (Singh et al., 2005). Importantly, this mechanism also adapts to data-wise heterogeneity: for queries that fall into domains where certain encoders are weak or out-of-distribution, their variances naturally increase, causing UEC to down-weight them without requiring explicit domain labels. Under this interpretation, the resulting coefficient vector $π^∗_k(x)$ is precisely the minimizer of the uncertainty-aware surrogate loss in Eq. 3, representing the Bayes-optimal solution under the surrogate formulation. Full derivation appears in Appendix A.2.
> >
> > We also revised the relevant sentences to remove ambiguity and avoid suggesting that Eq. (3) behaves like an unconstrained scalar minimization. We hope the improved clarity resolves the reviewer’s concerns.

---

> > > ### Author Response · Authors · 2025-11-29
> > >
> > > Thank you again for the constructive feedback. We have posted our responses and corresponding clarifications during the discussion period. If there are any additional questions or points that would benefit from further explanation, we would be happy to address them.

---

### Author Response · Authors · 2025-11-21
**Global Response**

We sincerely appreciate all reviewers for their thoughtful and constructive feedback. We appreciate that the reviewers acknowledged our contributions in the following aspects:
1. **Principled and Well-Motivated Framework** : Reviewers highlighted that UEC introduces a clean, probabilistically grounded way to ensemble embedding models by converting deterministic embeddings into uncertainty-aware Gaussian representations and leveraging principled Bayes-optimal weighting.
2. **Strong Theoretical Foundations** : The derivation of Gaussian embedding posteriors via Laplace Approximation, the Bayes-optimal ensemble coefficients, and the ranking-preserving link between probit similarity and Wasserstein/Jeffreys distances were noted as rigorous, meaningful contributions.
3. **Practicality, Scalability, and Generality** : UEC was appreciated for being post-hoc, model-agnostic, computationally efficient, and directly applicable to large pretrained encoders—making the method both practical and broadly usable in real-world scenarios.
4. **Comprehensive Empirical Validation** : Reviewers acknowledged the extensive experiments across retrieval, classification, STS, bitext mining, clustering, and reranking on large multilingual benchmarks (MMTEB, MIRACL), along with clear ablations, calibration diagnostics, and sensitivity studies demonstrating consistent improvements over strong baselines.


During the rebuttal period, we have significantly revised and improved our manuscript by addressing the raised concerns. The major updates are as follows:
1. **Significant revision of Section 3 for clarity** : We thoroughly reorganized Section 3 to improve clarity, readability, and theoretical precision. This includes rewriting Section 3.1 to provide a high-level, intuition-focused explanation of Laplace Approximation while moving derivations to Appendix A.1, expanding the intuition in Section 3.2, clarifying the transition from Eq. (2) to Eq. (3), adding missing retrieval-time constraints, and strengthening the Bayes-optimal interpretation with improved derivations and explanations.
2. **Clarified surrogate-loss formulation and unique solution** : We added retrieval-specific justification for dropping document-dependent terms, cited relevant literature, explained why the resulting program remains convex and well-posed, and expanded the discussion of the inverse–variance weighting rule and its Bayesian interpretation.
3. **Strengthened empirical validation of uncertainty quality** : To validate uncertainty quality, we added a new uncertainty-diagnosis subsection in the main text and expanded Appendix C with extensive calibration analyses, including ECE for probabilistic embeddings and Var-ECE for similarity variance. Furthermore, we evaluated uncertainty robustness across multiple LA configurations—diagonal vs. KFAC covariance, deeper curvature modeling (1/2/4/6 layers), and extensive data sparsity settings—showing that uncertainty estimates remain stable and well-calibrated under all perturbations.
4. **Hyperparameter sensitivity analysis** : We added $\beta$ and temperature $T$ sensitivity studies across MIRACL and MMTEB (Appendix D), confirming that UEC is stable and the optimal ranges are consistent across datasets.
5. **Expanded multilingual and OOD robustness experiments** : We added multilingual evaluations (Appendix B.5) and conducted OOD tests, validating robustness when LA is fitted on MS MARCO and evaluated on MMTEB tasks.
6. **Analysis of independence and diagonal assumptions**: We clarified how cross-covariances would affect the optimal coefficients and added comparisons with KFAC-based covariance structures, noting that while KFAC slightly improves calibration, diagonal LA remains far more efficient.

We hope these revisions have clarified our contributions and addressed the reviewers’ questions. The full responses to each comment are provided individually and all revised portions of manuscript are highlighted in magenta for clarity.

Once again, we thank the reviewers for their valuable feedback, which has been instrumental in improving the quality and clarity of our work.

---

### Author Response · Authors · 2025-12-02
**[To New AC] Summary of Rebuttal**

**Dear New Area Chair**,
We sincerely regret the situation that has arisen and the additional burden it may have placed on you. To support an efficient decision process, we provide a concise summary of the rebuttal. More detailed point-by-point responses follow below.

-----
-----

## **Summary of Rebuttal**

To assist the AC in evaluating the revised manuscript, we emphasize that the rebuttal period led to **substantial improvements across theory, clarity, and empirical validation**. Many sections—especially Section 3, Appendix A, and Appendix C/D—were meaningfully rewritten, with multiple new experiments added. Collectively, these updates resolve every question raised during the review and significantly strengthen both the rigor and readability of the paper.

-----

### **1) Reviewer ed1F** (Original Score : 2)
#### **Initial Concerns**
Section 3 was difficult to follow—unclear explanation of how embeddings become probabilistic, how Eq.(2) leads to Eq.(3), and why Eq.(4) does not collapse to a one-hot solution.

#### **Resolutions**
- **Section 3 rewritten** with clearer intuition; full derivations moved to Appendix.
- **Retrieval-time assumption clarified**, explaining the Eq.(2) → Eq.(3) surrogate.
- **Inverse-variance solution justified** as a stable convex program, not a degenerate minimizer collapsed to one-hot solution.
- Jargon removed and intuitive explanations added throughout.

Although further discussion was not possible, we believe the revised section now reads clearly and resolves the reviewer’s concerns.

---

### **2) Reviewer TC4d** (Original Score : 4)
#### **Initial Concerns**
Questions on: uncertainty type, independence/diagonal assumptions, calibration, benchmark scope/OOD, novelty, deeper LA, and hyperparameter sensitivity.

#### **Resolutions**
- **Explicitly clarified that UEC models epistemic uncertainty only**; added this explicitly and updated the Conclusion to discuss aleatoric/predictive extensions.
- **Independence/diagonal assumption justified** by (i) low empirical cross-covariance; (ii) diagonal LA matches KFAC calibration with far lower cost (Appendix C.1).
- **Added calibration analyses** (ECE + Var-ECE), confirming **meaningful uncertainties** (Section 4.3; Appendix C).
- **Benchmark scope clarified** (MMTEB already multilingual/multi-domain) (Appendix B.5) and added a new explicit OOD test.
- **Novelty articulated**: first (i) post-hoc probabilistic embedding ensemble, (ii) new surrogate-loss, and (iii) similarity linked to Wasserstein/Jeffreys theoretically.
- **Deeper LA shows negligible performance gain** with substantial computation cost (Appendix C.1).
- **Hyperparameter studies** show **minimal sensitivity and stable performance** (Appendix D).
- **Ranking-preservation validated** for probit similarity.

These revisions provide clearer theoretical grounding and stronger empirical support for UEC by addressing all reviewers' concerns and questions.

---

### 3) **Reviewer jJ7t** (Original Score : 4)
#### **Initial Concerns**
Concerns about novelty relative to existing tools, precision of the “Bayes-optimal” claim, calibration, and additional baselines.

#### **Resolutions**
- **Novelty clarified** across three dimensions: (i) post-hoc probabilistic ensemble, (ii) surrogate-loss for optimal weights, and (iii) Wasserstein-linked similarity.
- **Bayes-optimal phrasing refined** with clearer justification in Section 3.
- **Added calibration (ECE + Var-ECE)** confirming epistemic meaning of variances (Section 4.3; Appendix C).
- **Compared against BMA/uncertainty-fusion**, showing **UEC is more accurate and efficient**.

---

### 4) **Reviewer ivFt** (Original Score : 8)
#### **Initial Concerns**
Clarity of Section 3, hyperparameter sensitivity, comparison with LLM embeddings, performance with stronger models, and robustness under data sparsity.

#### **Resolutions**
- **Section 3 clarified** with intuition-focused explanations.
- **Sensitivity analysis** for $\beta$ and temperature added (Appendix D).
- **LLM comparison strengthened**, showing UEC consistently improves over generalist models.
- **Stronger-model ensemble test** (incl. GTE-MB) shows further gains.
- **Data-sparsity robustness** confirmed with minimal LA-fit samples.

**Rebuttal Status**: The reviewer confirmed all concerns were fully addressed.

---
---

## **Conclusion**
We sincerely thank all reviewers for their constructive feedback. Their comments significantly improved the manuscript. We carefully addressed ***all*** concerns through clearer theory, stronger empirical validation, and extended experiments. Although deeper dialogue was not possible, we hope the Area Chair will consider the substantial improvements reflected in the revised submission.

---

### Meta-Review · Area_Chair_RTQ3 · 2026-01-07

**Summary:**

This paper proposes a post-hoc, uncertainty-aware ensembling pipeline for text embeddings: convert deterministic embeddings to Gaussian (via last-layer Laplace approximation), compute uncertainty-informed ensemble weights from a retrieval-feasible surrogate objective, and use an uncertainty-aware (probit) similarity that approximates distributional distances. Reviewers generally agree the approach is practical and empirically strong across many embedding benchmarks, but raised concerns about (i) clarity and theoretical presentation in Section 3, (ii) strength/scope of “Bayes-optimal” claims given surrogate approximations, (iii) calibration/meaningfulness of the uncertainty estimates, and (iv) simplifying independence/diagonal covariance assumptions. The rebuttal and revised draft appear to improve clarity, add calibration and sensitivity analyses, and better scope the claims.

Overall, the paper looks like a solid poster candidate: deployable method, broad evaluation, and most substantive concerns addressed, with remaining limitations largely about assumptions and positioning rather than correctness. Therefore, I recommend acceptance.

**Reviewer Concerns:**

Addressed in rebuttal / revision:
* Clarity of Section 3: authors rewrote Section 3 with more intuition and moved derivations to appendix; they added explicit retrieval-time justification for dropping document-dependent terms.
* Calibration / whether variances are meaningful: authors added explicit calibration diagnostics (ECE for probabilistic embeddings; a variance-calibration metric for similarity variance) and report improved calibration for LA and best calibration for UEC.
* Hyperparameter sensitivity: authors added beta/temperature sweeps (MIRACL subset + MMTEB example) and claim stable optimal ranges.
* Missing baselines: authors added/compared against BMA-style sampling from LA and an uncertainty-fusion baseline; report UEC wins with lower overhead.
* Stronger models: authors added an experiment ensembling stronger GTE-MB and report further gains.

Still outstanding (or only partially resolved):
* Assumptions (independence across models, diagonal covariance): authors provide empirical evidence of low cross-model correlation and compare diagonal vs KFAC for calibration/cost, but the general validity remains a limitation and should be clearly acknowledged.
* Scope of uncertainty: only epistemic uncertainty via last-layer LA; no aleatoric/predictive uncertainty; deeper LA tested but only limited settings and gains appear small.
* Novelty: still primarily an integration paper (well-executed), so acceptance hinges more on practical impact + thorough evaluation than on fundamentally new theory.
* Fairness/bias risk: acknowledged conceptually; no direct empirical analysis. I would not treat this as an ethics blocker, but it should be framed as a limitation.

**Reviewer Scores:**

* ed1F: original 2. They were flagged by AC for low clarity in review; authors made major clarity and derivation edits. I’d expect an increase, but not necessarily to accept. Estimated final: 3-4.
* TC4d: original 4. Many concerns (calibration, OOD, assumptions, sensitivity) were directly addressed with new experiments/analyses. Estimated final: 5-6.
* jJ7t: original 4. Main issues were novelty framing, Bayes-optimal phrasing, calibration, additional baselines; all addressed. Estimated final: 5-6.
* ivFt: original 8. They confirmed their concerns were addressed. Estimated final: stays 8.

---

### Decision · Program_Chairs · 2026-01-26

Accept (Poster)